# CTCF depletion decouples enhancer-mediated gene activation from chromatin hub formation

Magdalena A. Karpinska ●[1,2,6], Yi Zhu ●[1,2,6], Zahra Fakhraei Ghazvini[1,2], Shyam Ramasamy[1,2], Mariano Barbieri[3], T. B. Ngoc Cao ●[1,2], Natalie Varahram[1,2], Abrar Aljahani[1,2,5], Michael Lidschreiber ●[4], Argyris Papantonis ●[3] & A. Marieke Oudelaar ●[1] ✉

Enhancers and promoters interact in three-dimensional (3D) chromatin structures to regulate gene expression. Here we characterize the mechanisms that drive the formation and function of these structures in a lymphoid-to-myeloid transdifferentiation system. Based on analyses at base pair resolution, we demonstrate a close correlation between binding of regulatory proteins, formation of chromatin interactions and gene expression. Multi-way interaction analyses and computational modeling show that tissue-specific gene loci are organized into chromatin hubs, characterized by cooperative interactions between multiple enhancers, promoters and CTCF-binding sites. While depletion of CTCF strongly impairs the formation of these chromatin hubs, the effects of CTCF depletion on gene expression are modest and can be explained by rewired enhancer–promoter interactions. These findings demonstrate a role for enhancer–promoter interactions in gene regulation that is independent of cooperative interactions in chromatin hubs. Together, these results contribute to our understanding of the structure–function relationship of the genome during cellular differentiation.

Precise spatiotemporal regulation of gene expression during differentiation and development is dependent on the *cis*-regulatory elements of the genome, which include enhancers and promoters. Active enhancers recruit transcription factors and coactivators, which stimulate assembly and activation of the transcription machinery at gene promoters[1,2]. Because mammalian enhancers can be separated by large genomic distances from their target gene promoters, enhancers interact with promoters in 3D chromatin structures to activate gene expression[3,4]. Interactions between enhancers and promoters predominantly occur within topologically associating domains (TADs)[5], which are relatively

insulated regions of the genome that are demarcated by CCCTC-binding factor (CTCF)-binding sites (CBSs) and formed by cohesin-mediated loop extrusion[6–8]. Although the general importance of loop extrusion for gene regulation remains unclear, it has been shown that loop extrusion contributes to the formation and specificity of (long-range) enhancer–promoter interactions in some contexts[9,10]. In addition, affinity between transcription factors and coactivators bound at enhancers and promoters is thought to have a role in the formation of specific interactions between these elements[11,12]. It has been suggested that these interactions form in the context of nuclear condensates, which

[1]Genome Organization and Regulation, Max Planck Institute for Multidisciplinary Sciences, Göttingen, Germany. [2]Georg August University of Göttingen, Göttingen, Germany. [3]Institute of Pathology, University Medical Center Göttingen, Göttingen, Germany. [4]Department of Molecular Biology, Max Planck Institute for Multidisciplinary Sciences, Göttingen, Germany. [5]Present address: Department of Developmental Biology, Stanford University, Stanford, CA, USA. [6]These authors contributed equally: Magdalena A. Karpinska, Yi Zhu. ✉e-mail: marieke.oudelaar@mpinat.mpg.de

are dependent on multivalent interactions between intrinsically disordered regions of transcription factors, coactivators, the transcription machinery and/or RNA molecules[13–16].

Although our understanding of the 3D structures into which the genome is organized has advanced over the last decades, the relationship between genome structure and function remains unclear. In particular, it is incompletely understood when and how enhancer–promoter interactions form during cellular differentiation and how they influence gene expression. The structure–function relationship of the genome has been studied in several developmental contexts[17–30]. These studies have identified both instructive enhancer–promoter interactions, which co-occur with active gene expression, and permissive interactions, which are formed before gene activation[31]. However, because these studies are based on relatively low-resolution analyses of interactions between enhancers and promoters and do not take their 3D configuration in higher-order structures into consideration, the precise relationship between enhancer–promoter interactions and gene activation as well as the underlying molecular mechanisms remain poorly understood. In addition, it is not clear whether permissive, preformed interactions represent a distinct mode of enhancer-mediated gene activation or reflect instructive enhancer–promoter interactions that are dependent on subtle changes during cellular differentiation that cannot be detected with low-resolution analyses. Importantly, it has recently been demonstrated that small changes in genome structure can have a big effect on gene expression[32,33]. A better understanding of the structure–function relationship of the genome during cellular differentiation therefore requires analysis at very high resolution and sensitivity. In addition, the integration of perturbations during differentiation could facilitate causal inference and identification of the molecular mechanisms involved.

The analysis of enhancer–promoter interactions in the above-mentioned studies is based on chromosome conformation capture (3C) techniques, which rely on digestion and subsequent proximity ligation of cross-linked chromatin to detect spatial proximity between DNA sequences[34]. The resolution of 3C predominantly depends on the digestion and sequencing strategy[35]. By combining digestion with micrococcal nuclease (MNase), which cuts the genome largely independent of DNA sequence, with deep, targeted sequencing of multiplexed viewpoints of interest, the recently developed Micro-Capture-C (MCC) method supports 3C analysis at single base pair resolution[36]. Despite their superior resolution, MCC data cannot resolve how *cis*-regulatory elements interact together in higher-order 3D chromatin structures, because MCC is based on the detection of pairwise interactions in a cell population and therefore cannot distinguish simultaneous, cooperative interactions in individual cells from mutually exclusive interactions that occur independently in different cells. Three-dimensional relationships between multiple *cis*-regulatory elements can be disentangled by multi-way 3C techniques, in which multiple interactions derived from individual alleles are captured within (relatively) long sequencing reads[37–42], and by ligation-free techniques based on physical separation and labeling of cross-linked chromatin interactions[43,44]. Among these, the Tri-C method provides an effective approach to study multi-way enhancer–promoter interactions during cellular differentiation, as it allows for targeted analysis of multiplexed viewpoints of interest at a relatively high resolution (500–5,000 bp)[37]. Thus far, Tri-C and other targeted multi-way 3C techniques have only been used to study a few genetic loci and have not been combined with perturbations of regulatory proteins to study the mechanisms underlying multi-way chromatin interactions. In addition, neither MCC nor Tri-C have been applied throughout subsequent stages of differentiation and development.

In this study, we provide important insights into the relationship between genome structure and function by combining MCC and Tri-C experiments with analyses of nascent gene expression, chromatin accessibility and binding of regulatory proteins during cellular differentiation in control and CTCF-depleted conditions.

## Results

### Chromatin architecture through cellular differentiation

To characterize the structure–function relationship of the genome, we used a lymphoid-to-myeloid transdifferentiation system based on a B cell leukemia cell line (BLaER1) that can be efficiently converted into functional induced macrophages (iMacs) by exogenous expression of the transcription factor CCAAT enhancer-binding protein α (CEBPA) over the course of 168 h[45] (Fig. 1a and Extended Data Fig. 1a,b). Previous studies have analyzed nascent RNA synthesis by transient transcriptome sequencing (TT-seq)[46]; chromatin accessibility by assay for transposase-accessible chromatin with high-throughput sequencing (ATAC–seq)[46]; the distribution of histone H3 lysine 27 (H3K27) acetylation (H3K27ac), H3K27 trimethylation (H3K27me3) and CTCF by chromatin immunoprecipitation followed by sequencing (ChIP–seq)[47]; and genome organization by Hi-C[47] during BLaER1 transdifferentiation. We complemented these data by mapping the binding profiles of Mediator and cohesin using an optimized ChIPmentation protocol[48] at 0 h, 12 h, 24 h, 72 h and 96 h after differentiation induction (Methods; later time points were omitted as iMacs are quiescent). In addition, we generated Capture-C and MCC interaction profiles from the viewpoints of 51 promoters of B cell-specific and iMac-specific genes at 0 h, 24 h and 96 h (Fig. 1b–e, Extended Data Fig. 1c–e and Supplementary Table 1).

Integration of these high-resolution data provides insight into the timing of regulatory events and their relationship during cellular differentiation. *TRIB1* provides an example of a gene locus that is upregulated during transdifferentiation. Our data show that this is associated with a gradual increase in accessibility and occupancy of H3K27ac, Mediator and cohesin at regulatory regions, whereas CTCF binding is generally more stable (Fig. 1d and Extended Data Fig. 1c). As previously described, we observe strong cohesin accumulation at CBSs[49] and slightly weaker cohesin enrichment at Mediator-bound sites[50]. In addition, we observe changes in the structural conformation of the locus during differentiation in both the Capture-C and MCC data, which involve a gradual increase in chromatin interactions between the *TRIB1* promoter and putative enhancer elements, CBSs and other promoters in a ~600-kb TAD. The high resolution of the MCC data allows for distinguishing interactions between *cis*-regulatory elements in close proximity and facilitates systematic calling of chromatin interactions[36]. Quantification of these interactions across the targeted loci shows that the described interaction pattern for the *TRIB1* locus is representative for upregulated loci (Fig. 1f).

In contrast, we observe that deactivation of *MYB* is associated with a reduction in chromatin accessibility, H3K27ac and Mediator and cohesin binding at most putative enhancer elements in the ~400-kb TAD (Fig. 1e and Extended Data Fig. 1d). The interactions between the *MYB* promoter and these decommissioned elements gradually decreases as the cells transition into iMacs. Systematic quantification of chromatin interactions across the targeted loci shows a similar pattern for other downregulated loci (Fig. 1g). However, in contrast to upregulated loci, we do not observe consistent changes in promoter interactions with CBSs and other promoters in downregulated loci during differentiation (Fig. 1g). Interestingly, we also observe *cis*-regulatory elements that interact more frequently with the *MYB* promoter as it is deactivated. These elements are not characterized by repressive chromatin marks such as H3K27me3 (Extended Data Fig. 1d). Instead, they transiently gain chromatin accessibility, H3K27ac and binding of Mediator and cohesin. We therefore speculate that these elements may function as transient enhancers that modulate the kinetics of gene silencing, as recently described in the context of erythroid differentiation[51], although further characterization is required to confirm this. We observe elements with similar characteristics in approximately 60% of the targeted downregulated loci.

The *TRIB1* and *MYB* interaction patterns suggest that the temporal dynamics of gaining and losing Mediator and cohesin binding

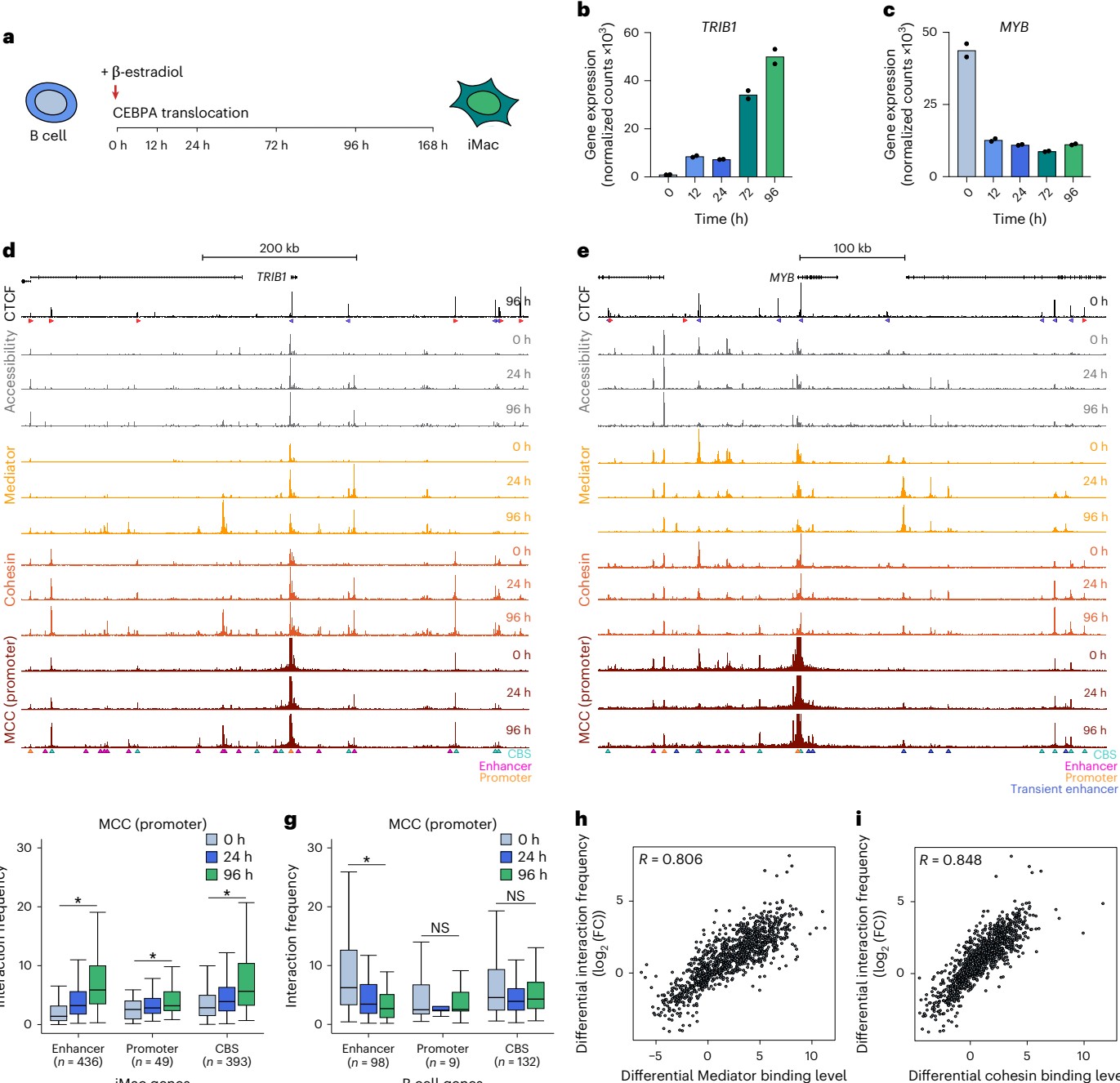

**Fig. 1 | Chromatin architecture through lymphoid-to-myeloid transdifferentiation. a**, Schematic overview of the BLaER1 lymphoid-to-myeloid transdifferentiation system. **b**, *TRIB1* expression through transdifferentiation, derived from TT-seq data. Bars represent the average of two biological replicates; corresponding data points are shown as dots. **c**, *MYB* expression through transdifferentiation, as in **b**. **d**, Chromatin landscape of the *TRIB1* locus (chr8:125,079,965–125,739,965; 660 kb; hg38) through transdifferentiation. From top to bottom: gene annotation, CTCF ChIP–seq, ATAC–seq, Mediator complex subunit 26 (MED26) ChIPmentation, structural maintenance of chromosomes 1A (SMC1A) ChIPmentation, MCC data from the viewpoint of the *TRIB1* promoter. Axes are scaled to signal with the following ranges: CTCF, 0–12,178; accessibility, 0–8,276; Mediator, 0–5,726; cohesin, 0–2,129; MCC, 0–40. Orientations of CTCF motifs are indicated by arrowheads (forward in red, reverse in blue). MCC interactions with CBSs, enhancers and promoters are annotated with cyan, magenta and orange triangles, respectively. **e**, Chromatin landscape of the *MYB* locus (chr6:134,992,474–135,472,474; 480 kb; hg38), as in **d**, with MCC interactions with transient enhancers annotated with blue triangles and the

following axis ranges: CTCF, 0–6,052; accessibility, 0–3,506; Mediator, 0–2,277; cohesin, 0–1,247; MCC, 0–30. **f**, Interaction frequencies of promoters of iMac-specific genes with enhancers, promoters and CBSs through transdifferentiation. Data are derived from three biological replicates; the number of data points (*n*) in each category is shown in the figure. Box plots show the interquartile range and median of the data; whiskers indicate the minima and maxima within 1.5 × interquartile range; asterisks indicate statistical significance (two-sided paired Wilcoxon signed-rank test, 96 h versus 0 h); enhancer, $P < 2.2 \times 10^{-16}$; promoter, $P = 5.6 \times 10^{-5}$; CBS, $P < 2.2 \times 10^{-16}$. **g**, Interaction frequencies of promoters of B cell-specific genes, as in **f**; enhancer, $P = 5.0 \times 10^{-10}$; promoter, $P = 0.83$; CBS, $P = 0.21$. NS, not significant. **h**, Correlation between differential enhancer–promoter interaction frequencies and differential Mediator binding levels at the interacting elements (24 h versus 0 h and 96 h versus 0 h), based on Spearman's correlation test. FC, fold change. **i**, Correlation between differential enhancer–promoter interaction frequencies and differential cohesin binding levels, as in **h**.

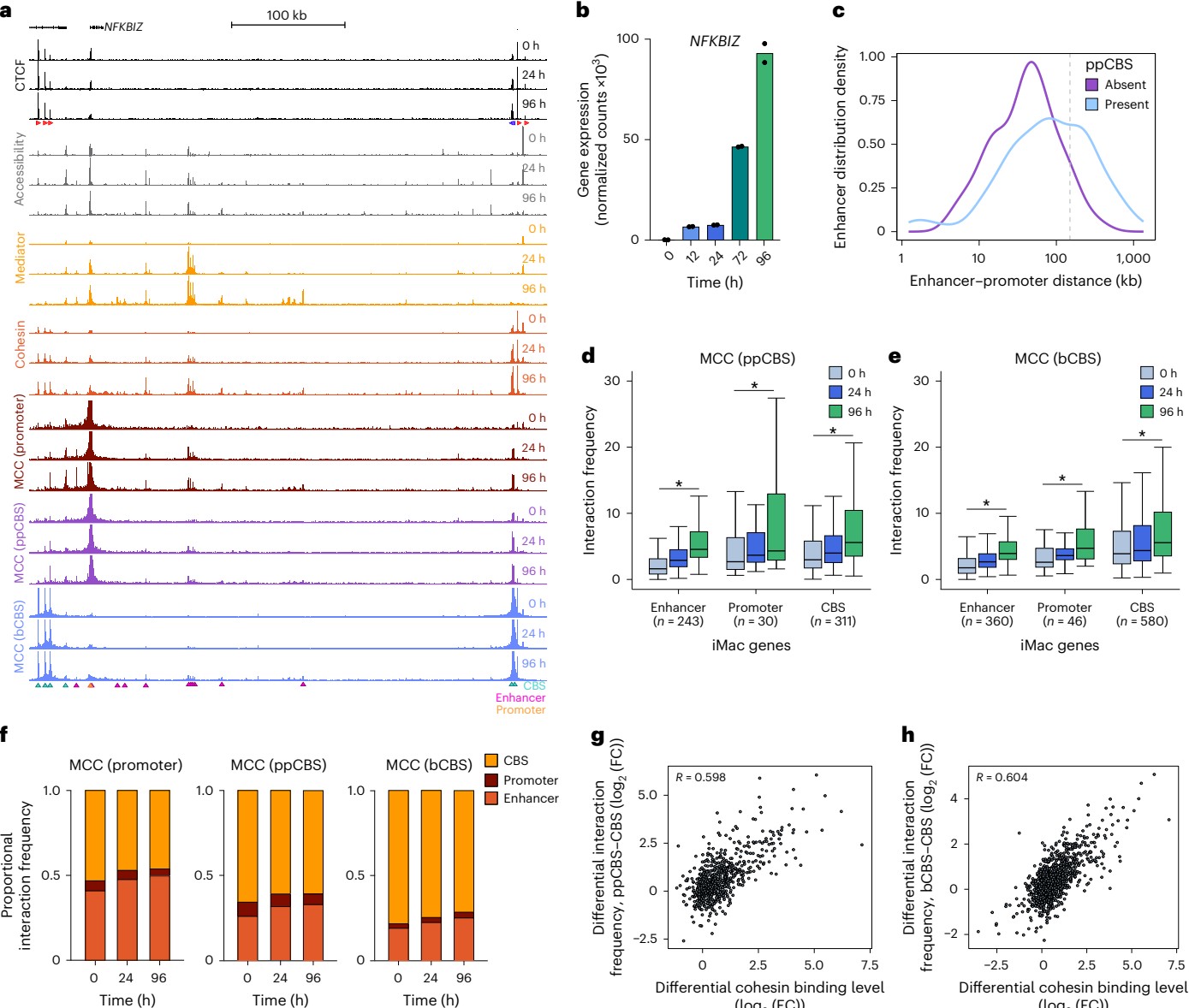

**Fig. 2 | Interaction patterns across classes of *cis*-regulatory elements.**
**a**, Chromatin landscape of the *NFKBIZ* locus (chr3:101,795,424–102,255,424; 460 kb; hg38), as in Fig. 1d, with MCC data from the viewpoint of the promoter, ppCBS and bCBS. Axes are scaled to signal with the following ranges: CTCF, 0–5,547; accessibility, 0–4,883; Mediator, 0–2,252; cohesin, 0–1,773; MCC, 0–40. **b**, *NFKBIZ* expression through transdifferentiation, as in Fig. 1b. **c**, Distribution of enhancer–promoter distances of promoters with and without a ppCBS (±5 kb from the promoter). The gray line marks the 150-kb threshold used to classify distal enhancers in Extended Data Fig. 2c. **d**, Interaction frequencies of ppCBSs of

iMac-specific genes, as in Fig. 1f; enhancer, $P < 2.2 \times 10^{-16}$; promoter, $P = 2.7 \times 10^{-5}$; CBS, $P < 2.2 \times 10^{-16}$. **e**, Interaction frequencies of bCBSs of iMac-specific genes, as in Fig. 1f; enhancer, $P < 2.2 \times 10^{-16}$; promoter, $P = 6.4 \times 10^{-10}$; CBS, $P < 2.2 \times 10^{-16}$. **f**, Comparison of the proportion of interactions of promoters, ppCBSs and bCBSs with enhancers, promoters and CBSs in iMac-specific gene loci. **g**, Correlation between differential interaction frequencies between ppCBS and CBSs and differential cohesin binding levels, as in Fig. 1h. **h**, Correlation between differential interaction frequencies between bCBS and CBSs and differential cohesin binding levels, as in Fig. 1h.

at enhancer elements correspond closely to the strengthening and weakening of enhancer–promoter interactions over the course of differentiation (Fig. 1d,e). Quantification of these patterns across all targeted loci shows a strong correlation between differential binding levels of Mediator and cohesin and enhancer–promoter interaction frequencies, signified by correlation coefficients of 0.81 and 0.85, respectively (Fig. 1h,i). Interestingly, we find that H3K27ac and enhancer RNA (eRNA) levels, which are more commonly used as proxies for enhancer activity, have a weaker correlation with enhancer–promoter interactions, with coefficients of 0.59 and 0.50, respectively (Extended Data Fig. 1f,g). This may be explained by a lower precision and dynamic range of H3K27ac compared to Mediator and cohesin binding and technical

challenges in robustly identifying eRNA levels. Together, the integration of high-resolution 3C and ChIP data allows us to characterize the 3D *cis*-regulatory landscapes of targeted tissue-specific gene loci through lymphoid-to-myeloid transdifferentiation in detail (Extended Data Fig. 1h–k).

## Interaction patterns of classes of *cis*-regulatory elements

Many of the gene promoters that we targeted for MCC analysis have a CBS within 5 kb. It has previously been suggested that such promoter-proximal CBSs (ppCBSs) promote long-range enhancer–promoter communication[52] and may have a distinct function compared to CBSs at TAD boundaries (boundary CBSs; bCBSs)[53]. However, the

interaction patterns of ppCBSs have not been described, as they cannot be distinguished from those of promoters by most 3C techniques. Because MCC supports base pair-resolution analysis, we leveraged this approach to compare the interaction profiles from the viewpoints of promoters and ppCBSs (if present) in the targeted loci over the course of lymphoid-to-myeloid transdifferentiation. In addition, we generated interaction profiles from the viewpoints of bCBSs in all selected loci.

The *NFKBIZ* and *ARL4C* loci show representative interaction patterns for the promoter, ppCBS and bCBS viewpoints in upregulated and downregulated loci, characterized by relatively long-range enhancer–promoter interactions (Fig. 2a,b and Extended Data Fig. 2a–e). To investigate the relationship between the presence of a ppCBS and long-range enhancer–promoter communication, we compared the distance distribution of enhancers interacting with gene promoters with and without a ppCBS (Fig. 2c). This analysis confirms that promoters with a ppCBS form interactions with enhancers over larger distances compared to promoters without a ppCBS. In addition, we compared the frequencies of long-range (>150 kb) enhancer–promoter interactions to enhancer–promoter interactions across all distances in iMac-specific loci with and without a ppCBS over the differentiation course (Extended Data Fig. 2c). This analysis shows that genes without a ppCBS form stronger interactions with enhancers overall compared to genes with a ppCBS. These interactions are present to some degree in B cells and further strengthen over the differentiation course. By contrast, the loci containing ppCBSs form stronger long-range interactions, which are not pre-established and form specifically during differentiation.

The interaction profiles from the ppCBS viewpoints in the *NFKBIZ* and *ARL4C* loci show that ppCBSs form interactions with the enhancers in the region, although these are weaker compared to the promoter viewpoints, and with other CBSs in the region, which are stronger compared to the promoter viewpoints. The bCBS viewpoints most strongly interact with the CBSs at the other TAD boundary but also form weak interactions with the enhancers and promoters in the region. These interactions all gradually increase over the differentiation course. Systematic quantification of chromatin interactions in the targeted loci shows a similar pattern across the upregulated loci (Fig. 2d,e). Comparison of these data to the interaction patterns of gene promoters (Fig. 1f,g) shows that promoter interactions are generally more dynamic over the differentiation course compared to ppCBS and especially bCBS interactions. For a more direct comparison of interaction patterns across the classes of *cis*-regulatory elements, we calculated changes in the relative proportions of interactions over the differentiation course (Fig. 2f and Extended Data Fig. 2f). This comparison shows that enhancers interact more frequently with ppCBSs than with bCBSs, which suggests that ppCBSs may have a direct role in bringing enhancers in the vicinity of gene promoters. In addition, these analyses indicate that promoters, ppCBSs and bCBSs interact promiscuously with active *cis*-regulatory elements within TADs, although they do have distinct preferences for specific types of elements. The common feature at

these *cis*-regulatory elements is enrichment of cohesin, suggesting a role for loop extrusion in mediating these interactions. In line with this hypothesis, we find that differential cohesin binding correlates well with the interaction frequencies between CBSs (Fig. 2g,h), with coefficients of 0.6. These correlations are lower compared to enhancer–promoter interactions (Fig. 1i), which may be explained by the larger dynamic range in interactions between enhancers and promoters through differentiation compared to CBSs. Of note, we find that CBS interaction frequencies correlate better with cohesin binding than with CTCF binding itself (Extended Data Fig. 2g,h), which is in agreement with the more constitutive binding patterns of CTCF.

### Dynamic hub formation and dissolution during differentiation

The MCC data show complex interaction patterns involving multiple promoters, enhancers and CBSs in the targeted regions. However, because MCC is based on the detection of pairwise interactions, these data cannot resolve how these elements interact together in higher-order 3D chromatin structures. To analyze 3D interactions between *cis*-regulatory elements, we used the Tri-C technique to measure multi-way interactions in the 51 targeted regions of interest. Tri-C uses the restriction enzyme NlaIII for chromatin digestion and therefore generates lower-resolution data compared to MCC, which complicates direct quantitative comparisons between these datasets. Because Tri-C requires the restriction fragments at the targeted viewpoints to be relatively small (<300 bp) to facilitate efficient detection of multiple interactions[37], we targeted enhancers instead of promoters to assess multi-way enhancer–promoter interactions, as this provides more flexibility to select suitable viewpoints. In addition to enhancers, we targeted bCBSs in the selected regions. The Tri-C interactions are represented in viewpoint-specific contact matrices, in which the frequencies with which two chromatin fragments interact simultaneously with the viewpoint are plotted (Fig. 3a). Mutually exclusive interactions are depleted from these matrices, whereas preferential simultaneous interactions between *cis*-regulatory elements in higher-order structures are visible as enrichments at the intersections between these elements. Note that these matrices typically show strong signals along the proximity-excluded region at the viewpoint, which represent viewpoint-proximal interactions throughout the targeted region. As regions in close genomic proximity are expected to form strong interactions, these stripe patterns do not represent specific higher-order chromatin conformations.

The *CCR1* and *RAG2* loci provide representative examples of the establishment and dissolution of higher-order chromatin structures through lymphoid-to-myeloid transdifferentiation (Fig. 3b–e). At the *CCR1* locus, we observe the formation of a hub structure, characterized by simultaneous interactions between the enhancer viewpoint, the *CCR1* promoter and other enhancer clusters in the region (Fig. 3d, top matrix, cyan circles). Interestingly, we observe that the CBSs in the region are included in these hubs (Fig. 3d, top matrix, magenta

**Fig. 3 | Dynamic chromatin hub formation and dissolution during differentiation. a**, Schematic overview of Tri-C data visualization. Viewpoint (VP)-specific contact matrices show the frequencies at which two regions interact simultaneously with the viewpoints. Proximity signals around the viewpoint are excluded. Quantified regions for enhancer viewpoints include three-way interactions involving two enhancers and a promoter (E–E–P hubs) and three-way interactions involving an enhancer, a CBS and any other *cis*-regulatory element (E–C–X hubs). Quantified regions for the CBS viewpoints include three-way interactions involving three CBSs (C–C–C hubs) and three-way interactions involving a CBS and any combination of enhancers and/or promoters (C–E/P hubs). **b**, *CCR1* expression through transdifferentiation, as in Fig. 1b. **c**, *RAG2* expression through transdifferentiation, as in Fig. 1b. **d**, Tri-C contact matrices of the *CCR1* locus (chr3:45,902,299–46,427,299; 525 kb; 2.5-kb resolution; hg38) through transdifferentiation. The top right matrix shows Tri-C data from the viewpoint of an enhancer; the bottom left matrix shows the viewpoint of a bCBS. Viewpoints are indicated with white triangles and black arrows. E–E–P contacts

are highlighted with cyan circles, E–C–X and C–E/P contacts are shown with magenta circles, and C–C–C contacts are shown with dark blue circles. Profiles below show CTCF ChIP–seq, MED26 ChIPmentation and SMC1A ChIPmentation. Axes are scaled to signal with the following ranges: CTCF, 0–10,069; cohesin, 0–2,960; Mediator, 0–6,075. **e**, Tri-C contact matrices of the *RAG2* locus (chr11:36,486,822–36,756,822; 170 kb; 1.5-kb resolution; hg38), as in **d**, with the following axis ranges: CTCF, 0–14,993; cohesin, 0–2,868; Mediator, 0–2,831. **f**, Multi-way interaction frequencies of E–E–P and E–C–X hubs in iMac-specific loci through transdifferentiation, as in Fig. 1f; E–E–P hubs, $P = 1.2 \times 10^{-8}$; E–C–X hubs, $P = 4.5 \times 10^{-14}$. **g**, Multi-way interaction frequencies of C–E/P and C–C–C hubs in iMac-specific loci, as in Fig. 1f; C–E/P hubs, $P = 6.1 \times 10^{-5}$; C–C–C hubs, $P = 0.14$. **h**, Multi-way interaction frequencies of E–E–P and E–C–X hubs in B cell-specific loci, as in Fig. 1f; E–E–P hubs, $P = 2.4 \times 10^{-4}$; E–C–X hubs, $P = 1.3 \times 10^{-4}$. **i**, Multi-way interaction frequencies of C–E/P and C–C–C hubs in B cell-specific loci, as in Fig. 1f; C–E/P hubs, $P = 0.64$; C–C–C hubs, $P = 0.031$.

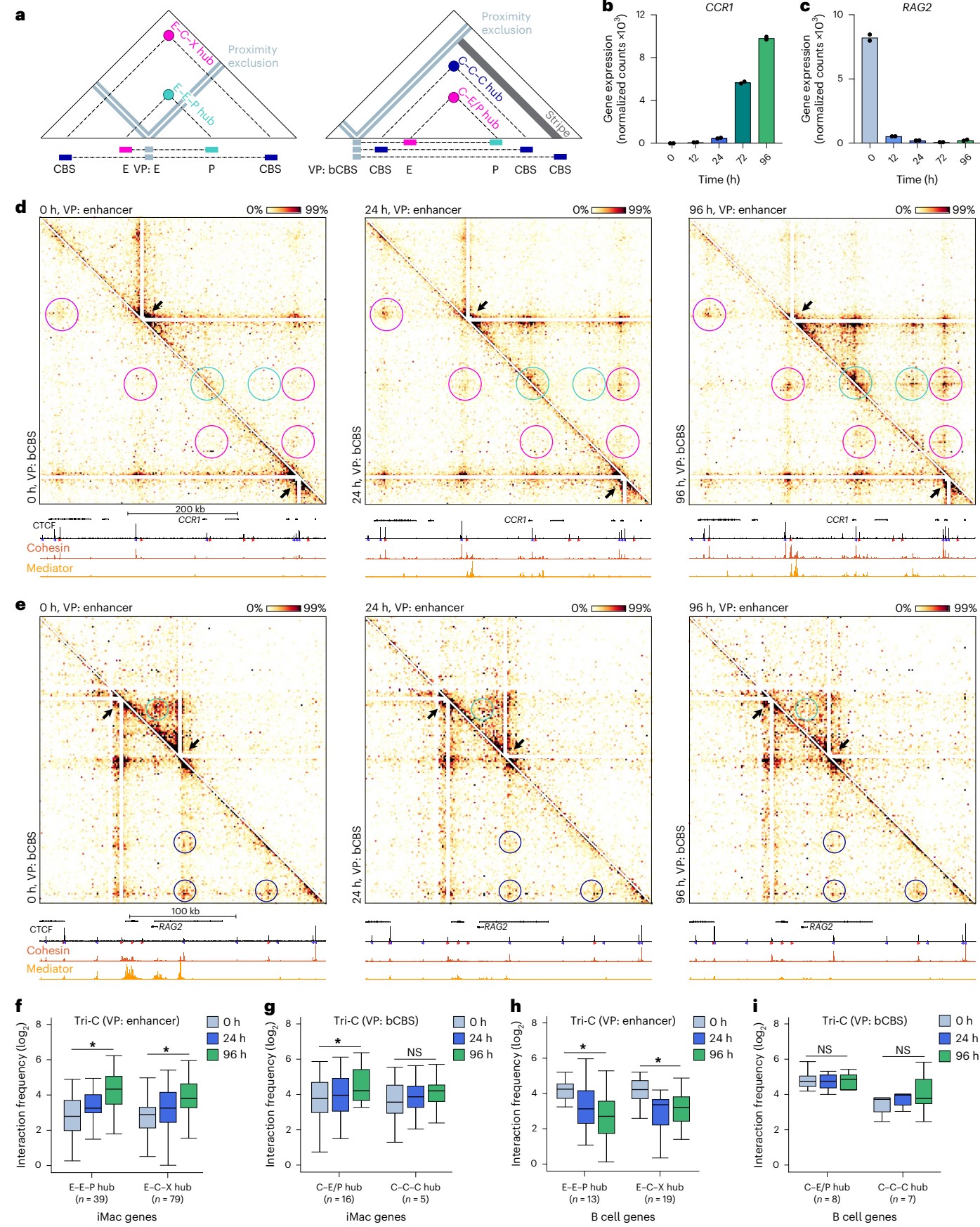

circles), which can also be appreciated from the contact matrices from the viewpoint of the CBS at the downstream boundary of the *CCR1* TAD (Fig. 3d, bottom matrix, magenta circles). Both the enhancer and the CBS viewpoint in the *CCR1* locus show that cooperative interactions between the promoter, enhancers and CBSs in chromatin hubs form gradually over the differentiation course. Tri-C data in the *NFKBIZ* and *TRIB1* loci (Extended Data Fig. 3a,b) and quantification of the multi-way chromatin interactions across all upregulated loci (Fig. 3f,g) show a similar pattern of gradual chromatin hub formation. Analysis of the CBS contact matrices across the targeted loci shows that these are characterized by distinct interaction foci as well as stripes that emanate from CBSs throughout the region and likely represent active loop extrusion by cohesin molecules (Fig. 3d,e and Extended Data Fig. 3a–d). We observe that loci with relatively little CTCF binding, such as *NFKBIZ*, are characterized by CBS stripes, whereas CTCF-dense regions, such as *CCR1*, also form clear CBS foci. The *RAG2* locus shows that cooperative interactions between enhancers and promoters gradually dissolve as *RAG2* is downregulated (Fig. 3e, top matrix, cyan circles). By contrast, multi-way CBS interactions are relatively stable over the differentiation course (Fig. 3e, bottom matrix, dark blue circles), which is in line with the relatively stable pairwise CBS interactions identified by MCC (Fig. 1e,g). Quantification of the multi-way chromatin interactions shows a similar pattern across the targeted downregulated loci (Fig. 3h,i). We detect multi-way chromatin interactions with enhancers and CBSs in the majority of the targeted upregulated and downregulated loci (Extended Data Fig. 3e,f). Together, the Tri-C data therefore show that the clustering of *cis*-regulatory elements into higher-order chromatin hubs is a common feature of tissue-specific gene loci, and that these hubs form and dissolve dynamically as genes are upregulated and downregulated during cellular differentiation.

### CTCF is not required for pairwise enhancer–promoter interactions

The incorporation of CBSs into chromatin hubs suggests that CTCF may have a function in the formation of these hubs and, more generally, in the regulation of enhancer–promoter communication during cellular differentiation. To test this, we performed 3C experiments in iMacs after auxin-mediated CTCF depletion[47] throughout the differentiation course (96 h; Fig. 4a). To confirm efficient depletion, we mapped CTCF binding in auxin- and control-treated iMacs (Extended Data Fig. 4a). In addition, we confirmed that the BLaER1 cells are still efficiently converted into iMacs in the absence of CTCF (Extended Data Fig. 4b).

To assess the effects of CTCF depletion on the gradual formation of enhancer–promoter interactions during lymphoid-to-myeloid transdifferentiation, we performed MCC with the gene promoters in the selected loci as viewpoints. Analysis of the *CCR1*, *NFKBIZ* and *TRIB1* loci shows that, as expected, CTCF depletion leads to a strong reduction in interactions between promoters and CBSs (Fig. 4b–d). By contrast, the interactions between promoters and enhancers in these loci are not strongly affected. As exemplified in the *TRIB1* and *SRGN* loci, we observe that many promoter–promoter interactions are increased after CTCF depletion, both within and beyond TAD boundaries (Fig. 4d,e). Systematic quantification of the MCC interactions shows similar patterns as in the described examples (Fig. 4f). Across upregulated loci, promoter–CBS interactions are weakened, enhancer–promoter interactions are relatively stable and promoter–promoter interactions are increased after CTCF depletion.

Given the strong correlation between differential Mediator and cohesin binding levels and enhancer–promoter interaction frequencies during differentiation, we performed additional ChIPmentation experiments after CTCF depletion to investigate whether we could explain the observed changes in interaction patterns by changes in the binding levels of Mediator and cohesin. As expected, CTCF depletion leads to a reduction in cohesin occupancy at CBSs (Fig. 4b–e).

By contrast, we observe minor changes in the distribution of Mediator and cohesin at enhancers, which do not correlate well with the minor changes we observe in enhancer–promoter interactions after CTCF depletion (Extended Data Fig. 4c,d).

### CTCF supports the formation of chromatin hubs

To characterize the role of CTCF in the formation of chromatin hubs during cellular differentiation, we performed Tri-C analysis, with the enhancers in the selected loci as viewpoints. In stark contrast to the minor effects of CTCF depletion on enhancer–promoter interactions measured by MCC, the Tri-C contact matrices of the *CCR1*, *NFKBIZ* and *TRIB1* loci show drastic changes in higher-order 3D chromatin structures following CTCF depletion (Fig. 5a–c). In the absence of CTCF, the formation of cooperative interactions between the targeted enhancers and other *cis*-regulatory elements in these regions is impaired. Unsurprisingly, cooperative interactions that involve a CBS are most strongly reduced. However, interestingly, cooperative interactions involving only enhancers and promoters are also decreased. This effect appears stronger in CBS-dense regions (for example, *CCR1*) compared to regions with relatively little CTCF binding (for example, *NFKBIZ*), although it is detectable across all targeted regions. The loss of these interactions is unlikely to reflect changes in the distribution of the pairwise interactions with the targeted enhancer viewpoint, as the MCC data show that enhancer–promoter interactions remain relatively stable (and in some cases even increase) after CTCF depletion (Fig. 4f).

We used molecular dynamics simulations of 3D chromatin folding in the *CCR1*, *NFKBIZ* and *TRIB1* loci to obtain a more comprehensive understanding of the effects of CTCF depletion. To this end, we built on our existing modeling framework[54,55] to model a 2-Mb region around each of these three genes (Methods). From these simulations, we extracted multi-way interactions with the same enhancers that were used as Tri-C viewpoints and produced contact matrices that closely resemble the experimental Tri-C results (Fig. 5a–f and Extended Data Fig. 5a–c). In the presence of CTCF, the targeted enhancers engage in multi-way interactions involving both their cognate gene promoter and other enhancers as well as CBSs. Removal of CTCF from the models does not significantly affect extracted pairwise interactions (Extended Data Fig. 5d,e) yet fully reproduces the impairment of chromatin hub formation across these loci (Fig. 5a–f and Extended Data Fig. 5f,g). The models therefore allow us to investigate the underlying mechanisms by interrogating the distribution and net flow of extruding cohesin molecules in the three loci. Consistent with the cohesin ChIPmentation data (Fig. 1d,e,i), the simulations show that cohesin accumulates strongly at CBSs and weakly at active enhancers and promoters in control conditions (Fig. 5d–f, bottom, green tracks). In the absence of strong CTCF-bound insulation sites, cohesin flows more freely (Fig. 5d–f, bottom, orange tracks), which results in a marked reduction of cohesin clustering to essentially only pairwise interactions (Fig. 5g–i). These analyses therefore suggest that, in the absence of relatively long-lived CTCF residence at CBSs, the spatiotemporal clustering of cohesin changes such that the formation of higher-order chromatin hubs is no longer supported. Importantly, the other upregulated loci show similar patterns as the *CCR1*, *NFKBIZ* and *TRIB1* loci. Cooperative interactions in CTCF-depleted iMacs are strongly reduced, with interactions directly involving CBSs (or elements in close vicinity to CBSs) most strongly affected (Fig. 5j and Extended Data Fig. 5h–m). Together, these results indicate that CTCF-mediated interactions provide a scaffold for the formation of chromatin hubs during cellular differentiation.

### Chromatin hubs are dispensable for gene regulation

To investigate the function of CTCF-dependent chromatin structures in gene regulation, we measured the effects of CTCF depletion during lymphoid-to-myeloid transdifferentiation on gene expression. In agreement with previous reports[47,56,57] and with the observation that BLaER1 cells still efficiently transdifferentiate into iMacs in the

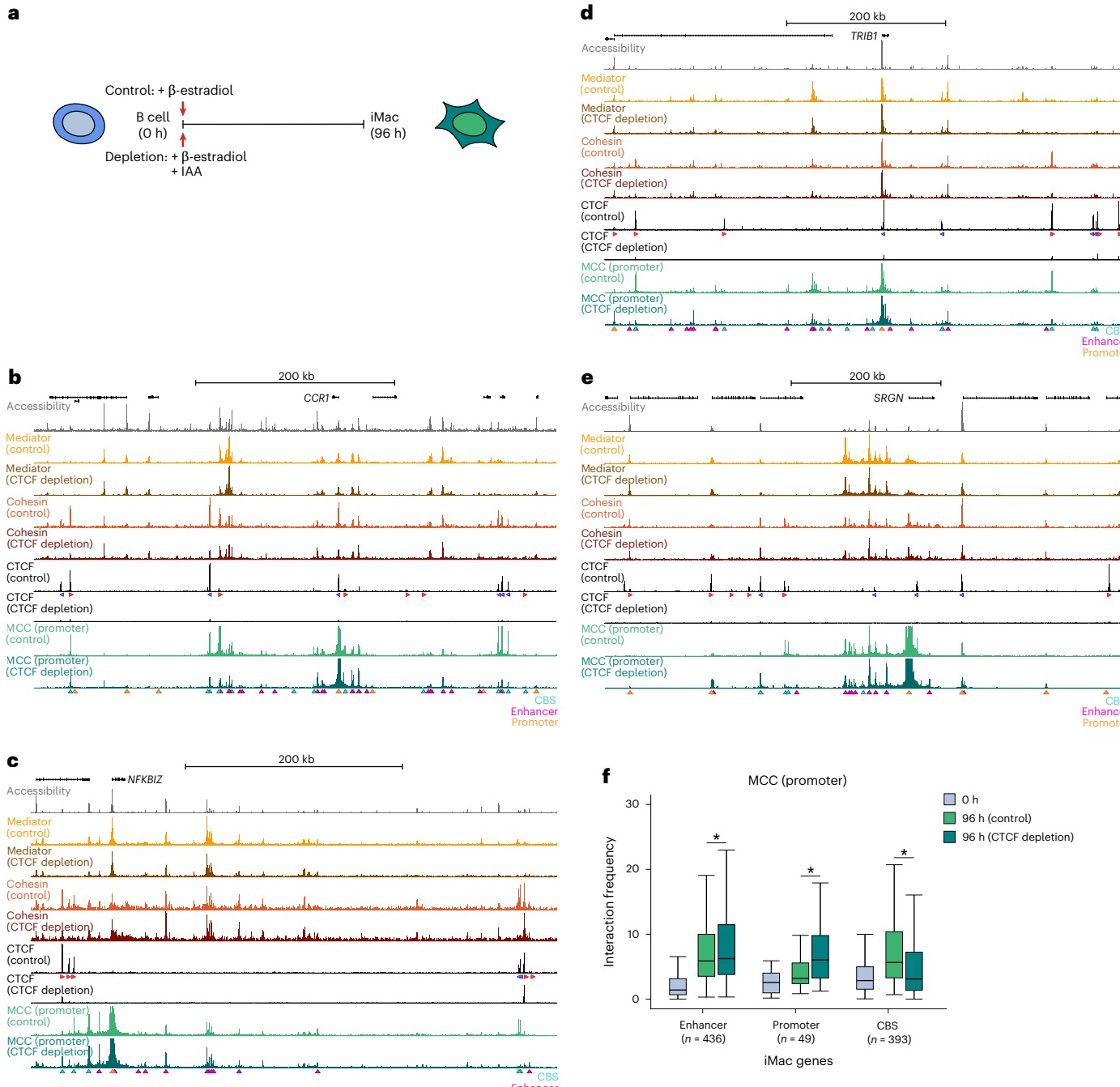

**Fig. 4 | CTCF is not required for pairwise enhancer–promoter interactions.**
**a**, Schematic overview of CTCF depletion during lymphoid-to-myeloid transdifferentiation. IAA, indole-3-acetic acid. **b**, Chromatin interactions in the *CCR1* locus (chr3:45,902,299–46,427,299; 525 kb; hg38) in control and CTCF-depleted cells at 96 h after differentiation induction. From top to bottom: gene annotation, ATAC–seq, MED26 ChIPmentation, SMC1A ChIPmentation, CTCF ChIPmentation, MCC data from the viewpoint of the promoter. Axes are scaled to signal with the following ranges: accessibility, 0–1,565; Mediator, 0–1,463; cohesin, 0–975; CTCF, 0–468; MCC, 0–40. Annotation as in Fig. 1d. **c**, Chromatin interactions in the *NFKBIZ* locus (chr3:101,766,932–102,266,932; 500 kb; hg38), as in **b**, with the following axis ranges: accessibility, 0–4,883; Mediator,

0–760; cohesin, 0–367; CTCF, 0–286; MCC, 0–40. **d**, Chromatin interactions in the *TRIB1* locus (chr8:125,079,965–125,739,965; 660 kb; hg38), as in **b**, with the following axis ranges: accessibility, 0–8,276; Mediator, 0–1,858; cohesin, 0–1,660; CTCF, 0–468; MCC, 0–40. **e**, Chromatin interactions in the *SRGN* locus (chr10:68,884,514–69,234,514; 350 kb; hg38), as in **b**, with the following axis ranges: accessibility, 0–5,900; Mediator, 0–1,430; cohesin, 0–777; CTCF, 0–318; MCC, 0–50. **f**, Interaction frequencies of promoters of iMac-specific genes with enhancers, promoters and CBSs at 0 h and at 96 h in control and CTCF-depleted cells, as in Fig. 1f; enhancer, $P = 4.1 \times 10^{-7}$; promoter, $P = 8.9 \times 10^{-7}$; CBS, $P = 7.2 \times 10^{-16}$.

absence of CTCF (Extended Data Fig. 4b), we find that CTCF depletion has modest effects on gene expression during lymphoid-to-myeloid transdifferentiation (Fig. 6a). Upon CTCF depletion, 718 genes are significantly downregulated and 558 genes are significantly upregulated,

with a $\log_2$ (fold change) < 2 for 90% of the significantly differentially expressed genes (Extended Data Fig. 6a and Supplementary Table 2). Differentially expressed genes are enriched for B cell- and iMac-specific genes and are more likely to have a CBS near their promoter and to be

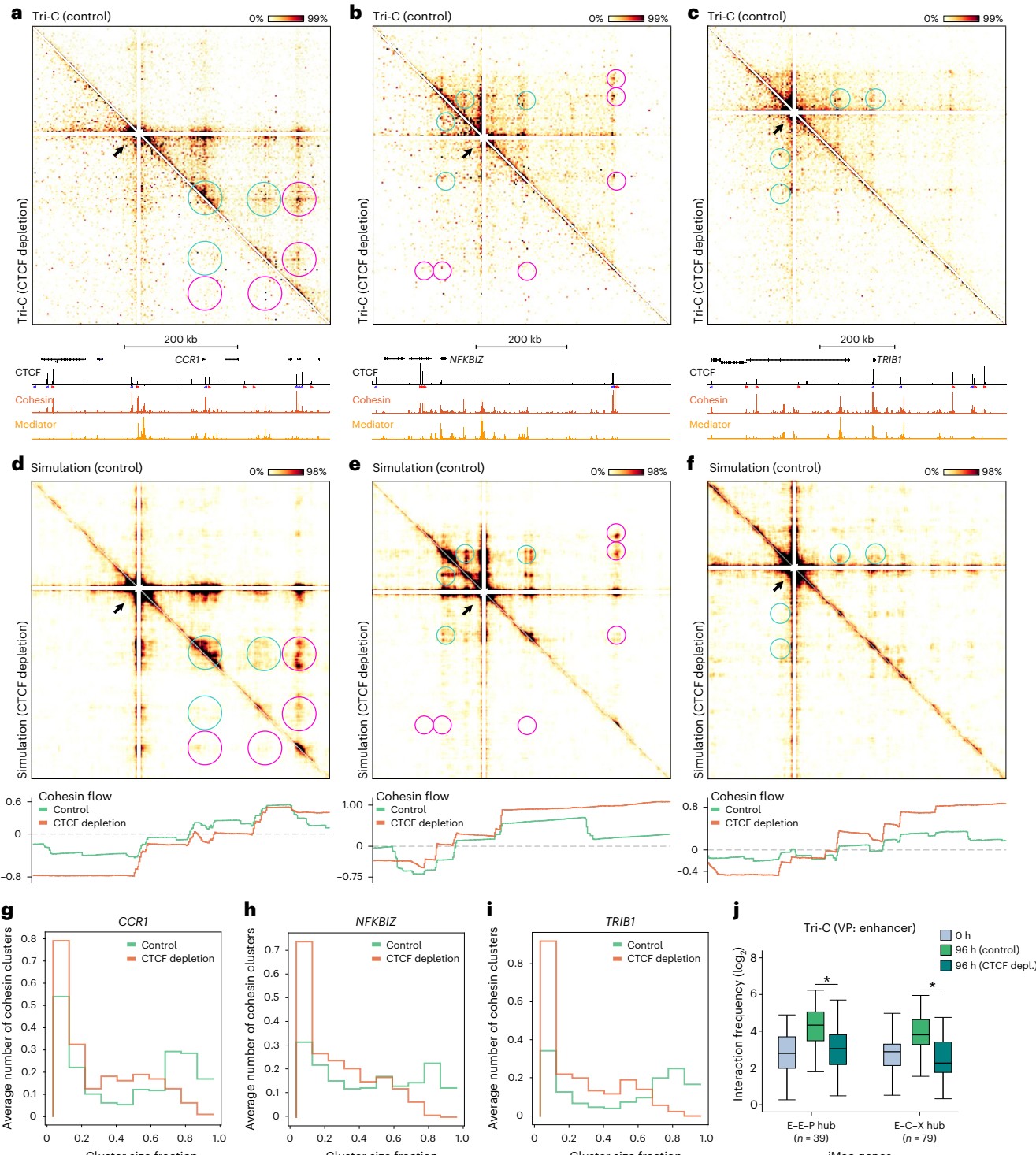

**Fig. 5 | CTCF supports the formation of chromatin hubs. a**, Tri-C contact matrices of the *CCR1* locus (chr3:45,902,299–46,427,299; 525 kb; 2.5-kb resolution; hg38) from the viewpoint of an enhancer in control (top right matrix) and CTCF-depleted (bottom left matrix) cells at 96 h after differentiation induction, as in Fig. 3d. **b**, Tri-C contact matrices of the *NFKBIZ* locus (chr3:101,699,405–102,350,405; 651 kb; 3.5-kb resolution; hg38), as in **a**, with the following axis ranges: CTCF, 0–5,547; cohesin, 0–1,773; Mediator, 0–2,252. **c**, Tri-C contact matrices of the *TRIB1* locus (chr8:124,988,732–125,788,732; 800 kb; 4-kb resolution; hg38), as in **a**, with the following axis ranges: CTCF, 0–12,178; cohesin, 0–2,129; Mediator, 0–5,726. **d**, Tri-C contact matrices of the *CCR1* locus generated with molecular dynamics simulations, as in **a**. The profile below shows cohesin flow (net number of cohesin molecules moving through the locus per minute) in control and CTCF-depleted conditions. Positive values

indicate cohesin movement in the sense direction; negative values indicate the antisense direction; extreme values imply less constrained cohesin movement. **e**, Tri-C contact matrices of the *NFKBIZ* locus generated with molecular dynamics simulations, as in **b**,**d**. **f**, Tri-C contact matrices of the *TRIB1* locus generated with molecular dynamics simulations, as in **c**,**d**. **g**, Cohesin cluster size frequencies, measured as the fraction of the total number of molecules from the simulations of the *CCR1* locus. **h**, Cohesin cluster size frequencies in the *NFKBIZ* locus, as in **g**. **i**, Cohesin cluster size frequencies in the *TRIB1* locus, as in **g**. **j**, Frequencies of three-way interactions involving two enhancers and a promoter (E–E–P hub) and three-way interactions involving an enhancer, a CBS and any other *cis*-regulatory element (E–C–X hub) in iMac-specific loci at 0 h and at 96 h in control and CTCF-depleted (depl.) cells, as in Fig. 1f; E–E–P hubs, $P = 7.1 \times 10^{-8}$; E–C–X hubs, $P = 9.3 \times 10^{-14}$.

regulated by distal enhancers compared to unaffected genes (Extended Data Fig. 6b–e). However, interestingly, we do not generally observe a significant decrease in gene expression in the upregulated gene loci in which we observe a strong impairment in chromatin hub formation, including *CCR1*, *NFKBIZ* and *TRIB1* (Extended Data Fig. 6a). This suggests that CTCF-dependent chromatin hubs do not have a critical role in the regulation of gene expression during cellular differentiation.

### Pairwise interactions correlate with gene expression levels

It has previously been shown that CBSs contribute to the specificity of enhancer–promoter communication by preventing interactions between enhancers and promoters across TAD borders[58]. In agreement with this model, we observe that CTCF depletion leads to the formation of ectopic interactions in two of the targeted upregulated loci. In the *LMO2* locus, we observe increased interactions with the promoter of *CAPRIN1* (Fig. 6b); in the *KDM7A* locus, we observe increased interactions with *cis*-regulatory elements of the *SLC37A3* gene (Fig. 6c). These rewired interactions are associated with a significant increase in *CAPRIN1* and *SLC37A3* expression (Fig. 6d,e). In addition to the ectopic interactions in the *LMO2* and *KDM7A* loci, we find subtle changes in the interaction profiles in some of the other targeted loci, which are also associated with small changes in gene expression. For example, in the *IRF8* and *MAFB* loci, both enhancer–promoter interactions and gene expression levels are slightly decreased and increased, respectively, after CTCF depletion (Extended Data Fig. 6a,f,g). Furthermore, consistent with the observation that downregulated genes are more likely to have distal enhancers (Extended Data Fig. 6d), we find that decreased enhancer–promoter interactions upon CTCF depletion are more likely to be distal (>150 kb) from the promoter compared to increased enhancer–promoter interactions (Extended Data Fig. 6e).

It is of interest that both gene expression patterns and pairwise enhancer–promoter interactions are relatively stable after CTCF depletion and that the minor changes in gene expression following CTCF depletion can often be explained by changes in pairwise enhancer–promoter interaction frequencies. This suggests that basic contacts between enhancers and promoters (in the form of pairwise interactions) are more relevant for gene regulation than their cooperative interactions within chromatin hubs. To further explore the relationship between pairwise enhancer–promoter interactions and gene expression, we computed a total enhancer–promoter interaction score for the targeted gene promoters based on the MCC data through lymphoid-to-myeloid transdifferentiation. Interestingly, we find that changes in enhancer–promoter interactions correlate very well with changes in gene expression over the differentiation course, with a coefficient of 0.82 (Fig. 6f). Integration of the accessibility of the gene promoters further increases this correlation to a coefficient of 0.87 (Fig. 6g). Together, these results indicate that the gradual formation and dissolution of enhancer–promoter interactions are predictive of dynamic changes in gene expression during cellular differentiation but that these gene expression changes do not strongly depend on the higher-order configuration of enhancers and promoters in CTCF-dependent hub structures (Fig. 6h).

## Discussion

In this study, we characterize structural and functional features of the genome over the course of lymphoid-to-myeloid transdifferentiation. We find that interactions between promoters, enhancers and CBSs gradually form and dissolve in upregulated and downregulated gene loci during differentiation. For enhancer–promoter interactions, we find that their frequencies correlate well with changes in the levels of Mediator and cohesin binding at these elements. This is consistent with previous studies that have suggested a role for Mediator[59,60] and cohesin-mediated loop extrusion[61–64] in the formation of enhancer–promoter interactions. Analysis of multi-way interactions shows that the selected upregulated and downregulated gene loci are characterized by

cooperative interactions between all classes of *cis*-regulatory elements in the locus, which gradually form and dissolve over the differentiation course. This indicates that chromatin hubs, which have thus far only been described at high resolution in a few selected loci[37,38], are a common structural property of tissue-specific gene loci. Consistent with previous analyses based on multi-way 3C[38] and imaging experiments[65], we observe cooperative interactions between multiple CBSs, which likely reflect stacking of CBS-anchored loops into rosette structures. To better understand the role of CTCF in the formation of tissue-specific chromatin hubs, we investigate the effects of CTCF depletion over the differentiation course. Interestingly, we observe that depletion of CTCF leads to a strong reduction in cooperative interactions between all *cis*-regulatory elements. This shows that CTCF-dependent interactions provide a scaffold that is required for the formation of cooperative enhancer–promoter interactions in chromatin hubs.

In contrast to the impact of CTCF depletion on chromatin hubs, we observe that depletion of CTCF does not strongly affect pairwise enhancer–promoter interactions. CTCF depletion therefore allows us to separate the function of chromatin hubs and pairwise enhancer–promoter interactions in the regulation of gene expression during cellular differentiation. Consistent with previous CTCF perturbation studies[47,61,66–70], we do not observe widespread misregulation of gene expression in the absence of CTCF. In most of the regions in which chromatin hubs are lost, gene expression is not significantly affected. By contrast, gene loci with significant changes in gene expression are often characterized by changes in pairwise enhancer–promoter interactions after CTCF depletion. This indicates that pairwise enhancer–promoter interactions have an important role in gene regulation, which is not dependent on their cooperative interactions in chromatin hubs. In agreement with our observations in the CTCF depletion experiments, we also show that changes in pairwise enhancer–promoter interaction frequencies are strongly correlated with changes in gene expression during differentiation. Of note, the correlation coefficients that we obtain are much higher than those in previous studies, which may be explained by the high resolution of our data. Because our data do not reveal any instances of preformed, permissive interactions, we conclude that pairwise enhancer–promoter interactions have an instructive role in gene regulation during lymphoid-to-myeloid transdifferentiation.

Consistent with the concept of distinct classes of functional CBSs[53], we observe that ppCBSs interact more frequently with enhancers compared to bCBSs. In addition, we observe that genes with a ppCBS are more likely to engage in long-range interactions and to be downregulated upon CTCF depletion compared to genes without a ppCBS. These and previous observations[52] suggest that ppCBSs may contribute to the formation of (distal) enhancer–promoter interactions. However, the general importance of ppCBSs for gene regulation remains unclear because we do not observe significant changes in gene expression in many of the targeted loci that contain a ppCBS and are characterized by long-range enhancer–promoter interactions. It is possible that there are subtle changes in the expression of these genes that are difficult to detect due to previously observed increased variability of gene expression in the context of cohesin and CTCF perturbations[65]. In addition, it is of interest that we observe a tendency for proximal enhancers to interact more frequently with their cognate promoters upon CTCF depletion. Although speculative at this stage, this could provide a mechanism to compensate for loss of long-range CTCF-dependent interactions and thereby to buffer gene expression changes after CTCF depletion.

The observation that cooperative interactions in chromatin hubs are strongly reduced in the absence of CTCF without a consistent effect on gene expression indicates that these hubs may not have an essential function in gene regulation. Our data therefore suggest that enhancer cooperativity is not strictly dependent on simultaneous action of enhancers at a gene promoter. This is consistent with a recent preprint

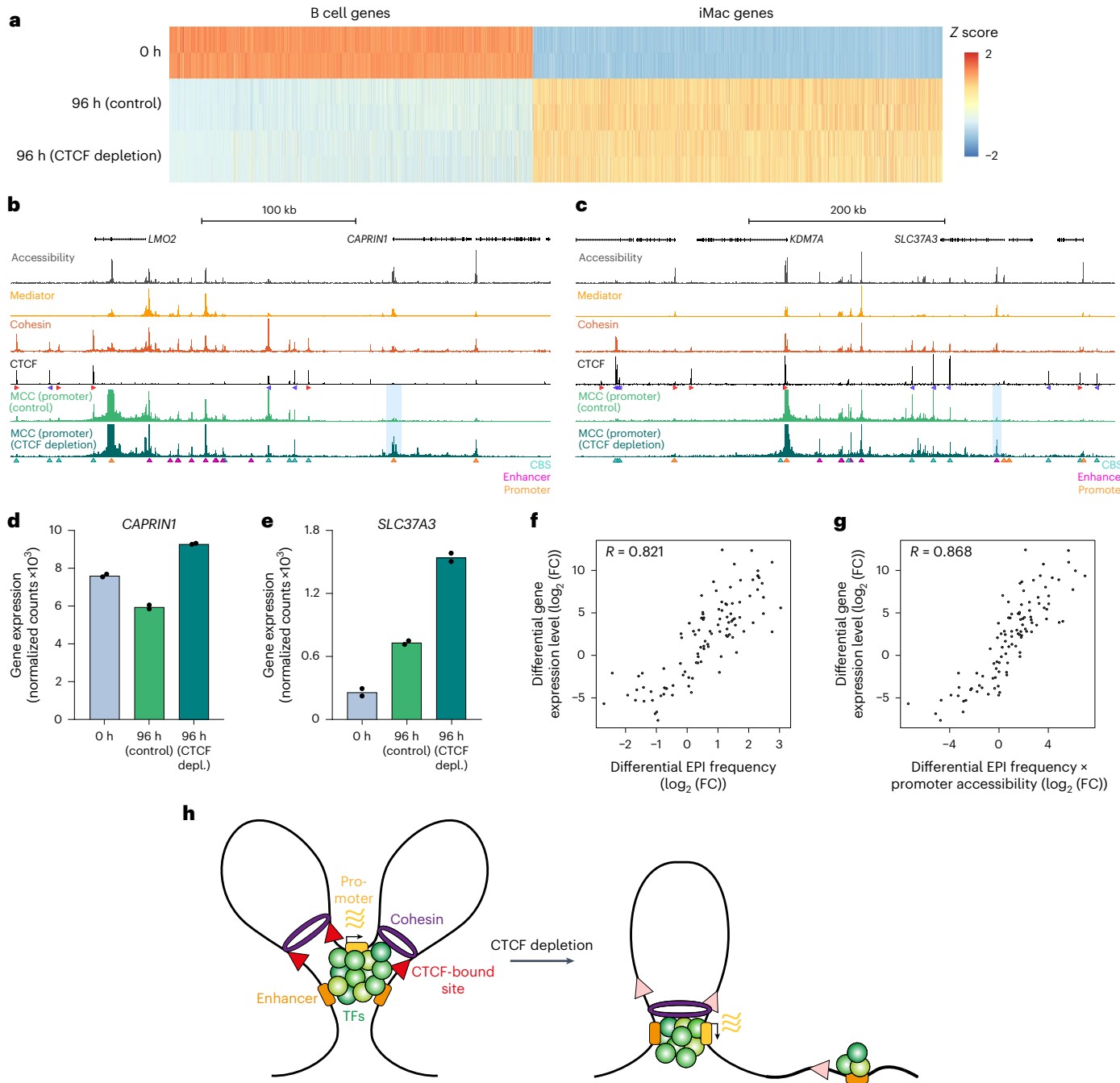

**Fig. 6 | Pairwise enhancer–promoter interactions correlate with gene expression levels. a**, Heatmap showing the $Z$ score of normalized RNA counts of B cell-specific genes ($n = 2,968$) and iMac-specific genes ($n = 3,341$) at 0 h and at 96 h in control-treated and CTCF-depleted cells. RNA-seq experiments were performed in two biological replicates. **b**, Chromatin interactions in the *LMO2* locus (chr11:33,804,269–34,154,269; 350 kb; hg38) in control and CTCF-depleted cells at 96 h after differentiation induction. From top to bottom: gene annotation, ATAC–seq, MED26 ChIPmentation, SMC1A ChIPmentation, CTCF ChIPmentation, MCC data from the viewpoint of the promoter. Axes are scaled to signal with the following ranges: accessibility, 0–4,018; Mediator, 0–7,353; cohesin, 0–1,875; CTCF, 0–12,417; MCC, 0–40. Annotation as in Fig. 1d, with ectopic interactions in CTCF-depleted cells highlighted in light blue. **c**, Chromatin interactions in the *KDM7A* locus (chr7:139,961,429–140,511,429; 550 kb; hg38), as in **b**, with the following axis ranges: accessibility, 0–3,566; Mediator, 0–7,621; cohesin,

0–2,469; CTCF, 0–7,069; MCC, 0–40. **d**, *CAPRIN1* expression levels at 0 h and at 96 h in control-treated and CTCF-depleted cells, derived from RNA-seq data. Bars represent the average of two biological replicates; corresponding data points are shown as dots. **e**, *SLC37A3* expression levels, as in **d**. **f**, Correlation between differential expression levels and total enhancer–promoter interaction (EPI) frequencies (24 h versus 0 h and 96 h versus 0 h) based on Spearman's correlation test. **g**, Correlation between differential expression levels and the product of total EPI frequencies and promoter accessibility levels, as in **f**. **h**, Graphical summary: extruding cohesin molecules are stalled at CBSs, promoters and enhancers; in the presence of CTCF (left), this leads to detectable clustering of these elements in chromatin hubs; in the absence of CTCF (right), these clusters form less frequently; however, enhancers still interact with their cognate promoters in a pairwise manner (example only shown for one of the two enhancers) and thereby maintain gene expression levels. TFs, transcription factors.

based on imaging experiments that shows that hubs are rare[71]. Although not directly tested in our study, this may have implications for the theory that transcriptional activation is dependent on the recruitment of a critical concentration of transcription factors and coactivators by clustered *cis*-regulatory elements in nuclear condensates[13–16,72]. However, it is important to note that our conclusions mostly pertain to cooperativity between multiple clusters of enhancers in a locus, because the resolution of Tri-C is not sufficient to distinguish closely spaced enhancer elements within individual enhancer clusters. It is therefore possible that higher-resolution data and/or methods that are not dependent on proximity ligation uncover cooperative interactions that are CTCF independent and potentially more relevant for gene regulation. Furthermore, it is possible that CTCF-dependent hubs have a more pronounced function in specific cellular contexts or in regulating dynamic aspects of gene expression that are not reflected in RNA-sequencing (RNA-seq) data. In this regard, it is of interest that it has previously been shown that CTCF-depleted iMacs have impairments in their acute inflammatory response[47].

Without a clear function in gene regulation, it is not obvious why CTCF-dependent chromatin hubs exist. Computational modeling of our data indicates that cooperative interactions between CBSs form as a result of cohesin-mediated loop extrusion. A relatively high density of extruding cohesin molecules and relatively stable stalling of cohesin at CBSs are in principle sufficient to explain the clustering of CBSs at the bases of multiple extruded loops, consistent with previously described rosette structures and CTCF clusters[38,65,73]. Cohesin also stalls at enhancers and promoters, which may depend on interactions with Mediator[50,59,60] and RNA polymerase II (RNAPII)[54,74,75]. This may therefore lead to clustering of enhancers and promoters with the CBS anchors. However, because cohesin stalling at enhancers and promoters is less stable than at CBSs, cohesin-mediated loop extrusion may not lead to strong clustering of *cis*-regulatory elements in the absence of the relatively stable CTCF-mediated interactions. Our model therefore suggests that chromatin hubs form as a result of the strong boundary function of CBSs during loop extrusion. Despite the minor general role of CTCF in gene regulation, it has been shown that CTCF boundaries are critical for context-specific regulation of important gene loci and that their perturbation can have severe consequences for development and disease[76,77]. It is therefore plausible that clustering of *cis*-regulatory elements in chromatin hubs arises as a by-product of strong loop extrusion boundaries, which primarily function to regulate the specificity of enhancer–promoter communication.

The development of innovative technologies to map genome architecture has led to an increasingly better understanding of the dynamic 3D structures into which the genome is organized. While some of these structures may be critical for nuclear functions, such as transcription, replication and DNA repair, other structural features may more simply reflect the processes required to compact and organize the genome in the cell nucleus[78]. A remaining challenge therefore lies in distinguishing these two possibilities and relating genome structure to function. By combining a dynamic differentiation system, protein perturbation and high-resolution pairwise and multi-way analyses of chromatin interactions, our study decouples the function of pairwise enhancer–promoter interactions from their higher-order clustering in chromatin hubs in the regulation of gene expression. Further developments in technologies to map and perturb structural and functional features of the genome will allow for more detailed dissections of their cause–consequence relationships across biological contexts in the future.

## Online content

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

## Methods

### Cell culture

BLaER1 (ref. 45) and BLaER1-CTCF-mAID[47] cells were cultured in RPMI 1640 medium (Gibco, 31870025), supplemented with 10% FBS (Gibco, 10270106), 25 mM HEPES (Fisher Scientific, 15-630-080) and 2% GlutaMAX Supplement (Thermo Scientific, 35050038) at 37 °C with 5% $CO_2$. The cells were kept in the range of 0.2–1.5 million cells per ml and passed to fresh medium every 2 d. To transdifferentiate the B cells into functional macrophages, fresh medium containing transdifferentiation factors (100 nM 17β-estradiol (Calbiochem, 3301), 10 ng ml$^{-1}$ human IL-3 (PeproTech, 200-03) and 10 ng ml$^{-1}$ human CSF-1 (PeproTech, 300-25)) was added. The cells were differentiated up to 168 h and collected at the specified time points. As a nondifferentiated control (B cell stage, the 0-h time point), the cells were cultured in medium without transdifferentiation factors. CTCF was depleted in BLaER1-CTCF-mAID cells simultaneously with inducing transdifferentiation. To achieve this, medium containing transdifferentiation factors and 500 μM IAA (Sigma-Aldrich, I5148), which was prepared freshly before use, was added to the cells. The cells were collected at the 96-h time point. To ensure that IAA was not degraded, after 48 h of culture, the medium was exchanged with fresh medium supplemented with transdifferentiation factors and IAA. BLaER1 and BLaER1-CTCF-mAID cell lines were obtained from T. Graf (Centre for Genomic Regulation, Barcelona, Spain) and G. Stik (Josep Carreras Leukaemia Research Institute, Barcelona, Spain).

### Micro-Capture-C

MCC experiments were performed in three biological replicates per condition using a procedure based on the published protocol[79] that was further optimized for BLaER1 cells. Briefly, multiple aliquots of 15 million cells were cross-linked in 10 ml of culture medium with 2% formaldehyde (Thermo Fisher, 28908). Next, the cells were permeabilized with digitonin (0.0025%, Sigma-Aldrich, D141). Digestion was performed with MNase (NEB, M0247) in low-$Ca^{2+}$ MNase buffer (10 mM Tris-HCl, pH 7.5, 10 mM $CaCl_2$) for 1 h at 37 °C. After confirming efficient digestion, samples were ligated and decross-linked. DNA was extracted using the DNeasy Blood and Tissue Kit (Qiagen, 69504). Before sonication, samples were size selected to remove fragments corresponding to mononucleosomes using Mag-Bind TotalPure NGS beads (Omega Bio-tek, M1378-01). MCC libraries were sheared using a Covaris S220 Focused-ultrasonicator to a mean size of 200 bp (time, 280 s; duty factor, 10%; peak incident power, 175 W; cycles per burst, 200). Afterward, the sonicated DNA fragments were indexed using 2 μg DNA per indexing reaction. To increase library complexity, each reaction was indexed in duplicate in seven PCR cycles using Herculase II polymerase (Agilent, 600677). The duplicate reactions were pooled, and the libraries were enriched for viewpoints of interest in a double-capture procedure using the KAPA HyperCapture Reagent Kit (Roche, 9075828001). A total of 8 μg of indexed sample per biological replicate was used as input for the first capture. Oligonucleotides used for enrichment were 120 nucleotides long and were designed with the Python-based oligo tool[26] (https://oligo.readthedocs.io/en/latest/). The oligonucleotides were ordered as a multiplexed panel of ssDNA 5′-biotinylated oligonucleotides (IDT, xGen Custom Hybridization Capture Panels). Each oligonucleotide of the panel was used at a final concentration of 2.9 nM. Captured DNA was pulled down using M-270 Streptavidin Dynabeads (Invitrogen, 65305) and amplified in 12 PCR cycles. All captured DNA was used as input for the second hybridization reaction. After library quality assessment with a fragment analyzer, the libraries were sequenced on the Illumina sequencing platform with 300 cycles of paired-end reads.

### Capture-C and Tri-C

Capture-C and Tri-C experiments were performed with two and three biological replicates per condition, respectively, following published protocols[80,81]. Briefly, multiple aliquots of 15 million cells were cross-linked in 10 ml of culture medium with 2% formaldehyde (Thermo Fisher, 28908). The aliquots were then divided into three digest reactions of 5 million cells and digested with the NlaIII restriction enzyme (NEB, R0125L). After subsequent proximity ligation and DNA extraction, the three reactions were combined and sheared using a Covaris S220 Focused-ultrasonicator. The 3C library (6–8 μg) in 130 μl TE buffer was sheared to a mean size of 200 bp for Capture-C experiments (time, 180 s; duty factor, 10%; peak incident power, 175 W; cycles per burst, 200) and 450 bp for Tri-C experiments (time, 55 s; duty factor, 10%; peak incident power, 140 W; cycles per burst, 200). After sonication, Tri-C samples were size selected with 0.7× volume of Mag-Bind TotalPure NGS beads (Omega Bio-tek, M1378-01) for fragments longer than 300 bp. To increase library complexity, indexing was performed for each sample in duplicate (2 × 2 μg of sheared sample). Capture-C samples were indexed in six PCR cycles, while Tri-C samples were indexed in seven PCR cycles. To enrich the libraries for ligated fragments containing viewpoints of interest, a double-capture enrichment was performed, as described for MCC. The oligonucleotides used in Capture-C experiments were 70 nucleotides long, while oligonucleotides used in Tri-C experiments were 120 nucleotides in length. In the first enrichment round, an input of 4 μg per biological replicate was used in Capture-C experiments. To increase the complexity of enriched fragments in Tri-C experiments, an input of 9 μg per biological replicate was used. Captured DNA was pulled down using M-270 Streptavidin Dynabeads (Invitrogen, 65305) and amplified in 11 PCR cycles. All captured material was used as input for the second enrichment reaction, which was performed in the same manner. The quality of the final libraries was assessed with a fragment analyzer, and the libraries were sequenced on the Illumina sequencing platform. Capture-C samples were sequenced with 150 cycles of paired-end reads, while Tri-C samples were sequenced with 300 cycles of paired-end reads.

### ChIPmentation

ChIPmentation experiments for CTCF were performed with two biological replicates per condition; ChIPmentation experiments for MED26 and SMC1A were performed with three biological replicates per condition during lymphoid-to-myeloid transdifferentiation and with two biological replicates per condition in CTCF-depleted and control cells. We followed a published protocol[48], which we heavily optimized to improve the signal-to-noise ratio of the data. Briefly, aliquots of 1.5 million cells were cross-linked in a single fixation reaction with 1% formaldehyde (28908, Thermo Scientific) for 10 min at room temperature, quenched with ice-cold glycine (Sigma-Aldrich, G7126) at a final concentration of 125 mM and washed two times with ice-cold PBS. Fixed cells were first gently lysed with Farnham lysis buffer (5 mM PIPES, pH 8, 85 mM KCl, 0.5% NP-40) to isolate nuclei, which was followed by nuclear lysis with 0.5% SDS buffer (10 mM Tris-HCl, pH 8, 1 mM EDTA, 0.5% SDS). Next, chromatin was fragmented using an optimized procedure for each target. In the CTCF ChIPmentation experiments, chromatin was fragmented by sonication (time, 7 min; duty factor, 5%; peak incident power, 140 W; cycles per burst, 200). In all MED26 and SMC1A ChIPmentation experiments, chromatin was first fragmented by a titrated amount of MNase (2.5 Kunitz) and then gently sonicated (time, 1 min; duty factor, 2%; peak incident power, 105 W; cycles per burst, 200). To perform immunoprecipitation, a mixture of Protein A (10008D, Invitrogen) and Protein G (10003D, Invitrogen) Dynabeads in a 1:1 ratio was blocked with 0.5% BSA, washed and pre-incubated with 2 μg primary antibody (anti-CTCF, Diagenode, C15410210-50; anti-SMC1A, Abcam, ab9262; anti-MED26, Bethyl Laboratories, A302-370) and 1 μg spike-in antibody (Biozol, 61686) for 6 h at 4 °C. Sheared chromatin was diluted in IP buffer (10 mM Tris-HCl, pH 8, 1 mM EDTA, 150 mM NaCl, 1% Triton X-100, 1× Protease Inhibitor Cocktail (Sigma-Aldrich, 11873580001)) and added along with 50 ng *Drosophila* spike-in chromatin to the bead-bound antibodies. The samples were incubated overnight on a rotator at 4 °C and washed with stringent buffers the next morning. Sequencing adaptors were added to the bead-bound

DNA by tagmentation with the Illumina Tagment DNA Enzyme and Buffer Kit (Illumina, 20034197). After washes, the bead-bound samples were decross-linked overnight at 65 °C in the presence of proteinase K (0.2 mg ml⁻¹, final) and cleaned with 1.8× volume of Mag-Bind TotalPure NGS beads (Omega Bio-tek, M1378-01). The tagmented DNA was used for library preparation with the NEBNext High-Fidelity 2× PCR Master Mix (NEB, M0541). The quality of the libraries was assessed with a fragment analyzer, and the libraries were sequenced on the Illumina sequencing platform with 75 cycles of paired-end reads.

## RNA sequencing

RNA-seq was performed with two biological replicates per condition. Briefly, cells were collected in QIAzol Lysis Reagent (Qiagen, 79306). RNA was isolated using phenol–chloroform extraction, and the quality was assessed on a NanoDrop and a fragment analyzer. RNA (150 µg) for each sample was diluted in 130 µl RNase-free water and sheared using a Covaris S220 Focused-ultrasonicator (time, 10 s; duty factor, 1%; peak incident power, 100 W; cycles per burst, 200). Sonicated RNA (2 µl per sample) was purified with the miRNeasy Micro Kit (Qiagen, 217084) and treated with DNase I (Qiagen, 79254). Purified RNA (500 ng) was used to prepare sequencing libraries with the Illumina Stranded Total RNA Prep, Ligation with Ribo-Zero Plus kit (Illumina, 20040525). The quality of the libraries was assessed with a fragment analyzer, and the libraries were sequenced on the Illumina sequencing platform with 100 cycles of paired-end reads.

## Quantitative PCR with reverse transcription

Quantitative PCR with reverse transcription experiments were performed as previously described[46]. Briefly, cells were collected in 1 ml of QIAzol Lysis Reagent (Qiagen, 79306). Nucleic acids were isolated using phenol–chloroform extraction, and DNA was depleted using the TURBO DNA-free Kit (Thermo Fisher, AM1907) according to the manufacturer's instructions. After confirming the quality of the RNA with an RNA Screen-Tape (Agilent, 5067-5576), reverse transcription was performed with Maxima Reverse Transcriptase (Thermo Scientific, EP0741) with random hexamer primers. qPCR was performed with the SYBR Select Master Mix (Applied Biosystems, 4472908) using previously validated primer sequences[45] (Supplementary Table 3) in a qTower 2.0/2.2 (Analytik Jena) thermocycler. The reaction parameters were as follows: 50 °C for 2 min, 95 °C for 2 min, (95 °C for 15 s, 60 °C for 60 s) repeated for 40 cycles. The data were normalized to the housekeeping gene *GAPDH*.

## Flow cytometry

To monitor changes in cell surface markers during lymphoid-to-myeloid transdifferentiation, flow cytometry experiments were performed at different differentiation stages. Cells were collected, washed with PBS and blocked with human Fc Receptor Binding Antibody (1:5, eBioscience, 16-9161-73) for 20 min at 4 °C. Next, cells were stained with anti-CD19–APC-Cy7 (1:20, BD Pharmingen, 348794) and anti-CD11b–APC (1:5, BD Pharmingen, 550019) antibodies for 30 min at 4 °C. After washing with PBS, the cells were analyzed on a Sony SH800 cytometer. The data were analyzed with FlowJo (version 10.8.1).

## Analysis of Micro-Capture-C data

MCC data were processed using the MCC pipeline (hg38 genome assembly)[36]. After sequence alignment with Bowtie 2 (ref. 82), BAM files from all replicates were merged. Because capture oligonucleotide-mediated enrichment is performed in a single hybridization reaction containing all multiplexed samples, the capture efficiency of each viewpoint is similar across samples; interaction profiles from the same viewpoint are therefore comparable across samples. To make the interaction profiles of the different viewpoints comparable, the interaction profiles were normalized for an equal number of interactions on the chromosome containing the viewpoint to correct for potential variation in the enrichment efficiency of the different capture oligonucleotides.

Statistical comparisons between samples were made per viewpoint and therefore not influenced by potential variability between viewpoints. Peaks were called with MACS2 (ref. 83) (version 2.1.2), using a *q* value of 0.05 and without building a shifting model. Peaks from three time points were combined and collapsed into a single unified peak set. Before peak annotation, the called peaks were filtered to exclude peaks outside a ±2-Mb region surrounding the viewpoint and within ±1 kb of the viewpoint. MCC peaks were annotated as CBSs, promoters or enhancers based on their overlap with relevant genomic features. CBSs were defined by overlap with CTCF ChIP–seq peaks. Promoters were defined by overlap with ATAC–seq peaks located within ±500 bp of the transcription start site of an expressed gene. In the remaining peaks, enhancers were defined by their overlap with H3K27ac ChIP–seq peaks. Transient enhancers were defined as the enhancers of B cell-specific genes with increased Mediator occupancy over the differentiation course. To quantify MCC interaction frequencies, MCC bigWig tracks were imported into R (version 4.2.0), and the coverage values per base pair over MCC peaks were extracted using the R package rtracklayer (version 1.56.1). The interaction frequency of every MCC peak was measured by calculating the average of the highest 70% of base pair bigWig coverage values. For quantification of the proportion of inter-actions of promoter, ppCBS and bCBS viewpoints with *cis*-regulatory elements across the targeted loci (Fig. 2f), CBS viewpoints were classified as ppCBS (and not bCBS) in case they were proximal to both a promoter and a boundary. Similar to quantification of the MCC signal, the occupancy of SMC1A, MED26 and CTCF at annotated MCC peaks was quantified by extracting the coverage information from the corresponding bigWig files and calculating the average of the highest 70% of base pair coverage values across the MCC peaks.

## Analysis of Capture-C and Tri-C data

Capture-C and Tri-C data were analyzed using the CapCruncher pipeline[80] (version 0.2.3) in capture mode (hg38 genome assembly). Data from all replicates were merged. Capture-C data were normalized as described for MCC. For Tri-C analysis, a custom script was used to extract reads with two or more *cis* reporters. These reads were used to calculate multi-way interaction counts between reporter fragments for each viewpoint. Contact matrices in which multiple three-way interactions were calculated from reads containing more than three reporters did not show any qualitative differences compared to contact matrices in which only a single three-way interaction per read was included. All detected three-way interactions were therefore included in the contact matrices to boost the complexity of the data. The interaction counts were binned in 1,500–4,000-bp bins and corrected for the number of restriction fragments present in each bin. The matrices were further normalized for the total multi-way interaction counts within a 2-Mb region (which was corrected for the number of restriction fragments present in the region) surrounding the viewpoint. To quantify the interaction frequencies between two regions of interest, a 3 × 3 binning matrix surrounding the pair of regions of interest was extracted from the normalized contact matrices and averaged.

## Analysis of ChIPmentation data

ChIPmentation data were analyzed with the NGseqBasic pipeline[12] (hg38 genome assembly). After sequence alignment and dedupli-cation, bigWig coverage tracks were generated with the deepTools (version 3.3.0) bamCoverage function using a binning size of 10 bp. CTCF ChIPmentation coverage tracks were normalized using spike-in read count, while SMC1A and MED26 coverage tracks were normalized using RPKM. Coverage tracks of replicates were merged using UCSC bigWigMerge (version 2).

## Analysis of RNA-seq data

RNA-seq data were processed using the nf-core/rnaseq pipeline (version 3.12.0, hg38 genome assembly). After sequence alignment with

STAR[84] (version 2.6.1d), the expression of protein-coding exons was quantified with featureCounts (version 2.0.6). DESeq2 (ref. [85]) (version 1.36.0) was used to perform differential gene expression analysis, with a threshold of adjusted $P$ value < 0.01 and absolute $\log_2$ (fold change) > 0.6. To compare the genomic landscapes and enhancer distributions of differentially expressed genes with those of stable genes upon CTCF depletion, promoters and enhancers were paired based on the previously described TAD pairing strategy[46], in which promoters and enhancers within the same TAD are paired if the Spearman's correlation coefficient between mRNA and eRNA read counts during transdifferentiation is higher than 0.4. The distances between all paired promoters and enhancers were plotted in the distance distributions (Extended Data Fig. 6d).

## Correlation analyses

Spearman rank correlation was used to measure the association between changes in pairwise chromatin interactions and changes in various features of interest, including SMC1A, MED26 and CTCF occupancy, H3K27ac levels and eRNA expression. Changes in interaction frequencies and different signals at corresponding MCC peaks were calculated at 24 h and 96 h relative to the baseline at 0 h. To correlate pairwise enhancer–promoter interaction frequencies and gene expression levels, a total enhancer–promoter interaction score was defined for every targeted promoter as the sum of the signal of the enhancer peaks in the MCC interaction profiles. For all targeted loci, changes in gene expression levels, total enhancer–promoter interaction scores and the products of the total enhancer–promoter interaction scores and promoter accessibility, based on TT-seq, MCC and ATAC–seq data, were calculated. Changes at 24 h and 96 h were calculated relative to the baseline at 0 h. Spearman rank correlation coefficients were computed to investigate the relationship between changes in pairwise enhancer–promoter interactions and changes in gene expression levels during lymphoid-to-myeloid differentiation as well as the relationship between changes in MED26 and SMC1A occupancy and pairwise enhancer–promoter interactions after CTCF depletion.

## Molecular dynamics simulations

Two-megabase regions around the *CCR1*, *NFKBIZ* and *TRIB1* loci were modeled based on a previously described modeling framework[54,55] using the multipurpose EspressoMD package[86]. Briefly, the chromatin fiber is modeled as a self-avoiding polymer chain consisting of equisized beads representing 2 kb of chromatin. Beads were classified as binding the transcription machinery (that is, promoters and enhancers, based on Mediator ChIPmentation peaks), binding CTCF (based on CTCF ChIPmentation peaks, including motif orientation), as transcribed genic regions (gene bodies of active genes, based on TT-seq data) or as neutral (with none of the above-mentioned features). Polymers were used in molecular dynamics simulations in a 3D space following Langevin equations to model the thermal motion of chromatin and its binding factors in an implicit solvent (the nucleoplasm) with the following postulations[54]: (1) the transcription machinery has affinity for promoters and enhancers and traverses gene bodies; (2) components of the transcription machinery (RNAPII, Mediator) also have affinity for each other to simulate condensate formation; (3) cohesin complexes can bind anywhere on the polymer and extrude loops but have a preference for loading at enhancers and promoters. To account for the experimentally observed accumulation of cohesin at CTCF- and Mediator-bound sites, extrusion of loops by cohesin stalls when encountering a CTCF-bound site with a motif oriented toward the direction of extrusion or when meeting the transcription machinery. By dynamically forming and dissolving protein–chromatin bonds, this framework simulates chromatin loop extrusion by the cohesin complex and traversing of RNAPII during transcription. The rates of loop extrusion and RNAPII translocation along chromatin were set within the range of experimentally deduced values (RNAPII, 1–5 kb min⁻¹; cohesin,

15–30 kb min⁻¹)[55]. RNAPII–cohesin and cohesin–cohesin crossing rates were set at 1.5 and 0.15 crossings per second, respectively.

The ensemble of chromatin conformations resulting from these simulations was used to generate pairwise interaction profiles and three-way contact matrices among the beads of each polymer, allowing for direct quantitative comparisons between the model and the experimental MCC and Tri-C data, respectively. The degree of triplet colocalization of the viewpoint with regions A and B, over what would be expected by the probability of independent, pairwise interactions (Extended Data Fig. 5), was estimated from the predicted structures using the correlation coefficient:

$$\mathrm{corr}(A, B) = \frac{P_{MYC,A,B} - P_{MYC,A}P_{MYC,B}}{\left(P_{MYC,A}(1 - P_{MYC,A})P_{MYC,B}(1 - P_{MYC,B})\right)^{1/2}}.$$

## Analysis of reference datasets

Available ATAC–seq[47] (GSE131620), CTCF ChIP–seq[47] (GSE140528) and H3K27ac ChIP–seq[47] (E-MTAB-9010) data were analyzed with the NGseqBasic pipeline[87] (hg38 genome assembly). ATAC–seq and H3K27ac ChIP–seq peaks were identified using MACS2 (ref. [83]) (version 2.1.2) in broad peak-calling mode with a $q$ value of 0.01 and a broad cutoff of 0.05. Peaks in CTCF ChIP–seq data were identified using MACS2 in narrow peak-calling mode with a $q$ value of 0.01. H3K27me3 ChIP–seq bigWig tracks (GSE259000, GSE256663 and GSE257306) were downloaded from ENCODE. Hi-C data[47] (GSE131620) were analyzed with HiC-Pro[88] (Servant, 2015) and plotted using cooltools[89]. TT-seq data[46] (GSE131620) were processed with the nf-core/rnaseq pipeline as described for RNA-seq data analysis. To quantify mRNA expression levels, featureCounts was used to count reads over the exons of all protein-coding genes. To quantify eRNA expression levels, feature-Counts was used to count reads over the MCC enhancer peak regions. The raw count data were normalized by DESeq2-estimated size factors.

## Statistics and reproducibility

The sample sizes in this study were chosen to ensure sufficient data depth and robust assessment of differences between conditions. Specifically, three biological replicates were used in MCC and Tri-C experiments, each including multiple technical replicates to enhance data complexity. No statistical methods were used to predetermine sample sizes, but our sample sizes are similar to those reported in previous publications[36,37,60,90], and the observed biological effects of interest are clearly detectable between conditions and consistent across replicates. The nonparametric Wilcoxon signed-rank test was used for statistical analysis, which does not assume normality. Therefore, normality was not formally tested. Data distributions are shown as individual data points in the relevant figures. Two viewpoints, *IFNGR2* and *TASL*, were excluded from the quantification of MCC and Tri-C experiments due to poor data quality. All experiments based on sequencing data, unless stated otherwise in the Methods, were conducted with three biologically independent samples, and all attempts were successful. Samples were randomly allocated into different experimental groups before their treatment with auxin or control conditions. The investigators were not blinded to allocation during experiments and outcome assessment, because blinding was not relevant to this study, as all samples were analyzed using the same computational pipelines, with results generated by scripts without researcher interference.

## Reporting summary

Further information on research design is available in the Nature Portfolio Reporting Summary linked to this article.

## Data availability

The MCC, Capture-C, Tri-C, ChIPmentation and RNA-seq datasets generated and analyzed for the current study are available from the Gene Expression Omnibus (GEO) as a SuperSeries under accession number

GSE263641. ATAC–seq[47] data are available from GEO under accession number GSE131620. CTCF ChIP–seq[47] data are available from GEO under accession number GSE140528. H3K27ac ChIP–seq[47] data are available from ENCODE under accession number E-MTAB-9010. Hi-C data[47] are available from GEO under accession number GSE131620. TT-seq data[46] are available from GEO under accession number GSE131620. H3K27me3 ChIP–seq bigWig tracks are available from GEO under accession numbers GSE259000, GSE256663 and GSE257306. Source data are provided with this paper.

## Code availability

The custom code used for MCC and Tri-C data analysis is available on Zenodo at https://doi.org/10.5281/zenodo.14932098 (ref. 91).

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

## Acknowledgements

We thank T. Graf (Centre for Genomic Regulation, Barcelona, Spain) and G. Stik (Josep Carreras Leukaemia Research Institute, Barcelona, Spain) for providing the BLaER1 and BLaER1-CTCF-mAID cell lines. We are grateful to P. Cramer for advice, discussions and infrastructure support. We are grateful for support from the Facility for Light Microscopy at the Max Planck Institute for Multidisciplinary Sciences, particularly from J. Jakobi, P. Lénart and A. Politi. We thank J. Söding for advice on bioinformatic analysis and J. Choi, K. Lysakovskaia, K. Maier, M. Rohm, P. Rus, G. Stik and K. Zumer for experimental advice and support. We are grateful to K. Schmitt for support with data visualization and members of the Oudelaar group for helpful discussions and feedback. This work was supported by the Max Planck Society (A.M.O); the Deutsche Forschungsgemeinschaft via SFB 1565 (project 469281184/P02 to A.M.O. and project 469281184/P03 to A.P.), SPP 2202 (project 507778679 to A.M.O. and project 422389065 to A.P.) and SPP 2191 (project 506296585 to A.P.); the PhD program 'Genome Science', at the International Max Planck Research School at the Georg August University Göttingen (M.A.K., S.R., T.B.N.C. and A.A.); and the MSc–PhD program 'Molecular Biology' at the International Max Planck Research School at the Georg August University Göttingen (Y.Z., Z.F.G. and N.V.).

## Author contributions

M.A.K. carried out most of the experiments, performed basic data analyses and prepared the figures. Y.Z. performed the majority of the bioinformatic analyses. Z.F.G. and S.R. contributed equally. Z.F.G. performed MCC experiments over the differentiation course. S.R. optimized the ChIPmentation protocol and performed ChIPmentation experiments. M.B. performed molecular dynamics simulations. T.B.N.C. performed RNA-seq experiments. N.V. performed ChIPmentation experiments and bioinformatic analyses. A.A. performed ChIPmentation experiments. M.L. assisted with bioinformatic analyses. A.P. provided conceptual advice, supervised the molecular dynamics simulations and acquired funding. A.M.O. conceived and supervised the project, acquired funding and wrote the paper, with input from M.A.K., Y.Z., M.B. and A.P.

## Funding

## Competing interests

The authors declare no competing interests.

## Additional information

**Extended data** is available for this paper at https://doi.org/10.1038/s41594-025-01555-z.

**Correspondence and requests for materials** should be addressed to A. Marieke Oudelaar.

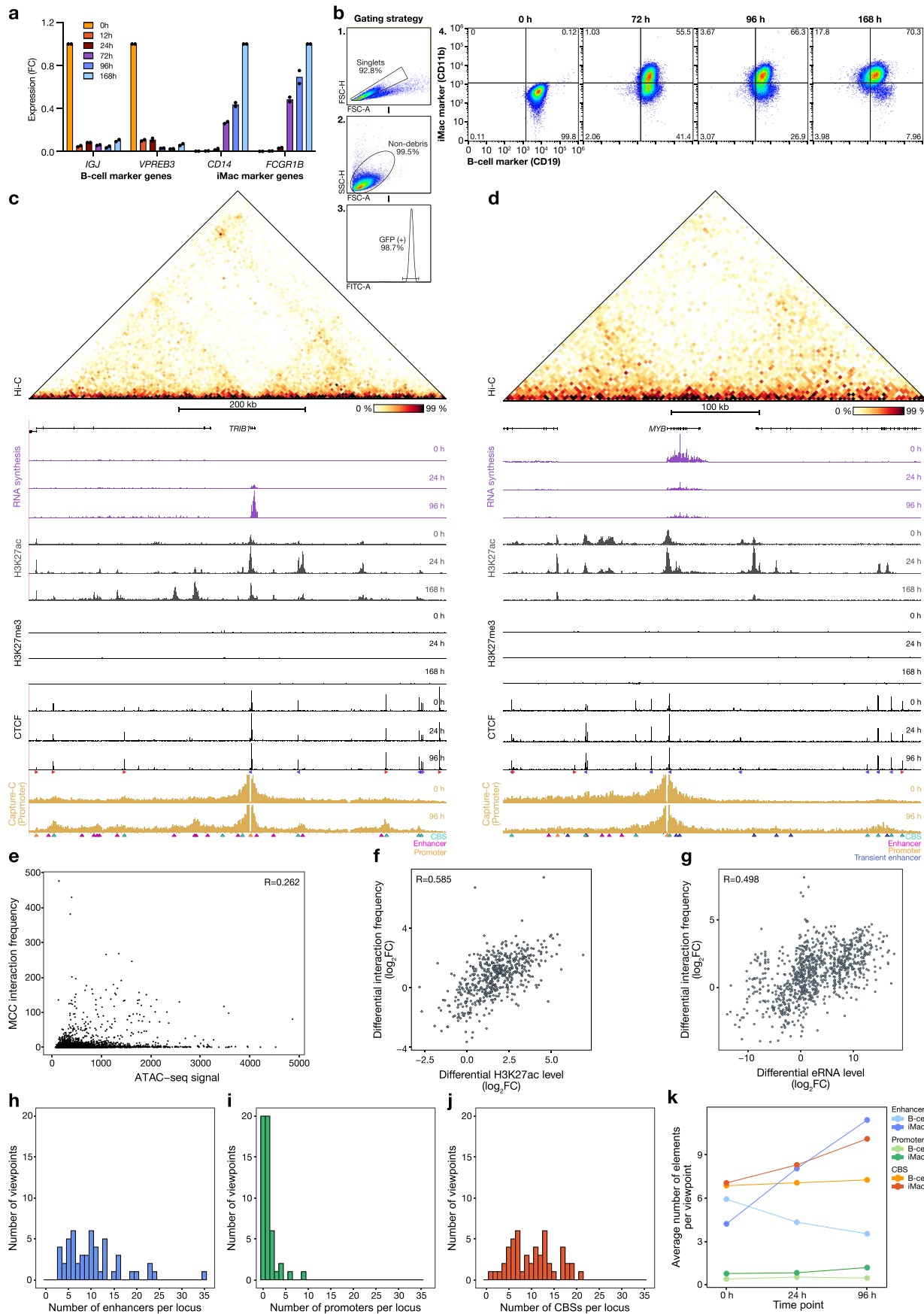

**Extended Data Fig. 1 | See next page for caption.**

**Extended Data Fig. 1 | Characterization of the BLaER1 lymphoid-to-myeloid transdifferentiation system.** (**a**) Expression of marker genes through transdifferentiation, measured with RT-qPCR. Expression of B-cell marker genes (*IGJ*, *VPREB3*) is shown as fold change (FC) relative to the 0 h timepoint; expression of iMac marker genes (*CD14*, *FCGR1B*) is shown relative to 168 h. Bars represent the average of 2 biological replicates; corresponding data points are shown as dots. (**b**) Changes in cell surface markers during lymphoid-to-myeloid transdifferentiation, measured by flow cytometry. A representative example of 2 biological replicates is shown. Gating based on froward scatter height (FSC-H) and forward scatter area (FSC-A) was used to exclude doublets (panel 1); gating based on side scatter height (SSC-H) and FSC-A was used to exclude debris (panel 2); GFP + , alive cells were selected (panel 3); quadrant gates were set on the 0 h sample, which is CD19+ and CD11b- (panel 4). (**c**) Chromatin landscape of the *TRIB1* locus (chr8:125,079,965-125,739,965; 660 kb; hg38) through transdifferentiation. From top to bottom: Hi-C contact matrix (5 kb resolution), gene annotation, TT-seq, H3K27ac ChIP-seq, H3K27me3 ChIP-seq, CTCF ChIP-seq, Capture-C data from the viewpoint of the *TRIB1* promoter. Axes are scaled

to signal with the following ranges: RNA synthesis = 0–10455; H3K27ac = 0–5311; H3K27me3 = 0–100; CTCF = 0–12178; Capture-C = 0–1800. Annotation as in Fig. 1d. (**d**) Chromatin landscape of the *MYB* locus (chr6:134,992,474-135,472,474; 480 kb; hg38), as in panel c and Fig. 1e, with the following axes ranges: RNA synthesis = 0–12791; H3K27ac = 0–3930; H3K27me3 = 0–100; CTCF = 0–6052; Capture-C = 0–2500. (**e**) Correlation between ATAC-seq signals ( ± 500 kb from the viewpoint) and MCC interaction frequencies, based on Spearman's correlation test. (**f**) Correlation between differential enhancer-promoter interaction frequencies and differential H3K27ac levels at the interacting elements (24 h vs 0 h), based on Spearman's correlation test. (**g**) Correlation between differential enhancer-promoter interaction frequencies and differential eRNA levels (24 h vs 0 h and 96 h vs 0 h), derived from TT-seq, as in panel f. (**h-j**) Histogram showing the distribution of the number of enhancer (**h**), promoter (**i**) and CBS (**j**) interactions with the promoter viewpoints per locus as detected by MCC. (**k**) Overview of the average number of enhancer, promoter, and CBS interactions with the promoter viewpoints as detected by MCC.

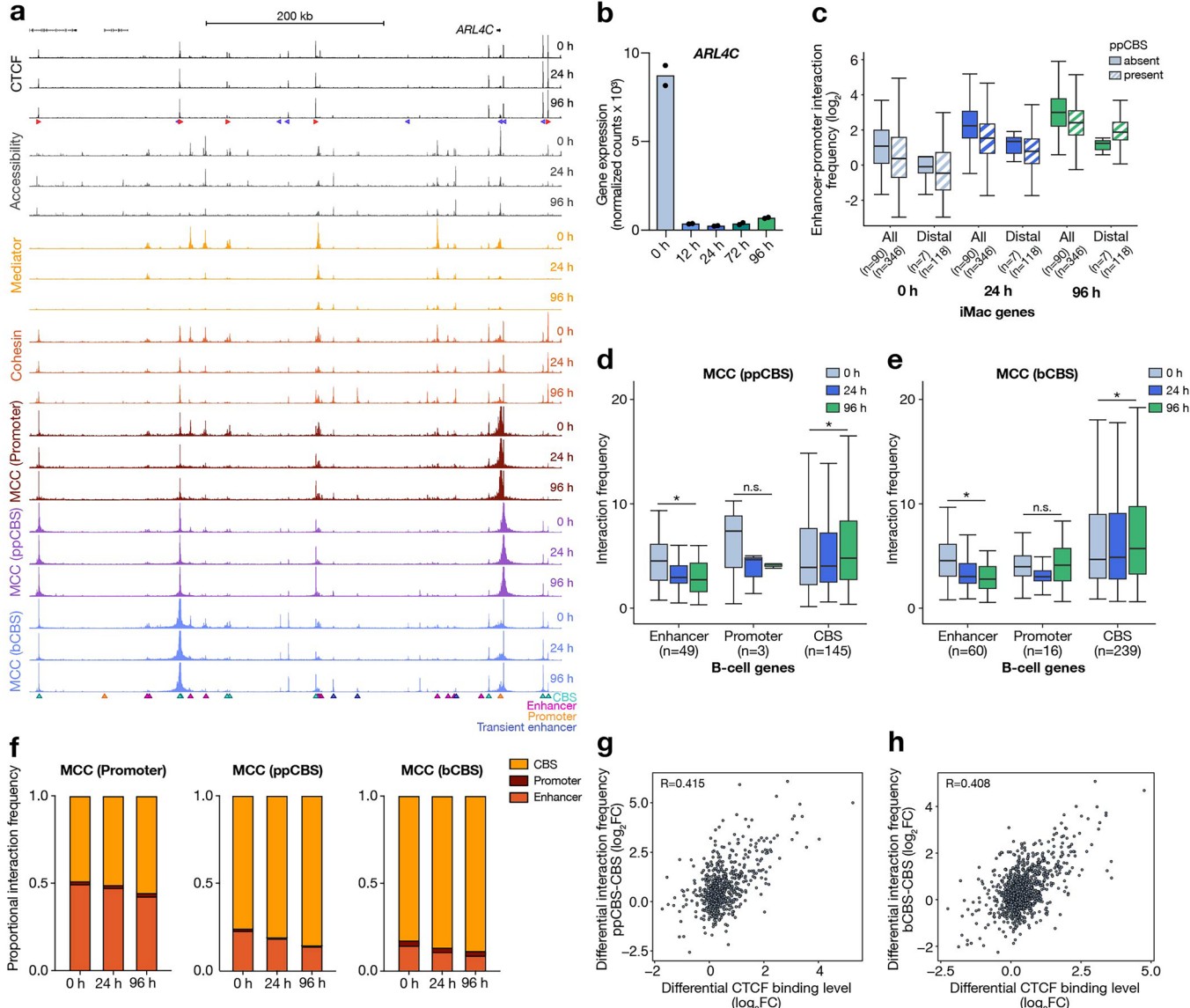

**Extended Data Fig. 2 | Interaction patterns of promoters and CTCF-binding sites through lymphoid-to-myeloid transdifferentiation.** (**a**) Chromatin landscape of the *ARLC4* locus (chr2:233,966,201-234,566,201; 600 kb; hg38) through transdifferentiation, as in Fig. 2a, with the following axes ranges: CTCF = 0–9200; Accessibility = 0–3204; Mediator = 0–5636; Cohesin = 0–2515; MCC = 0–40. (**b**) *ARL4C* expression through transdifferentiation, as in Fig. 1b. (**c**) Comparison of enhancer-promoter interaction frequencies of iMac-specific genes with and without a ppCBS ( ± 5 kb from the promoter), as in Fig. 1f. Analysis is performed for all interacting enhancers and distal ( > 150 kb) enhancers only and shows that iMac genes with a ppCBS have lower baseline interactions with all enhancers compared to genes without a ppCBS but increased interactions

specifically with distal enhancers at 96 h. (**d**) Interaction frequencies of ppCBSs of B-cell-specific genes with enhancers, promoters, and CBSs, as in Fig. 1f; enhancer: $p = 8.8 \times 10^{-4}$; promoter: $p = 0.88$; CBS: $p = 4.0 \times 10^{-4}$. (**e**) Interaction frequencies of bCBSs of B-cell-specific genes, as in Fig. 1f; enhancer: $p = 9.7 \times 10^{-6}$; promoter: $p = 0.82$; CBS: $p = 4.0 \times 10^{-4}$. (**f**) Comparison of the proportion of interactions of promoters, ppCBSs, and bCBSs with enhancers, promoters, and CBSs in B-cell-specific gene loci (**g**) Correlation between differential interaction frequencies between ppCBS and CBSs and differential CTCF binding levels at the interacting elements (24 h vs 0 h and 96 h vs 0 h), based on Spearman's correlation test. (**h**) Correlation between differential interaction frequencies between bCBS and CBSs and differential CTCF binding levels, as in panel g.

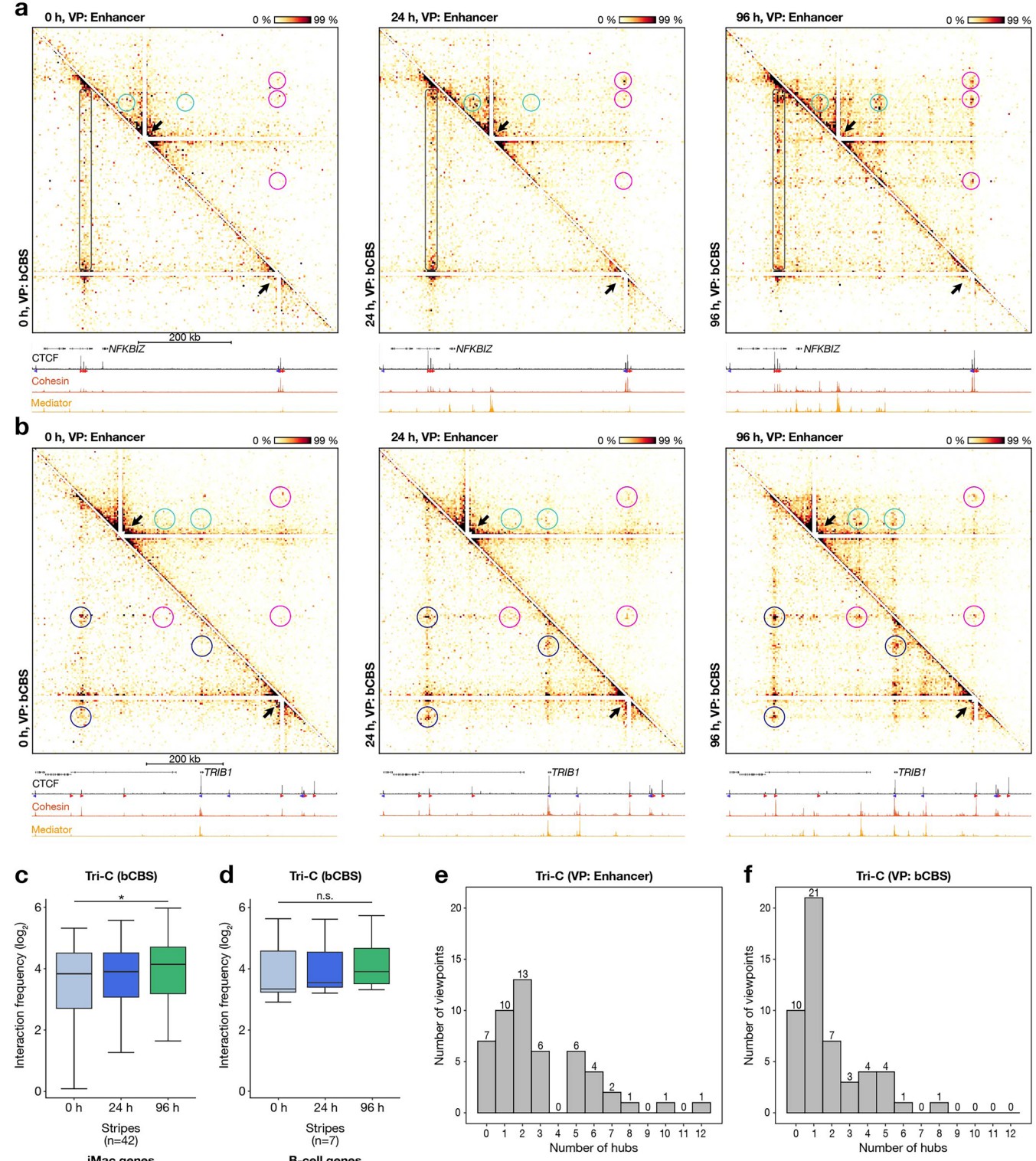

**Extended Data Fig. 3 | Dynamic chromatin hub formation during lymphoid-to-myeloid transdifferentiation.** (**a**) Tri-C contact matrices of the *NFKBIZ* locus (chr3:101,699,405-102,350,405; 651 kb; 3.5 kb resolution; hg38) through transdifferentiation, as in Fig. 3d, with E-E-P contacts highlighted in cyan circles, E-C-X and C-E/P contacts in magenta circles, C-C-C contacts in dark blue circles, and CBS stripes in grey rectangles, and with the following axes ranges: CTCF = 0–5547; Cohesin = 0–1773; Mediator = 0–2252. (**b**) Tri-C contact matrices of the *TRIB1* locus (chr8:124,988,732-125,788,732; 800 kb; 4 kb resolution; hg38)

through transdifferentiation, as in panel a, with the following axes ranges: CTCF = 0–12178; Cohesin = 0–2129; Mediator = 0–5726. (**c**) Multi-way interaction frequencies of stripes involving a CBS viewpoint and an interacting CBS in iMac-specific loci through transdifferentiation, as in Fig. 1f; p = 7.5 × 10⁻⁷. (**d**) Multi-way interaction frequencies of CBS stripes in B-cell-specific loci, as in panel 1 f; p = 0.30. (**e-f**) Histogram showing the distribution of the number of multi-way interactions with the enhancer viewpoints (**e**) and bCBS viewpoints (**f**) per locus as detected by Tri-C.

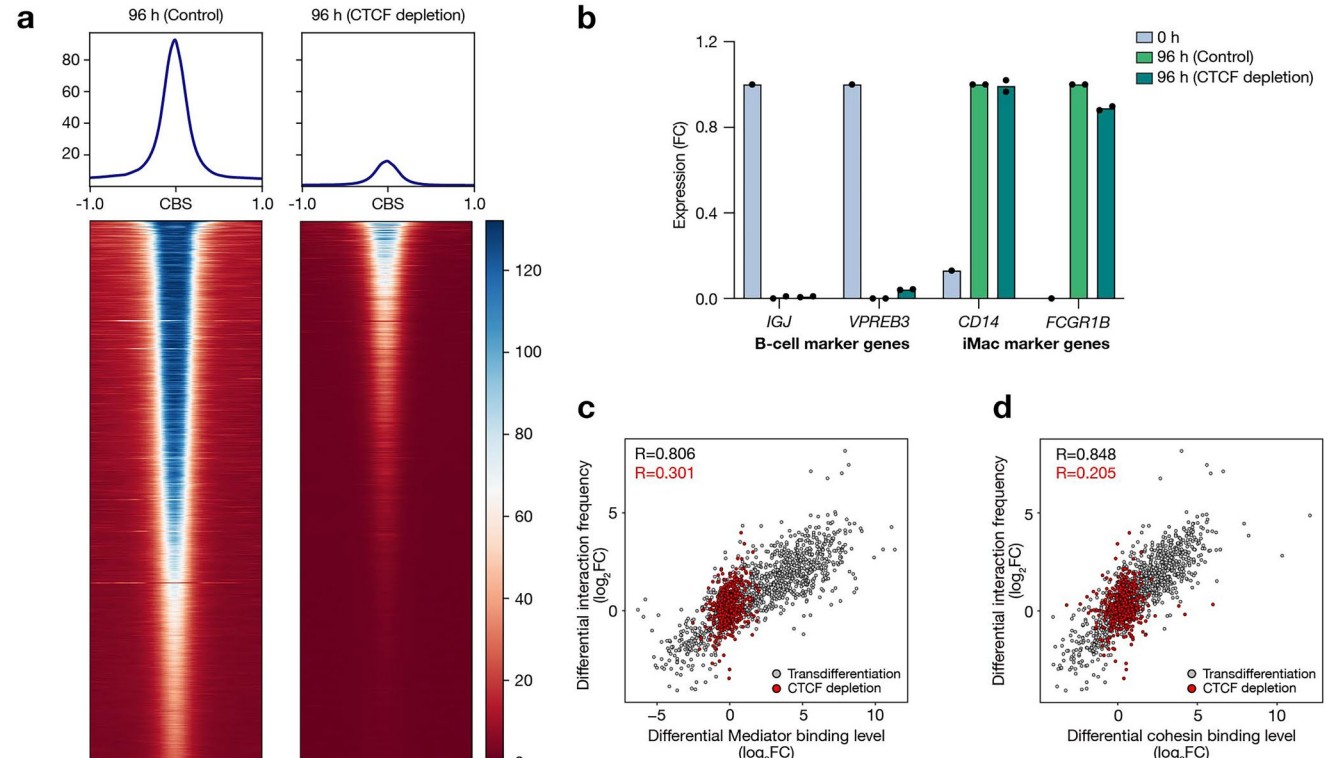

**Extended Data Fig. 4 | Characterization of CTCF depletion during lymphoid-to-myeloid transdifferentiation.** (**a**) Comparison of CTCF occupancy in control-treated and CTCF-depleted cells at 96 h after differentiation induction, measured by ChIPmentation. Data are represented in heatmaps of ±1 kb regions surrounding CTCF-binding sites (CBSs), with units in the gradient color key indicating coverage scores per genomic region. Merged data of 2 biological replicates are shown. (**b**) Expression of marker genes through transdifferentiation upon CTCF depletion, as in Extended Data Fig. 1a. Bars represent the average of 2 biological replicates, except for the 0 h condition,

which shows 1 biological replicate; corresponding data points are shown as dots. (**c**) Correlation between differential enhancer-promoter interaction frequencies and differential Mediator binding levels at the interacting elements during lymphoid-to-myeloid transdifferentiation (grey datapoints; 24 h vs 0 h and 96 h vs 0 h) and upon CTCF depletion (red datapoints; 96 h control vs 96 h CTCF depletion), based on Spearman's correlation test. (**d**) Correlation between differential enhancer-promoter interaction frequencies and differential cohesin binding levels, as in panel c.

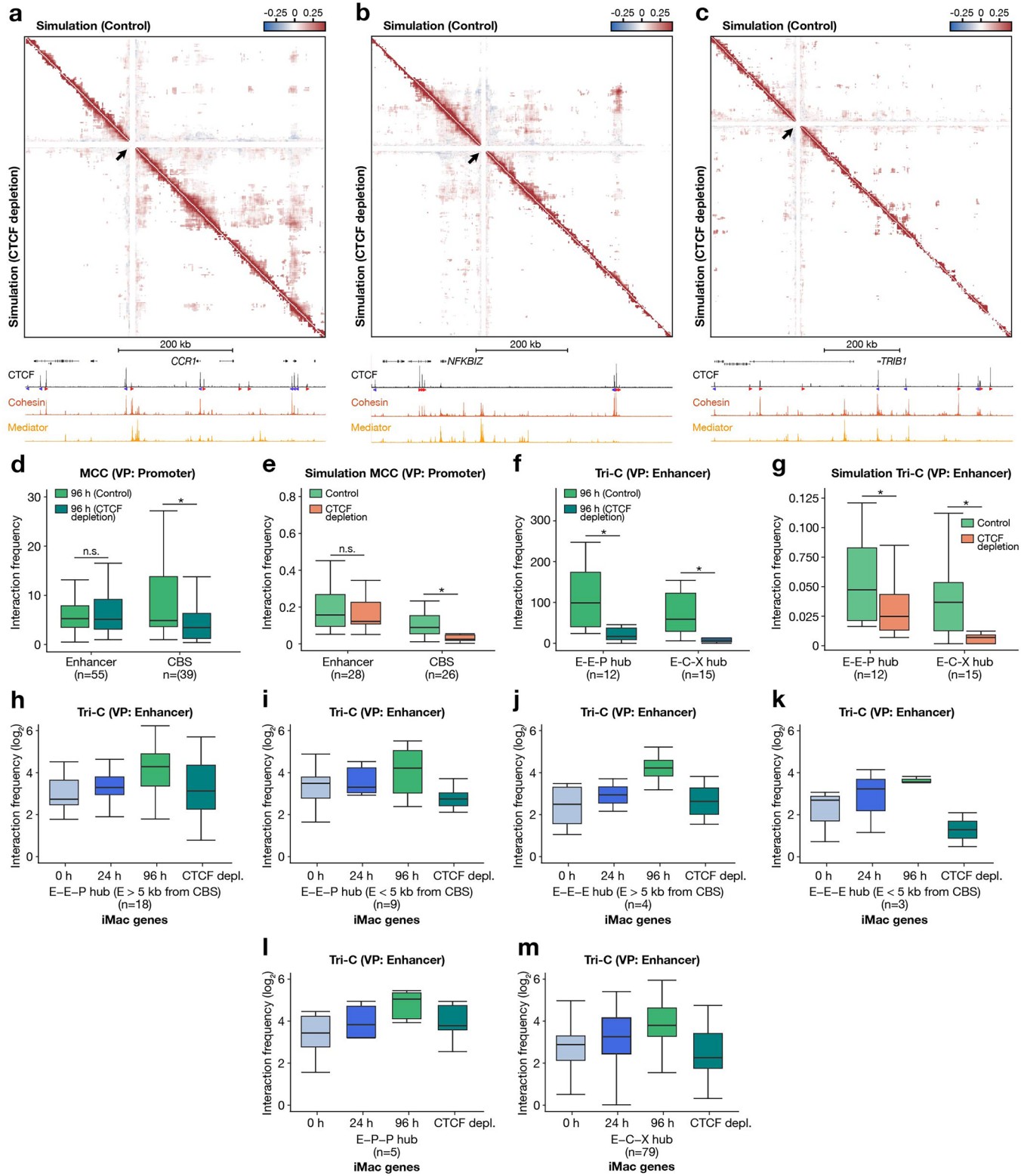

**Extended Data Fig. 5 | See next page for caption.**

**Extended Data Fig. 5 | CTCF supports the formation of enriched multi-way interactions in chromatin hubs.** (**a**) Triplet correlation coefficients of multi-way interactions with the enhancer viewpoint, generated with molecular dynamics simulations of the *CCR1* locus in control and CTCF-depleted conditions, as in Fig. 5a. Correlated (red) regions show a higher propensity for cooperative interactions with the viewpoint than expected based on their pair-wise interaction frequencies (white). (**b**) Triplet correlation coefficients of multi-way interactions with the enhancer viewpoint in the *NFKBIZ* locus in control and CTCF-depleted cells at 96 h, as in panel a and Fig. 5b. (**c**) Triplet correlation coefficients of multi-way interactions with the enhancer viewpoint in the *TRIB1* locus in control and CTCF-depleted cells at 96 h, as in panel a and Fig. 5c. (**d**) Interaction frequencies of the promoters of modelled gene loci (*CCR1*, *NFKBIZ*, and *TRIB1*) with enhancers and CBSs at 96 in control and CTCF-depleted cells, derived from experimental MCC data, as in Fig. 1f; enhancer: p = 0.84; CBS: p = 0.0019). (**e**) Interaction frequencies of the promoters of modelled gene loci (*CCR1*, *NFKBIZ*, and *TRIB1*) with enhancers and CBSs at 96 h in control and CTCF-depleted cells, extracted from the models, as in Fig. 1f; enhancer: p = 0.24; CBS: p = $1.7 \times 10^{-4}$. (**f**) Multi-way interaction frequencies of E-E-P and E-C-X hubs in the modelled gene loci at 96 h in control and CTCF-depleted cells, derived from experimental Tri-C data, as in Fig. 1f; E-E-P hubs: p = $4.9 \times 10^{-4}$; E-C-X hubs: p = $6.1 \times 10^{-5}$. (**g**) Multi-way interaction frequencies of E-E-P and E-C-X hubs in the modelled gene loci at 96 h in control and CTCF-depleted cells, extracted from the models, as in Fig. 1f; E-E-P hubs: p = $4.9 \times 10^{-4;}$ E-C-X hubs: p = $6.1 \times 10^{-5}$. (**h-m**) Frequencies of three-way interactions involving two enhancers and a promoter (E-E-P hubs), three-way interactions involving three enhancers (E-E-E hubs), three-way interactions involving one enhancer and two promoters (E-P-P hubs), and three-way interactions involving an enhancer, CTCF-binding site, and any other *cis*-regulatory element (E-C-X hubs) in iMac-specific loci at 0 h and 24 h and at 96 h in control and CTCF-depleted cells, as in Fig. 1f.

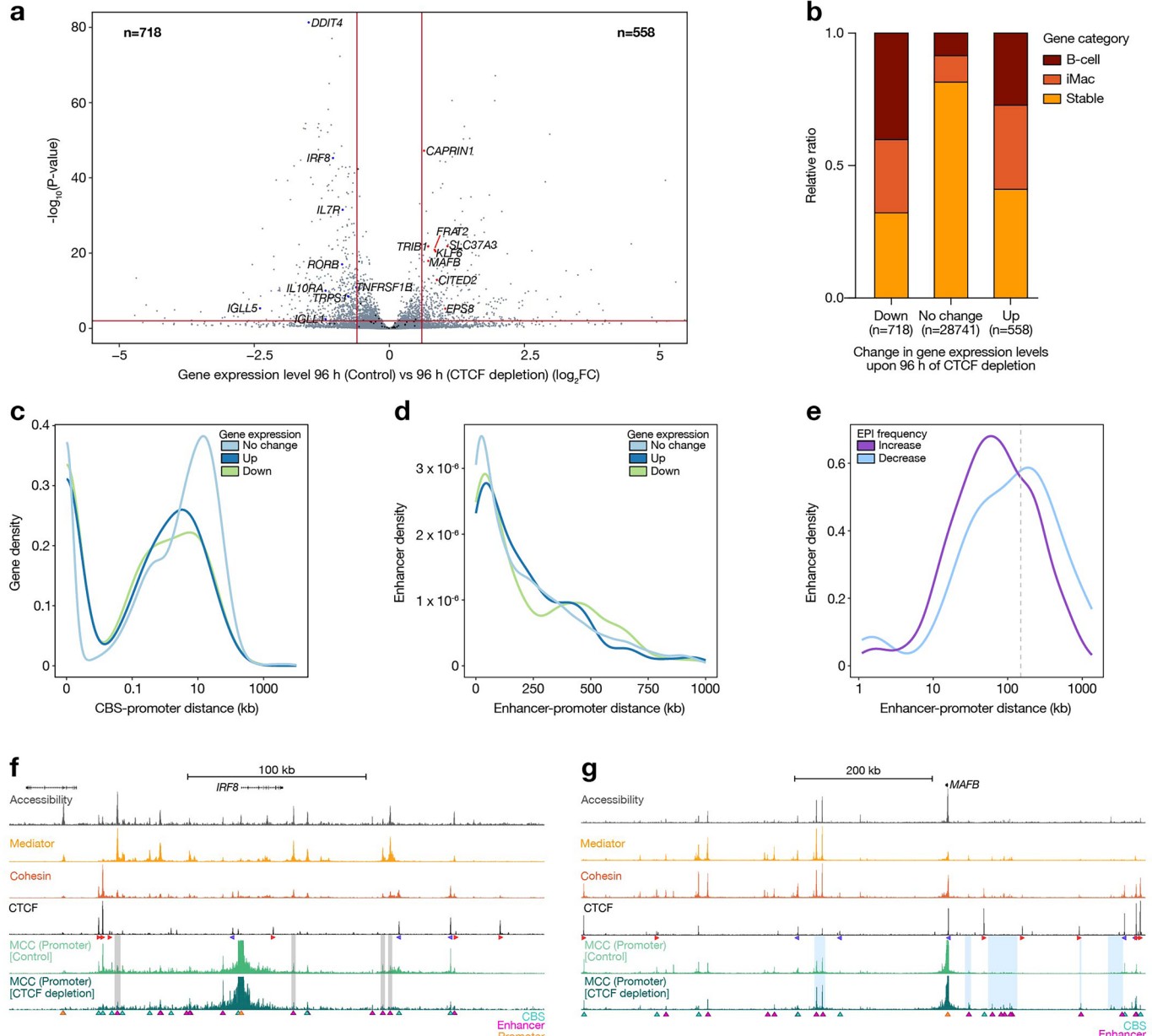

**Extended Data Fig. 6 | Changes in gene expression following CTCF depletion can be explained by rewired pair-wise enhancer-promoter interactions. (a)** Volcano plot showing differentially expressed genes between control-treated and CTCF-depleted cells at 96 h after differentiation induction, as measured by RNA-seq in 2 biological replicates. The x-axis shows the fold change (FC) in expression; the y-axis shows the adjusted p-value. The horizontal line and vertical lines indicate the significance threshold of adjusted p-value < 0.01 and effect size threshold of $log_2$FC > 0.6 or < -0.6, respectively. Statistical significance is assessed with DESeq2 using the Wald test (two-sided). P-values are adjusted for multiple comparisons using the Benjamini-Hochberg method to control the false discovery rate (FDR). Targeted genes are highlighted and those that are significantly changed upon CTCF depletion are labelled. Targeted genes described in the text that are not labelled (for example, *CCR1* and *NFKBIZ*) are not significantly up- or down-regulated upon CTCF depletion. **(b)** Comparison of the proportion of B-cell-specific genes, iMac-specific genes, and genes that are stably expressed during lymphoid-to-myeloid-transdifferentiation among

the genes that are downregulated, unchanged, or upregulated upon CTCF depletion. **(c)** Distribution of CBS-promoter distances (of the nearest CBS) of genes that are unchanged, upregulated, or downregulated upon CTCF depletion. **(d)** Distribution of enhancer-promoter distances (of all paired enhancers, see Methods) of genes that are unchanged, upregulated, or downregulated upon CTCF depletion. **(e)** Distribution of enhancer-promoter distances of increased and decreased enhancer-promoter interactions upon CTCF depletion in targeted iMac-specific loci as identified by MCC. The grey line marks the 150 kb threshold used to classify distal enhancers in Extended Data Fig. 2c. **(f)** Chromatin interactions in the *IRF8* locus (chr16:85,769,160-86,069,160; 300 kb; hg38) in control and CTCF-depleted cells, as in Fig. 6b, with the following axes ranges: Accessibility = 0-1776; Mediator = 0-4261; Cohesin = 0-3315; CTCF = 0-12417; MCC = 0-40. **(g)** Chromatin interactions in the *MAFB* locus (chr20:40,156,606-40,976,606; 820 kb; hg38) in control and CTCF-depleted cells, as in Fig. 6b, with the following axes ranges: Accessibility = 0-3567; Mediator = 0-5231; Cohesin = 0-2232; CTCF = 0-10212; MCC = 0-30.

# Reporting Summary

## Statistics

For all statistical analyses, confirm that the following items are present in the figure legend, table legend, main text, or Methods section.

| n/a | Confirmed | |
|---|---|---|
| ☐ | ☒ | The exact sample size (*n*) for each experimental group/condition, given as a discrete number and unit of measurement |
| ☐ | ☒ | A statement on whether measurements were taken from distinct samples or whether the same sample was measured repeatedly |
| ☐ | ☒ | The statistical test(s) used AND whether they are one- or two-sided <br> *Only common tests should be described solely by name; describe more complex techniques in the Methods section.* |
| ☒ | ☐ | A description of all covariates tested |
| ☒ | ☐ | A description of any assumptions or corrections, such as tests of normality and adjustment for multiple comparisons |
| ☐ | ☒ | A full description of the statistical parameters including central tendency (e.g. means) or other basic estimates (e.g. regression coefficient) AND variation (e.g. standard deviation) or associated estimates of uncertainty (e.g. confidence intervals) |
| ☐ | ☒ | For null hypothesis testing, the test statistic (e.g. *F*, *t*, *r*) with confidence intervals, effect sizes, degrees of freedom and *P* value noted <br> *Give P values as exact values whenever suitable.* |
| ☒ | ☐ | For Bayesian analysis, information on the choice of priors and Markov chain Monte Carlo settings |
| ☒ | ☐ | For hierarchical and complex designs, identification of the appropriate level for tests and full reporting of outcomes |
| ☐ | ☒ | Estimates of effect sizes (e.g. Cohen's *d*, Pearson's *r*), indicating how they were calculated |

*Our web collection on statistics for biologists contains articles on many of the points above.*

## Software and code

Policy information about availability of computer code

| | |
|---|---|
| Data collection | Illumina NextSeq 550. |
| Data analysis | CapCruncher pipeline v.0.2.3 (https://github.com/sims-lab/CapCruncher); MCC pipeline v.1 (https://github.com/jojdavies/Micro-Capture-C), based on scripts available for academic use through the Oxford University Innovation software store (v1; https://process.innovation.ox.ac.uk/software/p/165294/micro-capture-c-academic/1); Bowtie2 v.2.3.5; HiC-Pro v.2.11.1; oligo design tool v.0.1.1b (https://oligo.readthedocs.io/en/latest/); MACS2 v.2.1.2; deepTools v.3.3.0; DESeq2 v.1.36.0; NGseqBasic pipeline v.1 (https://github.com/Hughes-Genome-Group/NGseqBasic/releases); R v.4.2.0; rtracklayer v.1.56.1; UCSC bigWigMerge v.2; nf-core/rnaseq pipeline v.3.12.0 (https://nf-co.re/rnaseq/3.12.0/); STAR v.2.6.1d; featureCounts v.2.0.6; cooltools v.0.5.4; FlowJo v.10.8.1. |

For manuscripts utilizing custom algorithms or software that are central to the research but not yet described in published literature, software must be made available to editors and reviewers. We strongly encourage code deposition in a community repository (e.g. GitHub). See the Nature Portfolio guidelines for submitting code & software for further information.

## Data

Policy information about availability of data

All manuscripts must include a data availability statement. This statement should provide the following information, where applicable:

  - Accession codes, unique identifiers, or web links for publicly available datasets
  - A description of any restrictions on data availability
  - For clinical datasets or third party data, please ensure that the statement adheres to our policy

The Micro-Capture-C, Capture-C, Tri-C, ChIPmentation and RNA-seq datasets generated and analyzed for the current study are available from the Gene Expression Omnibus (GEO) as a SuperSeries under accession number GSE263641 (reviewer token: wxmfyuqqhrcptsh).
Reference genome for data from BLaER1 and BLaER1-CTCF-mAID cell lines was human assembly Dec. 2013 (GRCh38/hg38). Reference genome for Dosophila spike-in was D. melanogaster assembly Aug. 2014 (BDGP Release 6 + ISO1 MT/dm6). Index files were downloaded from Bowtie2 website (https://bowtie-bio.sourceforge.net/bowtie2/manual.shtml).

## Research involving human participants, their data, or biological material

Policy information about studies with human participants or human data. See also policy information about sex, gender (identity/presentation), and sexual orientation and race, ethnicity and racism.

| | |
|---|---|
| Reporting on sex and gender | n/a |
| Reporting on race, ethnicity, or other socially relevant groupings | n/a |
| Population characteristics | n/a |
| Recruitment | n/a |
| Ethics oversight | n/a |

Note that full information on the approval of the study protocol must also be provided in the manuscript.

# Field-specific reporting

Please select the one below that is the best fit for your research. If you are not sure, read the appropriate sections before making your selection.

☒ Life sciences  ☐ Behavioural & social sciences  ☐ Ecological, evolutionary & environmental sciences

For a reference copy of the document with all sections, see nature.com/documents/nr-reporting-summary-flat.pdf

# Life sciences study design

All studies must disclose on these points even when the disclosure is negative.

| | |
|---|---|
| Sample size | The data presented in the manuscript represent the averages of three biological replicates, which is standard in the field. No statistical method was used to pre-determine sample sizes. These sample sizes were chosen to generate data at sufficient depth and assess differences between conditions robustly. These sample sizes are sufficient, since the observed biological effects of interest are clearly detectable between conditions and robust across replicates. For Micro-Capture-C and Tri-C experiments, multiple technical replicates for each biological replicate were included to boost the complexity of the data. |
| Data exclusions | Two viewpoints (IFNGR2 and TASL) were excluded from the quantifications of Micro-Capture-C and Tri-C experiments due to poor data quality. The capture coordinates targeting promoter in Micro-Capture-C were: IFNGR2 - chr21:33403238-33403358, TASL - chrX:30577754-30577874 and in Tri-C targeting enhancer were: IFNGR2 - chr21:33216352-33216472, TASL - chrX:30596991-30597111. |
| Replication | All experiments based on sequencing data were performed for n=3 biologically independent samples as described and all attempts were successful. |
| Randomization | Samples were randomly allocated into different experimental groups prior to their treatment with auxin or control conditions. |
| Blinding | All samples were analyzed with the same pipelines, in which results are generated by scripts without interference of the researchers. Since potential expectations of the researchers cannot influence the data analysis and results, blinding is not relevant to this study. |

# Reporting for specific materials, systems and methods

We require information from authors about some types of materials, experimental systems and methods used in many studies. Here, indicate whether each material, system or method listed is relevant to your study. If you are not sure if a list item applies to your research, read the appropriate section before selecting a response.

## Materials & experimental systems

| n/a | Involved in the study |
|---|---|
| ☐ | ☒ Antibodies |
| ☐ | ☒ Eukaryotic cell lines |
| ☒ | ☐ Palaeontology and archaeology |
| ☒ | ☐ Animals and other organisms |
| ☒ | ☐ Clinical data |
| ☒ | ☐ Dual use research of concern |
| ☒ | ☐ Plants |

## Methods

| n/a | Involved in the study |
|---|---|
| ☐ | ☒ ChIP-seq |
| ☐ | ☒ Flow cytometry |
| ☒ | ☐ MRI-based neuroimaging |

## Antibodies

| | |
|---|---|
| Antibodies used | Flow cytometry: Human FcR Binding Inhibitor (1:5, eBiosciences, 16-9161-73), APC-Cy7 Mouse Anti-Human CD19 (1:20, BD Pharmingen, 348794), APC Mouse Anti-Human CD11b (1:5, BD Pharmingen, 550019).<br>ChIPmentation: Rabbit anti-CTCF (2 µg, Diagenode, C15410210-50), Rabbit anti-SMC1A (2 µg, Abcam, ab9262), Rabbit anti-MED26 (2 µg, Bethyl Laboratories, A302-370), Drosophila spike-in antibody (1 µg, Biozol, 61686). |
| Validation | Validation was performed by the manufacturer. The antibodies were purified using immunogen affinity and validated by immunoprecipitation, immunohistochemical analysis, and western blotting.<br>- Human FcR Binding Inhibitor (eBiosciences, 16-9161-73) validated for flow cytometry (https://www.thermofisher.com/antibody/product/Fc-Receptor-Binding-Inhibitor-Antibody-Polyclonal/16-9161-73).<br>- APC-Cy7 Mouse Anti-Human CD19 (BD Pharmingen, 348794) validated for flow cytometry (https://www.bdbiosciences.com/en-us/products/reagents/flow-cytometry-reagents/clinical-discovery-research/single-color-antibodies-ruo-gmp/apc-cy-7-mouse-anti-human-cd19.348794?tab=format_details).<br>- APC Mouse Anti-Human CD11b (BD Pharmingen, 550019) validated for flow cytometry (https://www.bdbiosciences.com/en-ca/products/reagents/flow-cytometry-reagents/research-reagents/single-color-antibodies-ruo/apc-mouse-anti-human-cd11b.550019?tab=format_details).<br>- Rabbit anti-CTCF (2 µg, Diagenode, C15410210-50), ChIP-grade, validated for Western Blot and ChIP-seq (https://www.diagenode.com/en/p/ctcf-polyclonal-antibody-classic-50-mg).<br>- Rabbit anti-SMC1A (2 µg, Abcam, ab9262) validated for Western Blot and Immunoprecipitation (https://www.abcam.com/en-us/products/primary-antibodies/smc1a-antibody-ab9262?srsltid=AfmBOopPzSVjxgsokl23cisdMQFvcEjvAh9wVoN9u2UgEWLZLY1slpnW#). Validated for ChIP in Dluhosova et al., 2014: https://doi.org/10.1371/journal.pone.0092635.<br>- Rabbit anti-MED26 (2 µg, Bethyl Laboratories, A302-370) validated for Western Blot and Immunoprecipitation (https://www.thermofisher.com/antibody/product/CRSP7-Antibody-Polyclonal/A302-370A).<br>- Drosophila spike-in antibody (1 µg, Biozol, 61686) validated by Active Motif for ChIP-seq spike-in normalisation (https://www.activemotif.com/catalog/1091/chip-normalization). |

## Eukaryotic cell lines

Policy information about cell lines and Sex and Gender in Research

| | |
|---|---|
| Cell line source(s) | The wild type Human B-cell Precursor Leukemia Cell Line (BLaER1) was a gift from the laboratory of Patrick Cramer (MPI-NAT, Goettingen). This cell line was originally created by the laboratory of Thomas Graf (CGR, Barcelona) in Rapino et al., 2013, Cell Reports from the parental B cell precursor leukemia (RCH-ACV, ACC 548) cell line. The cell line is also available commercially from Sigma-Aldrich (SCC165). The BLaER1-CTCF-mAID cell line was a gift from Grégoire Stik (CRG, Barcelona), and was created from the parental BLaER1 cell line in Stik et al., 2020, Nature Genetics. |
| Authentication | The BLaER1 cells were authenticated using the KaryoStat+ assay (Thermo Fisher). The BLaER1-CTCF-mAID cell line was created from the authenticated BLaER1 line and not further authenticated. |
| Mycoplasma contamination | All cell lines tested negative for mycoplasma contamination. |
| Commonly misidentified lines<br>(See ICLAC register) | No commonly misidentified lines were used. |

## Plants

Seed stocks

n/a

Novel plant genotypes

n/a

Authentication

n/a

## ChIP-seq

### Data deposition

☒ Confirm that both raw and final processed data have been deposited in a public database such as GEO.

☒ Confirm that you have deposited or provided access to graph files (e.g. BED files) for the called peaks.

Data access links
*May remain private before publication.*

GEO link: https://www.ncbi.nlm.nih.gov/geo/query/acc.cgi?acc=GSE263641
Reviewer token: wxmfyuqqhrcptsh

Files in database submission

GSM8195732  BLaER, SMC1A_ChIP_0h_rep1
GSM8195733  BLaER, SMC1A_ChIP_0h_rep2
GSM8195734  BLaER, SMC1A_ChIP_0h_rep3
GSM8195735  BLaER, SMC1A_ChIP_12h_rep1
GSM8195736  BLaER, SMC1A_ChIP_12h_rep2
GSM8195737  BLaER, SMC1A_ChIP_12h_rep3
GSM8195738  BLaER, SMC1A_ChIP_24h_rep1
GSM8195739  BLaER, SMC1A_ChIP_24h_rep2
GSM8195740  BLaER, SMC1A_ChIP_24h_rep3
GSM8195741  BLaER, SMC1A_ChIP_72h_rep1
GSM8195742  BLaER, SMC1A_ChIP_72h_rep2
GSM8195743  BLaER, SMC1A_ChIP_72h_rep3
GSM8195744  BLaER, SMC1A_ChIP_96h_rep1
GSM8195745  BLaER, SMC1A_ChIP_96h_rep2
GSM8195746  BLaER, SMC1A_ChIP_96h_rep3
GSM8195747  BLaER, MED26_ChIP_0h_rep1
GSM8195748  BLaER, MED26_ChIP_0h_rep2
GSM8195749  BLaER, MED26_ChIP_0h_rep3
GSM8195750  BLaER, MED26_ChIP_12h_rep1
GSM8195751  BLaER, MED26_ChIP_12h_rep2
GSM8195752  BLaER, MED26_ChIP_12h_rep3
GSM8195753  BLaER, MED26_ChIP_24h_rep1
GSM8195754  BLaER, MED26_ChIP_24h_rep2
GSM8195755  BLaER, MED26_ChIP_24h_rep3
GSM8195762  BLaER subclone CTCF-AID, CTCF_ChIP _96h_control_rep1
GSM8195763  BLaER subclone CTCF-AID, CTCF_ChIP _96h_control_rep2
GSM8195764  BLaER subclone CTCF-AID, CTCF_ChIP _96h_depl_rep1
GSM8195765  BLaER subclone CTCF-AID, CTCF_ChIP _96h_depl_rep2
GSM8195756  BLaER, MED26_ChIP_72h_rep1
GSM8195757  BLaER, MED26_ChIP_72h_rep2
GSM8195758  BLaER, MED26_ChIP_72h_rep3
GSM8195759  BLaER, MED26_ChIP_96h_rep1
GSM8195760  BLaER, MED26_ChIP_96h_rep2
GSM8195761  BLaER, MED26_ChIP_96h_rep3
GSM8740797 BLaER subclone CTCF-AID, SMC1A_ChIP _96h_control_rep1
GSM8740798 BLaER subclone CTCF-AID, SMC1A_ChIP _96h_control_rep2
GSM8740799 BLaER subclone CTCF-AID, SMC1A_ChIP _96h_depl_rep1
GSM8740800 BLaER subclone CTCF-AID, SMC1A_ChIP _96h_depl_rep2
GSM8740801 BLaER subclone CTCF-AID, MED26_ChIP _96h_control_rep1
GSM8740802 BLaER subclone CTCF-AID,MED26_ChIP _96h_control_rep2
GSM8740803 BLaER subclone CTCF-AID, MED26_ChIP _96h_depl_rep1
GSM8740804 BLaER subclone CTCF-AID, MED26_ChIP _96h_depl_rep2

Genome browser session
(e.g. UCSC)

https://genome-euro.ucsc.edu/cgi-bin/hgTracks?
db=hg38&lastVirtModeType=default&lastVirtModeExtraState=&virtModeType=default&virtMode=0&nonVirtPosition=&posit
ion=chr6%3A133824244%2D136599852&hgsid=348343346_rZSyfdVVteG8voU2PaaW4P2jFgAO

## Methodology

| | |
|---|---|
| Replicates | ChIPmentation experiments for MED26 and SMC1A were performed in 3 biological replicates per condition; ChIPmentation experiments for CTCF were performed in 2 biological replicates per condition; ChIPmentation experiments for MED26 and SMC1A after CTCF depletion were performed in 2 biological replicates per condition. |
| Sequencing depth | The samples were sequenced using the NextSeq550 Illumina platform (75-bp paired-end reads) to a sequencing depth of ~20 M reads per sample. |
| Antibodies | ChIPmentation: Rabbit anti-CTCF (2 μg, Diagenode, C15410210-50), Rabbit anti-SMC1A (2 μg, Abcam, ab9262), Rabbit anti-MED26 (2 μg, Bethyl Laboratories, A302-370), Drosophila spike-in antibody (1 μg, Biozol, 61686). |
| Peak calling parameters | Paired-end reads were processed for adapter removal and duplicate filtering and mapped to the hg38 reference genome using Bowtie2. Peak calling was performed with MACS2 (consensus peaks with parameter q = 0.1 were selected). All peak profiles were generated using Deeptools. |
| Data quality | The quality of the data was assessed by comparing CTCF-control samples to available CTCF ChIP-seq data (Stik, et al., 2020, Nature Genetics). As SMC1A and MED26 ChiPmentation experiments have been performed for the first time in this cell line, data quality was assessed by comparing 3 biological replicates. |
| Software | Paired-end reads were processed for adapter removal and duplicate filtering and mapped to the hg38 reference genome using Bowtie2. Peak calling was performed with MACS2. All peak profiles were generated using Deeptools. |

# Flow Cytometry

## Plots

Confirm that:

☒ The axis labels state the marker and fluorochrome used (e.g. CD4-FITC).

☐ The axis scales are clearly visible. Include numbers along axes only for bottom left plot of group (a 'group' is an analysis of identical markers).

☒ All plots are contour plots with outliers or pseudocolor plots.

☒ A numerical value for number of cells or percentage (with statistics) is provided.

## Methodology

| | |
|---|---|
| Sample preparation | The BLaER1 cells were cultured and differentiated and flow cytometry analysis was performed at different differentiation stages for 2 biological replicates. Cells were harvested, washed with PBS, blocked with human Fc Receptor Binding Antibody, stained with antibodies, washed with PBS and analyzed. A more detailed protocol is described in the Methods section. |
| Instrument | Sony SH800 Cell Sorter (MPI-NAT, Goettingen). |
| Software | FloJo v.10.8.1 |
| Cell population abundance | Cells were analyzed for the expression of the cell surface markers CD19 (B cell marker) and CD11b (iMacs marker). All the BLaER1 cells were positive for CD19 and after 7 days of transdifferentiation around 90 % of the cells lost the CD19 marker and acquired the CD11b marker. |
| Gating strategy | Initial gating was performed based on forward and side scatters to identify single cells and exclude cell debris. Firstly, a gate was set on forward scatter height (FSC-H) vs forward scatter area (FSC-A) plot to exclude doublets. Secondly, a gate was set on side scatter height (SSC-H) vs forward scatter area (FSC-A) plot to exclude debris cells. Thirdly, a gate was set on the histogram to include only GFP positive, alive cells. Quadrant gates were based on non-differentiated B-cell population as negative control. They were set on 0 h sample, which is CD19 positive and CD11b negative. All relevant data are shown in Extended Data Fig. 1b. |

☒ Tick this box to confirm that a figure exemplifying the gating strategy is provided in the Supplementary Information.

