## [Peer Review File · Nature Structural & Molecular Biology]

CTCF depletion decouples enhancer-mediated gene activation from chromatin hub formation

Corresponding Author: Dr Marieke Oudelaar

Version 0:

Decision Letter:

14th Nov 2024

Dear Dr. Oudelaar,

Thank you again for your patience and for submitting your manuscript "CTCF depletion decouples enhancer-mediated gene activation from chromatin hub formation during cellular differentiation". We now have comments (below) from the 3 reviewers who evaluated your paper. In light of those reports, we remain interested in your study and would like to see your response to the comments of the referees, in the form of a revised manuscript.

Please be sure to address/respond to all concerns of the referees in full in a point-by-point response (particularly the concerns outlined by Reviewer #2, including the request for additional ChIP-seq profiling and integration of relevant public datasets). Please also highlight all changes in the revised manuscript text file. If you have comments that are intended for editors only, please include those in a separate cover letter.

We expect to see your revised manuscript within the next 3-6 months. If you cannot send it within this time, please contact us to discuss an extension; we would still consider your revision, provided that no similar work has been accepted for publication at NSMB or published elsewhere.

Reporting Summary:

Please note that all key data shown in the main figures as cropped gels or blots should be presented in uncropped form, with molecular weight markers. These data can be aggregated into a single supplementary figure item. While these data can be displayed in a relatively informal style, they must refer back to the relevant figures. These data should be submitted with the final revision, as source data, prior to acceptance, but you may want to start putting it together at this point.

Data availability: this journal strongly supports public availability of data. All data used in accepted papers should be available via a public data repository, or alternatively, as Supplementary Information. If data can only be shared on request, please explain why in your Data Availability Statement, and also in the correspondence with your editor. Please note that for some data types, deposition in a public repository is mandatory - more information on our data deposition policies and available repositories can be found below:

<https://www.nature.com/nature-research/editorial-policies/reporting-standards#availability-of-data>

Link Redacted

Best regards,

George

George Inglis, PhD
Senior Editor

<https://www.nature.com/nsmb/research-cross-journal-editorial-team> Research Cross-Journal Editorial Team
Nature Structural & Molecular Biology

Referee expertise:

Referee #1: Enhancer-promoter interactions, CTCF dynamics, 3D genome organization

Referee #2: CTCF dynamics, 3D genome organization

Referee #3: MD simulations in chromatin folding and genome organization

Reviewers' Comments:

Reviewer #1 (Remarks to the Author):

The study by Karpinska et al expands on the work by Stik and colleagues (PMID: 32514124) that had described modest gene expression changes upon CTCF degradation, during conversion of leukemic B cells to Macrophages. Here, the authors confirmed these observations and incorporate high-resolution Micro-Capture-C (MCC), and Tri-C during the same trans-differentiation experiment. They then assess the impact of degrading CTCF on gene expression, and on pairwise and multi-way interactions.

The data presented here are of very high quality and the claims are well supported. The MCC experiments show important correlations between changes in expression and enhancer–promoter contacts, but the major novelty of the work resides in the identification by Tri-C of three-way interactions at cell-type specific enhancers that require CTCF for their formation but not for gene regulation of the genes within the hubs. At a time that the physiological role of hubs is under intense discussion, this observation suggests that regulation of gene expression of genes within hubs does not require formation of these structures.

I have the following main comments suggestions/comments for the authors:

1 - B cells do not differentiate to Macrophages. I think it would be important to refer to this process as transdifferentiation or cellular conversion as the Graf lab normally does. Or at least clearly mention that this is not a physiological or a model of a differentiation process.

3 - Did all regions assessed by Tri-C show evidence of formation of enhancer-containing multi-way interactions similar to the loci shown in figure 3 and extended data figure 3? That is not possible to assess from the quantifications in figure 3f-i.

4 – Whether the number of gene expression changes upon CTCF depletion is large, modest, or minor, is a matter of interpretation and expectation. Rather than using these qualitative terms it would probably be better to show how many genes are labelled as upregulated and downregulated in the plot in Extended Data Fig6. I also couldn't see a table with that list. I also think that extended data figure 6 should be in the main figure. Probably more important than the current Figure 6A.

5- Along these lines, it isn't clear to me how much the expression of genes with MCC viewpoints changes upon CTCF depletion and how much their interactions change. While Figure 2 was highly correlational repeating the promoter and promoter-proximal CBS analyses with and without CTCF would be more functional and interesting. Throughout the paper the authors did an effort to always show a few gene examples and then quantify what happens at all viewpoints. However when it came to CTCF degradation, this was not done and instead we see just a few example loci. It would be nice to have a better understanding of how much CTCF depletion affected the loci for which viewpoints were designed.

6 - Some genes are strongly impacted by CTCF degradation. It would be important to characterize these. Are they B or iMac specific genes? Do they have proximal or distal enhancers? Do they form tri-way interactions. It seems to me that viewpoint choice shouldn't have been limited by genes that are differentially expressed during cell conversion but instead also include genes that are differentially expressed upon CTCF depletion.

7 – It would probably be helpful to readers to add a drawing/model on how the authors interpret their data. Especially representing the CTCF versus noCTCF condition

Minor issues:

1 - Although there is nothing wrong in the title, you need to read the whole paper before you understand what the authors mean. Maybe something that more directly reflects their main finding would be more appropriate. For example, something along the lines of: Formation of CTCF-mediated cell type-specific enhancer hubs does not affect gene regulation.

2 - In the abstract the authors highlight concepts and technical details that are hardly mentioned in their results. For example, single-allele topologies are not mentioned anywhere in the results and therefore either the concept shouldn't be used in the abstract or use the same language should be used in the results. The authors highlight base-pair resolution but in the description of the results it isn't really clear why this is important. 10 or 100bp bins would also work to distinguish promoter and promoter-proximal CBS, right? I understand that the base pair resolution is used for MCC quantification, but I don't believe the authors describe it in the results or explain its advantages. If they don't, then it shouldn't be highlighted in the abstract either. The sentence starting at line 26 is a bit ambiguous and does not clearly reflect the authors intent of saying that gene expression changes explained by rewired E-P contacts are modest. As it stands, it can also be interpreted as saying that the mild effects on gene expression can be explained by rewired E-P contacts. This is not that the authors mean, I believe.

3 - This is a comment for almost all boxplots but especially figure 2c. It would be helpful to know how many datapoints there are in each boxplot. Concerning this figure, I had a hard time following its description and I am not sure it adds much to the paper compared with the other plots in this same figure. It isn't clear to me what the differences between All and Distal only represent, and whether the presence or absence of Promoter Proximal CBS influences these interactions much. This needs to either be better summarized or potentially removed.

4 - Figure 4E-can the authors point to a promoter-promoter interaction that increased upon CTCF depletion? I had a hard

time finding one. If the genes shown don't have one maybe had to at least extended data?

Reviewer #2 (Remarks to the Author):

The manuscript by Karpinska, Zhu et al investigate the formation of 3D chromatin structures during cell differentiation, and their link with transcriptional regulation. For this purpose, the authors employ a B-cell leukemia cell line (BLaER1) that can be converted into induced macrophages (iMACSs) by the expression of CEBPA. At different time points of the differentiation process the authors generate state-of-the-art Micro-Capture-C and Tri-C data for a panel of loci containing B-cell-specific and iMAC-specific genes. The datasets include viewpoints on promoters and additional cis-regulatory elements, such as enhancers or CTCF-binding sites (CBS). These datasets are further complemented with binding profiles for Mediator and Cohesin, as well as with publicly available data for gene expression, chromatin accessibility and H3K27Ac and CTCF binding. This constitutes an optimal and comprehensive resource to pursue the questions that the authors pose.

Overall, the analyses performed in this differentiation system show that the dynamics enhancer-promoter interactions at activating /repressed promoters correlate well with those of Mediator and Cohesin binding. Tri-C experiments also revealed that interactions between promoters, enhancers and CBS exist in the context of 3D chromatin hubs, and that their formation and dissolution also mimics the dynamics of gene expression. The study also explores an interesting phenomenon, which is the recurrent presence of CBS near promoters. MCC data revealed that those sites display a preference to interact with enhancers than their counterparts located Topologically Associating Domains (TADs) boundaries, suggesting the existence of different functional classes of CBS. The authors further explore the role of CTCF in chromatin hub formation, by coupling the BLaER1 differentiation system with an auxin degron. These experiments revealed that CTCF impairment results has a considerable impact on 3D chromatin hub structures, but minor effects on individual enhancer-promoter interactions. This is a well-rounded study providing high quality datasets that provide novel insights into the regulatory dynamics of differentiation systems. The derived results contribute significantly to the ongoing discussion on the instructive role of chromatin structure on gene expression. Overall, the manuscript is suited for the readership of Nature Structural and Molecular Biology. Yet, there are certain aspects of the study that require further clarification.

Major comments:

- The number of viewpoints studied in MCC/Tri-C experiments seems sufficient to inform on generalizable gene regulatory principles. In that respect, it would be recommended that the authors provide additional statistics such as how many enhancers, CBS and additional promoters compose on average the regulatory landscapes of the studied genes. Since these analyses have been performed across several stages, it would be useful to examine how the composition of the landscapes evolves over differentiation time. Further, what percentage of the individual enhancers detected in MCC experiments engage in multi-way contacts in Tri-C data?

- Fig 1e displays data profiles that are representative of genes undergoing repression. Mediator and MCC profiles display dynamic changes at the upstream gene region, suggesting a loss of contact with enhancer elements. However, the downstream region shows opposite dynamics: contacts are gained with other elements. This profile may suggest that interaction with other type of regulatory elements (i.e. repressors) could be in play. Are those patterns recurrently observed for other repressed genes? Is the opposite trend also observed during the activation of genes (i.e. a loss of contacts with specific genomic regions)? Have the authors considered integrating additional epigenetic datasets that mark repression in their analysis (for example H3K27me3 from GSE257528), to gain further mechanistic insights?

- The authors generate MCC profiles for promoter proximal and boundary CBS (ppCBS and bCBS respectively), finding that each category has distinct interaction preferences. I missed the contextualization of those findings in the discussion, in particular with respect to previous studies that have also pointed to the existence of different CBS categories (for example Huang et al Nat Genet 2021).

Regarding this analysis, it is also not entirely clear what were the exact criteria followed to allocate CBS in the two categories. In which category would be allocated those CBS at the promoters of genes that are close or embedded in TAD boundary? Such "intermediate" cases seem to exist in the dataset. For example, at the TRIB1 locus from Fig1c, the gene is located in a region that may well be called as a TAD boundary, depending on the parameters of the detection algorithm. How would those "intermediate" cases behave in terms of interactions with respect to the effects observed in the ppCBS and bCBS categories?

- Lines 443-444: "However, interestingly, cooperative interactions involving only enhancers and promoters are also notably decreased."

The global data from Fig. 5j supports this observation. Yet, this effect does not seem to occur at the NFKBIZ locus (cyan circles in Fig 5b). In this case, cooperative interactions between enhancers and promoter are largely preserved. I wonder if this effect could be explained by the lack of CBS inside the NFKBIZ TAD, in comparison with the CCR1 and TRIB1 loci. Can the authors check if there is an influence of intra-TAD CBS density on enhancer-promoter cooperative interactions?

- Fig 4e. Does the increase of promoter-promoter interaction upon CTCF depletion occur preferentially between genes that

would be otherwise located in different TADs?

- The analysis on Figs 1h and I are interesting as they suggest that Mediator and Cohesin are important to support enhancer-promoter interactions. In that respect, it would have been nice to complement these analyses with ChIP-seq experiments for Mediator in Cohesin in CTCF-depleted cells and subsequent analogous correlation analyses with differential enhancer-promoter interactions against control cells.

- Lines 567-569: "...but that the higher-order configuration of enhancers and promoters in CTCF-dependent hub structures does not have a specific function in gene regulation".

This is a strong statement that cannot be supported by the data included in the manuscript. One cannot simply exclude that the alterations in chromatin hubs have also certain contribution on the observed changes in transcription. Furthermore, gene regulation is more complex than what can be inferred by simply looking at expression levels in a individual timepoint. For example, temporal changes in gene activation or alterations in spatial patterns are aspects that cannot be measured in this differentiation system, and that may well be compromised by the impairment of chromatin hubs. I would suggest that the authors tone down this statement.

Minor comments:

- Line 129: "Characterization of this system by qPCR and FACS shows that cellular differentiation occurs synchronously and completely, with >90% of B-cells converting into iMac3s over the differentiation course"

This percentage is not correct, based on the flow cytometry profile at 168 h.

- Capture-C experiments were also performed in this study. Yet, the only reference in the manuscript are a few profiles in Extended Data Fig 1c and d. It would be appropriate to refer to these experiments in the main text, explaining the motivation to perform them as well as discussing the results obtained.

- It would be useful to add the CTCF ChIP-seq profiles of the different timepoints in Fig 2 and Extended Data Fig. 2, as it is shown for the other experiments in these figures.

- It could be good for the flow of the text to provide background information on the modeling framework in the main text. Alternatively, the authors may indicate in the main text that the principles of the model are extensively described in the Methods section.

Reviewer #3 (Remarks to the Author):

The work by Karpinska et al. combines Micro-Capture-C, Tri-C, and simulations to investigate the interplay between gene expression and 3D genome structure over the course of cell differentiation. This is a challenging problem, with a literature showing contradictory evidence supporting alternative theories. The study is remarkable in that it produced rich and high-resolution information clarifying how CTCF and cohesin shape the structure of chromatin and support the formation of enhancer-promoter contacts that likely explain changes in gene expression. One major finding to me is the observation of a clear correlation between enhancer-promoter contacts and gene expression level, likely made possible by the high resolution of their data, which is important considering the role of 3D chromatin conformation for gene regulation is still being questioned in the literature. Another is the observation that the minor effect of CTCF degradation on gene expression is due to the fact that the original co-operative 3-way contacts favored by stable stalling at CBS is compensated by pairwise contacts in a way that the overall strength of enhancer-promoter interactions remains mostly unaffected. While these observations are perhaps not entirely surprising, I believe that they are very important for both the chromatin and gene regulation fields. Overall, I find the methodology and analysis appropriate and the conclusions robust.

I would like to comment mainly on the simulation methods and their presentation. I find that the simulations are critical to rationalize the observations that CTCF depletion affects 3-way contact while leaving pairwise enhancer-promoter interactions unaffected. For this, the specific way in which CTCF, cohesin and RNAPol are treated is fundamental, but this is not clarified in the paper itself. I believe adding more information about the model is important to explain what is actually going on inside the real system. My major issue about the modeling results is that the authors do not show evidence from the model that CTCF depletion would be expected to leave the pairwise enhancer-promoter contacts not affected, as only the effect on 3-way interactions is actually reported. A more detailed analysis of the simulations would therefore be helpful.

Version 1:

Decision Letter:

Dear Dr. Oudelaar,

Thank you for your patience and for submitting your revised manuscript "CTCF depletion decouples enhancer-mediated gene activation from chromatin hub formation during cellular differentiation" (NSMB-A49808A). It has now been seen by the original referees and their comments are below. The reviewers find that the paper has improved in revision, and therefore

we'll be happy in principle to publish it in Nature Structural & Molecular Biology, pending minor revisions to satisfy the referees' final requests and to comply with our editorial and formatting guidelines.

We are now performing detailed checks on your paper and will send you a checklist detailing our editorial and formatting requirements in the next 1-2 weeks. Please do not upload the final materials and make any revisions until you receive this additional information from us.

To facilitate our work at this stage, it is important that we have a copy of the main text as a word file. If you could please send along a word version of this file as soon as possible, we would greatly appreciate it; please make sure to copy the NSMB account (cc'ed above).

Best regards,

George

George Inglis, PhD
Senior Editor

[Research Cross-Journal Editorial Team](https://www.nature.com/nsmb/research-cross-journal-editorial-team)
Nature Structural & Molecular Biology

Reviewer #1 (Remarks to the Author):

Thanks for addressing my comments. I believe this paper is ready for publication at NSMB.

Reviewer #2 (Remarks to the Author):

I would like to thank the authors for performing additional experiments and analyses to complement their initial conclusions.

This new version of the manuscript addresses the concerns raised through the review process and I am happy to support the study for its publication.

Reviewer #3 (Remarks to the Author):

The reviewers have carefully addressed my previous comments on the simulations, I have nothing to add.

Version 2:

Decision Letter:

Dear Dr. Oudelaar,

Thank you for your patience during this last stage of the editorial process. We are now happy to accept your revised paper "CTCF depletion decouples enhancer-mediated gene activation from chromatin hub formation" for publication as an Article in Nature Structural & Molecular Biology.

Your paper will be published online soon after we receive proof corrections and will appear in print in the next available issue. You can find out your date of online publication by contacting the production team shortly after sending your proof corrections.

Best regards,

George

George Inglis, PhD
Senior Editor

[Research Cross-Journal Editorial Team](https://www.nature.com/nsmb/research-cross-journal-editorial-team)
Nature Structural & Molecular Biology

Reviewer #1 (Remarks to the Author):

The study by Karpinska et al expands on the work by Stik and colleagues (PMID: 32514124) that had described modest gene expression changes upon CTCF degradation, during conversion of leukemic B cells to Macrophages. Here, the authors confirmed these observations and incorporate high-resolution Micro-Capture-C (MCC), and Tri-C during the same trans-differentiation experiment. They then assess the impact of degrading CTCF on gene expression, and on pairwise and multi-way interactions.

The data presented here are of very high quality and the claims are well supported. The MCC experiments show important correlations between changes in expression and enhancer-promoter contacts, but the major novelty of the work resides in the identification by Tri-C of three-way interactions at cell-type specific enhancers that require CTCF for their formation but not for gene regulation of the genes within the hubs. At a time that the physiological role of hubs is under intense discussion, this observation suggests that regulation of gene expression of genes within hubs does not require formation of these structures.

We thank the Reviewer for their helpful and constructive feedback on our work. As detailed below, we have addressed the Reviewer's comments in our revised manuscript, which we think has significantly improved the manuscript.

I have the following main comments suggestions/comments for the authors:

1 - B cells do not differentiate to Macrophages. I think it would be important to refer to this process as transdifferentiation or cellular conversion as the Graf lab normally does. Or at least clearly mention that this is not a physiological or a model of a differentiation process.

We agree with the Reviewer and now refer to this process as "transdifferentiation" throughout the manuscript.

3 - Did all regions assessed by Tri-C show evidence of formation of enhancer-containing multi-way interactions similar to the loci shown in figure 3 and extended data figure 3? That is not possible to assess from the quantifications in figure 3f-i.

We thank the Reviewer for raising this point, which we agree is important to clarify in the manuscript. We detect multi-way interactions in the majority of the loci that we investigated, with 44/51 loci showing multi-way interactions with the enhancer viewpoints and 41/51 loci showing multi-way interactions with the CTCF binding site (CBS) viewpoints. Of note: many of the loci at which we do not detect hubs have very local interactions (which we cannot reliably distinguish from the general proximity signal) and/or relatively sparse data, which suggests that the actual proportion of tissue-specific loci containing multi-way interactions is even higher. We have added a histogram to show the distribution of the number of multi-way interactions per viewpoint in our dataset to **Extended Data Fig. 3e,f** (pasted below).

Extended Data Fig. 3: Dynamic chromatin hub formation during lymphoid-to-myeloid transdifferentiation. (e) Histogram showing the distribution of the number of multi-way interactions with the enhancer viewpoints per locus as detected by Tri-C. (f) Histogram showing the distribution of the number of multi-way interactions with the bCBS viewpoints per locus as detected by Tri-C.

4 – Whether the number of gene expression changes upon CTCF depletion is large, modest, or minor, is a matter of interpretation and expectation. Rather than using these qualitative terms it would probably be better to show how many genes are labelled as upregulated and downregulated in the plot in Extended Data Fig6. I also couldn't see a table with that list. I also think that extended data figure 6 should be in the main figure. Probably more important than the current Figure 6A.

We agree with the Reviewer that it is informative to add more quantitative information about the gene expression changes to the manuscript. We have now labelled the number of up- and downregulated genes in **Extended Data Fig. 6a** and included a list with all expression changes in **Supplementary Table 2**. We have also changed the text to include more objective, quantitative information about the gene expression changes. We would prefer not to swap the main figure and supplemental figure, because we find it informative that **Fig. 6A** shows the gene expression changes upon CTCF depletion in relation to differential gene expression over the transdifferentiation course. However, we have added more quantitative information describing **Extended Data Fig. 6a** in the main text, so readers can more easily access this information (lines 521-527; pasted below).

Lines 521-527:

*In agreement with previous reports^{54,68,69} and with the observation that BLaER1 cells still efficiently transdifferentiate into iMac3s in absence of CTCF (**Extended Data Fig. 4b**), we find that CTCF depletion has modest effects on gene expression during lymphoid-to-myeloid transdifferentiation (**Fig. 6a**). Upon CTCF depletion, 718 genes are significantly downregulated and 558 genes are significantly upregulated, with a \log_2 fold-change < 2 for 90% of the significantly differentially expressed genes (**Extended Data Fig. 6a** and **Supplementary Table 2**).*

5- Along these lines, it isn't clear to me how much the expression of genes with MCC viewpoints changes upon CTCF depletion and how much their interactions change. While Figure 2 was highly correlational, repeating the promoter and promoter-proximal CBS analyses with and without CTCF would be more functional and interesting. Throughout the paper the authors did an effort to always show a few gene examples and then quantify what happens at all viewpoints. However when it came to CTCF degradation, this was not done and instead we see just a few example loci. It would be nice to have a better understanding of how much CTCF depletion affected the loci for which viewpoints were designed.

We agree that it is informative to clarify specifically for the targeted loci how their expression levels change upon CTCF depletion. We have therefore labelled all targeted loci with

significant changes in **Extended Data Fig. 6a** and clarified this in the legend. In response to the Reviewer's previous comment, we have also included a list with all expression changes to **Supplementary Table 2**. Please note that we had already included a quantification of the MCC interaction changes upon CTCF depletion across all viewpoints in **Fig. 4f** (Fig. 4e in the previous version of the manuscript). We have clarified this in the main text (lines 407-410; pasted below).

Lines 407-410:

Systematic quantification of the MCC interactions shows similar patterns as in the described examples (Fig. 4f). Across upregulated loci, promoter-CBS interactions are weakened, enhancer-promoter interactions are relatively stable, and promoter-promoter interactions are increased after CTCF depletion.

As we do not have MCC data for promoter-proximal CBS (ppCBS) viewpoints after CTCF depletion, we are not able to perform the promoter and ppCBS analyses with and without CTCF. However, to address this point, we have analyzed the effect of the presence of a ppCBS on gene expression changes after CTCF depletion genome-wide, as discussed in the next point.

6 - Some genes are strongly impacted by CTCF degradation. It would be important to characterize these. Are they B or iMac specific genes? Do they have proximal or distal enhancers? Do they form tri-way interactions. It seems to me that viewpoint choice shouldn't have been limited by genes that are differentially expressed during cell conversion but instead also include genes that are differentially expressed upon CTCF depletion.

Following this helpful suggestion from the Reviewer, we have performed additional analyses to characterize the genes that are strongly affected by CTCF depletion. We have added these analyses to **Extended Data Fig. 6** (pasted below) and updated the main text accordingly (lines 527-551; pasted below). In **Extended Data Fig. 6b**, we show that B-cell- and iMac-specific genes are more likely to be significantly up- or downregulated compared to genes that are stable during transdifferentiation, with roughly equal representation of B-cell- and iMac-specific genes in the up- and downregulated gene groups. In **Extended Data Fig. 6c**, we show the distribution of the distances between the nearest CBS and the gene promoter for genes that are not significantly affected, upregulated, or downregulated upon CTCF depletion. This analysis shows that significantly changed genes are more likely to have a CBS close to their promoter compared to unaffected genes. In **Extended Data Fig. 6d**, we show the distribution of the distances between the enhancers and the gene promoter for genes that are not significantly affected, upregulated, or downregulated upon CTCF depletion. For this analysis, we have paired enhancers and promoters based on available TT-seq and Hi-C data (see Methods; in brief, we assigned an enhancer to a gene if they show similar changes in eRNA and gene expression levels over the differentiation course and are located in the same TAD). This analysis shows that genes that are downregulated after CTCF depletion are more likely to have long-range enhancers. In line with this, we show in **Extended Data Fig. 6e** that decreased enhancer-promoter interactions after CTCF depletion as measured by MCC are more distal compared to increased interactions. Please note that several of the targeted genes are differentially expressed upon CTCF depletion. We have included additional examples of these genes to the manuscript: in **Extended Data Fig. 6f,g**, as well as in **Fig. 6b-c**, we show significantly up- and downregulated gene loci. Please note that most targeted gene loci form multi-way interactions (see response to main comment #3) and that there is no correlation with their sensitivity to CTCF depletion.

Extended Data Fig. 6: Changes in gene expression following CTCF depletion can be explained by rewired pair-wise enhancer-promoter interactions. (a) Volcano plot showing differentially expressed genes between control-treated and CTCF-depleted cells at 96 h after differentiation induction, as measured by RNA-seq in $n = 2$ replicates. The x-axis shows the fold change (FC) in expression; the y-axis shows the adjusted P-value. The horizontal line and vertical lines indicate the significance threshold of adjusted P-value < 0.01 and effect size threshold of $\log_2FC > 0.6$ or < -0.6 , respectively. Targeted genes are highlighted and those that are significantly changed upon CTCF depletion are labelled. Targeted genes described in the text that are not labelled (e.g., *CCR1* and *NFKB1Z*) are not significantly up- or down-regulated upon CTCF depletion. (b) Comparison of the proportion of B-cell-specific genes, iMac-specific genes, and genes that are stably expressed during lymphoid-to-myeloid-transdifferentiation among the genes that are downregulated, unchanged, or upregulated upon CTCF depletion. (c) Distribution of CBS-promoter distances (of the nearest CBS) of genes that are unchanged, upregulated, or downregulated upon CTCF depletion. (d) Distribution of enhancer-promoter distances (of all paired enhancers, see Methods) of genes that are unchanged, upregulated, or downregulated upon CTCF depletion. (e) Distribution of enhancer-promoter distances of increased and decreased enhancer-promoter interactions upon CTCF depletion in targeted iMac-specific loci as identified by MCC. The grey line marks the 150 kb threshold used to classify distal enhancers in Extended Data Fig. 2c. (f) Chromatin interactions in the *IRF8* locus (chr16:85,769,160-86,069,160; 300 kb) in control and CTCF-depleted cells at 96 h after differentiation induction. From top to bottom: gene annotation; chromatin accessibility (ATAC-seq); Mediator occupancy (MED26 ChIPmentation); Cohesin occupancy (SMC1A ChIPmentation); CTCF occupancy (CTCF ChIP-seq); Micro-Capture-C (MCC) data from the viewpoint of the promoter. The axes of the profiles are scaled to signal and have the following ranges: Accessibility = 0–1776; Mediator = 0–4261; Cohesin = 0–3315; CTCF = 0–12417; MCC = 0–40. The orientations of CTCF motifs at prominent CBSs are indicated by arrowheads (forward orientation in red; reverse orientation in blue). MCC interactions with CBSs, enhancers, and promoters are annotated with cyan, magenta, and orange triangles, respectively, and prominent changes in CTCF-depleted cells are highlighted in grey. (g) Chromatin interactions in the *MAFB* locus (chr20:40,156,606-40,976,606; 820 kb) in control and CTCF-depleted cells, as described in panel

f, with prominent changes highlighted in blue. The axes of the profiles are scaled to signal and have the following ranges: Accessibility = 0–3567; Mediator = 0–5231; Cohesin = 0–2232; CTCF = 0–10212; MCC = 0–30.

Lines 527-551:

Differentially expressed genes are enriched for B-cell- and iMac-specific genes and are more likely to have a CBS near their promoter and to be regulated by distal enhancers compared to unaffected genes (**Extended Data Fig. 6b-e**). However, interestingly, we do not generally observe a significant decrease in gene expression in the upregulated gene loci in which we observe a strong impairment in chromatin hub formation, including *CCR1*, *NFKBIZ* and *TRIB1* (**Extended Data Fig. 6a**). This suggests that CTCF-dependent chromatin hubs do not have a critical role in the regulation of gene expression during cellular differentiation.

It has previously been shown that CBSs contribute to the specificity of enhancer-promoter communication by preventing interactions between enhancers and promoters across TAD borders⁷⁰. In agreement with this model, we observe that CTCF depletion leads to the formation of ectopic interactions in two of the targeted upregulated loci. In the *LMO2* locus, we observe increased interactions with the promoter of *CAPRIN1* (**Fig. 6b**); in the *KDM7A* locus, we observe increased interactions with cis-regulatory elements of the *SLC37A3* gene (**Fig. 6c**). These rewired interactions are associated with a significant increase in *CAPRIN1* and *SLC37A3* expression (**Fig. 6d,e**). In addition to the ectopic interactions in the *LMO2* and *KDM7A* loci, we find subtle changes in interaction profiles in some of the other targeted loci, which are also associated with small changes in gene expression. For example, in the *IRF8* and *MAFB* locus, both enhancer-promoter interactions and gene expression levels are slightly decreased and increased, respectively, after CTCF depletion (**Extended Data Fig. 6a,f,g**). Furthermore, consistent with the observation that downregulated genes are more likely to have distal enhancers (**Extended Data Fig. 6d**), we find that decreased enhancer-promoter interactions upon CTCF depletion are more likely to be distal (> 150 kb) from the promoter compared to increased enhancer-promoter interactions (**Extended Data Fig. 6e**).

7 – It would probably be helpful to readers to add a drawing/model on how the authors interpret their data. Especially representing the CTCF versus noCTCF condition

We thank the Reviewer for this suggestion and have included a graphical summary in **Fig. 7** (pasted below).

Fig. 7: Graphical summary. Extruding cohesin molecules are stalled at CBSs, promoters, and enhancers. In presence of CTCF (left), this leads to detectable clustering of these elements in chromatin hubs. In absence of CTCF (right), these clusters form less frequently. However, enhancers still interact with their cognate promoters in a pair-wise manner (example only shown for one of the two enhancers) and thereby maintain gene expression levels. TFs = transcription factors.

Minor issues:

1 - Although there is nothing wrong in the title, you need to read the whole paper before you understand what the authors mean. Maybe something that more directly reflects their main

finding would be more appropriate. For example, something along the lines of: Formation of CTCF-mediated cell type-specific enhancer hubs does not affect gene regulation.

We thank the Reviewer for this suggestion and understand their point. However, we think that the title suggested by the Reviewer includes a very strong statement, that we were asked to nuance by Reviewer 2 (major comment #7). We therefore prefer to keep the title as it is.

2 - In the abstract the authors highlight concepts and technical details that are hardly mentioned in their results. For example, single-allele topologies are not mentioned anywhere in the results and therefore either the concept shouldn't be used in the abstract or use the same language should be used in the results. The authors highlight base-pair resolution but in the description of the results it isn't really clear why this is important. 10 or 100bp bins would also work to distinguish promoter and promoter-proximal CBS, right? I understand that the base pair resolution is used for MCC quantification, but I don't believe the authors describe it in the results or explain its advantages. If they don't, then it shouldn't be highlighted in the abstract either. The sentence starting at line 26 is a bit ambiguous and does not clearly reflect the authors intent of saying that gene expression changes explained by rewired E-P contacts are modest. As it stands, it can also be interpreted as saying that the mild effects on gene expression can be explained by rewired E-P contacts. This is not that the authors mean, I believe.

We thank the Reviewer for their helpful feedback on the abstract. We have replaced "single-allele topologies" with "multi-way interaction analyses" and replaced the ambiguous sentence with the following two sentences: "Depletion of CTCF strongly impairs the formation of these structures. However, the effects of CTCF depletion on gene expression are modest and can be explained by rewired enhancer-promoter interactions."

The Reviewer is correct that the base-pair resolution is important for distinguishing interactions between elements in very close proximity. This makes it indeed possible to generate distinct interaction profiles from promoters and ppCBSs, but also to distinguish interactions between enhancers, promoters, and CBSs in close proximity within a given interaction profile. This is very clear when comparing the MCC data (which have base-pair resolution) in **Fig. 1d,e** with the Capture-C data (which are plotted at a resolution of single *N/a*III fragments with a median size of 132 bp) in **Extended Data Fig. 1c,d**. The Reviewer is correct that this is critical for accurate quantification. We have clarified this throughout the manuscript.

(In case the Reviewer is interested: we have included a comparison of the correlation between enhancer-promoter interaction frequencies and gene expression based on MCC data and Capture-C data in our reply to Reviewer 2, minor comment #2.)

3 - This is a comment for almost all boxplots but especially figure 2c. It would be helpful to know how many datapoints there are in each boxplot. Concerning this figure, I had a hard time following its description and I am not sure it adds much to the paper compared with the other plots in this same figure. It isn't clear to me what the differences between All and Distal only represent, and whether the presence or absence of Promoter Proximal CBS influences these interactions much. This needs to either be better summarized or potentially removed.

We agree with the Reviewer that **Fig. 2c** is not very intuitive. We have clarified this figure in the main text and in the legend. Furthermore, we have swapped this figure with **Extended Data Fig. 2c**, which makes a similar point but is easier to grasp and therefore better suited for a main figure. We have added the number of datapoints in the boxplots to all relevant figures.

4 - Figure 4E-can the authors point to a promoter-promoter interaction that increased upon CTCF depletion? I had a hard time finding one. If the genes shown don't have one maybe had to at least extended data?

We have included additional annotation to **Fig. 4b-e** (and similar figures throughout the manuscript) to highlight promoter-promoter interactions. There is an increased promoter-promoter interaction in the *TRIB1* locus (**Fig. 4d, far upstream**). We have also added an additional locus with a clearer example to **Fig. 4e** (pasted below).

Fig. 4: CTCF is not required for pair-wise enhancer-promoter interactions. (e) Chromatin interactions in the *SRGN* locus (chr10:68,884,514-69,234,514; 350 kb) in control and CTCF-depleted cells, as described in panel b. The axes of the profiles are scaled to signal and have the following ranges: Accessibility = 0–5900; Mediator = 0–1430; Cohesin = 0–777; CTCF = 0–318; MCC = 0–50.

Reviewer #2 (Remarks to the Author):

The manuscript by Karpinska, Zhu et al investigate the formation of 3D chromatin structures during cell differentiation, and their link with transcriptional regulation. For this purpose, the authors employ a B-cell leukemia cell line (BLaER1) that can be converted into induced macrophages (iMACSs) by the expression of CEBPA. At different time points of the differentiation process the authors generate state-of-the-art Micro-Capture-C and Tri-C data for a panel of loci containing B-cell-specific and iMAC-specific genes. The datasets include viewpoints on promoters and additional cis-regulatory elements, such as enhancers or CTCF-binding sites (CBS). These datasets are further complemented with binding profiles for Mediator and Cohesin, as well as with publicly available data for gene expression, chromatin accessibility and H3K27Ac and CTCF binding. This constitutes an optimal and comprehensive resource to pursue the questions that the authors pose.

Overall, the analyses performed in this differentiation system show that the dynamics enhancer-promoter interactions at activating /repressed promoters correlate well with those of Mediator and Cohesin binding. Tri-C experiments also revealed that interactions between promoters, enhancers and CBS exist in the context of 3D chromatin hubs, and that their formation and dissolution also mimics the dynamics of gene expression. The study also explores an interesting phenomenon, which is the recurrent presence of CBS near promoters. MCC data revealed that those sites display a preference to interact with enhancers than their counterparts located Topologically Associating Domains (TADs) boundaries, suggesting the existence of different functional classes of CBS. The authors further explore the role of CTCF in chromatin hub formation, by coupling the BLaER1 differentiation system with an auxin degron. These experiments revealed that CTCF impairment results has a considerable impact on 3D chromatin hub structures, but minor effects on individual enhancer-promoter interactions.

This is a well-rounded study providing high quality datasets that provide novel insights into the regulatory dynamics of differentiation systems. The derived results contribute significantly to the ongoing discussion on the instructive role of chromatin structure on gene expression. Overall, the manuscript is suited for the readership of Nature Structural and Molecular Biology. Yet, there are certain aspects of the study that require further clarification.

We thank the Reviewer for their helpful and constructive feedback on our work. As detailed below, we have addressed the Reviewer's comments in our revised manuscript, which we think has significantly improved the manuscript.

Major comments:

1) - The number of viewpoints studied in MCC/Tri-C experiments seems sufficient to inform on generalizable gene regulatory principles. In that respect, it would be recommended that the authors provide additional statistics such as how many enhancers, CBS and additional promoters compose on average the regulatory landscapes of the studied genes. Since these analyses have been performed across several stages, it would be useful to examine how the composition of the landscapes evolves over differentiation time. Further, what percentage of the individual enhancers detected in MCC experiments engage in multi-way contacts in Tri-C data?

We thank the Reviewer for raising this point, which we agree is important to clarify in the manuscript. We have quantified the number of enhancer, promoter, and CBS interactions per

viewpoint and how the interaction averages per category change during the differentiation course. We have included these data in **Extended Data Fig. 1 h-k** (pasted below).

Extended Data Fig. 1: Characterization of the BLAER1 lymphoid-to-myeloid transdifferentiation system. (h) Histogram showing the distribution of the number of enhancer interactions with the promoter viewpoints per locus as detected by MCC. (i) Histogram showing the distribution of the number of promoter interactions with the promoter viewpoints per locus as detected by MCC. (j) Histogram showing the distribution of the number of CBS interactions with the promoter viewpoints per locus as detected by MCC. (k) Overview of the average number of enhancer, promoter, and CBS interactions with the promoter viewpoints as detected by MCC at 0 h, 24 h, and 96 h.

We have also calculated the number of multi-way interactions per viewpoint, which we have included in **Extended Data Fig. 3e,f** (pasted below).

Extended Data Fig. 3: Dynamic chromatin hub formation during lymphoid-to-myeloid transdifferentiation. (e) Histogram showing the distribution of the number of multi-way interactions with the enhancer viewpoints per locus as detected by Tri-C. (f) Histogram showing the distribution of the number of multi-way interactions with the bCBS viewpoints per locus as detected by Tri-C.

We generally detect many more pair-wise interactions compared to multi-way interactions. However, since the resolution and sensitivity of MCC is much higher compared to Tri-C, we cannot draw any meaningful conclusions about the percentage of pair-wise interactions that engage in multi-way interactions. We have clarified this in the text (lines 311-313; pasted below).

Lines 311-313:

Tri-C uses the restriction enzyme NlaIII for chromatin digestion and therefore generates lower-resolution data compared to MCC, which complicates direct quantitative comparisons between these datasets.

2) - Fig 1e displays data profiles that are representative of genes undergoing repression. Mediator and MCC profiles display dynamic changes at the upstream gene region, suggesting a loss of contact with enhancer elements. However, the downstream region shows opposite dynamics: contacts are gained with other elements. This profile may suggest that interaction with other type of regulatory elements (i.e. repressors) could be in play. Are those patterns recurrently observed for other repressed genes? Is the opposite trend also observed during

the activation of genes (i.e. a loss of contacts with specific genomic regions)? Have the authors considered integrating additional epigenetic datasets that mark repression in their analysis (for example H3K27me3 from GSE257528), to gain further mechanistic insights?

We thank the Reviewer for bringing this up. Indeed, we observe increased interactions in many downregulated gene loci (60%). We do not observe the opposite trend for upregulated genes. We had not commented on these interactions in the manuscript, as we have not yet been able to characterize these elements extensively. We also thank the Reviewer for pointing us to the H3K27me3 dataset, which we were not aware of. However, none of these elements overlap with H3K27me3 peaks, as can be appreciated from the example of the *MYB* locus in **Extended Data Fig. 1d** and the larger region surrounding *MYB* and additional example below (**Rebuttal Fig. 1**). (We tried to quantify this systematically, but as none of the elements overlap with H3K27me3 peaks, this turned out to be a quantification of ChIP-seq noise and not meaningful.) Instead, we find that these elements overlap with H3K27ac and Mediator. We therefore do not think that they correspond to silencers and think that it is more likely that they function as "transient enhancers", as described by Vermunt et al. *Molecular Cell* 2023, because many of these interactions increase mid-transdifferentiation and decrease in a later stage. We have included annotation of these elements in **Fig. 1** and **Extended Data Fig. 1,2** and added the H3K27me3 data to **Extended Data Fig. 1**. We have also added a paragraph to the main text to describe these elements (lines 170-178; pasted below). We hope to be able to characterize these elements and their function in more detail in the future, by knocking them out and assessing the effects on gene expression.

Lines 170-178:

Interestingly, we also observe cis-regulatory elements that interact more frequently with the MYB promoter as it is de-activated. These elements are not characterized by repressive chromatin marks such as H3K27me3 (Extended Data Fig. 1d). Instead, they transiently gain chromatin accessibility, H3K27ac, and binding of Mediator and cohesin. We therefore speculate that these elements may function as transient enhancers that modulate the kinetics of gene silencing, as recently described in the context of erythroid differentiation⁶⁰, although further characterization is required to confirm this. We observe elements with similar characteristics in approximately 60% of the targeted downregulated loci.

Rebuttal Fig. 1a. UCSC screenshot of a wide region surrounding the MYB gene. H3K27me3 peaks do not overlap with putative transient enhancer elements, which are highlighted in red.

Rebuttal Fig. 1b. UCSC screenshot of a wide region surrounding the IRAG2 gene. H3K27me3 peaks do not overlap with putative transient enhancer elements, which are highlighted in red.

3) - The authors generate MCC profiles for promoter proximal and boundary CBS (ppCBS and bCBS respectively), finding that each category has distinct interaction preferences. I missed the contextualization of those findings in the discussion, in particular with respect to previous studies that have also pointed to the existence of different CBS categories (for example Huang et al Nat Genet 2021).

Regarding this analysis, it is also not entirely clear what were the exact criteria followed to allocate CBS in the two categories. In which category would be allocated those CBS at the promoters of genes that are close or embedded in TAD boundary? Such “intermediate” cases seem to exist in the dataset. For example, at the *TRIB1* locus from Fig1c, the gene is located in a region that may well be called as a TAD boundary, depending on the parameters of the detection algorithm. How would those “intermediate” cases behave in terms of interactions with respect to the effects observed in the ppCBS and bCBS categories?

We thank the Reviewer for pointing this out. We have now added a section to the Discussion in which we discuss the different CBS categories and their potential function in the context of the current literature (incl. Huang et al. *Nature Genetics* 2021; lines 643-657; pasted below).

Lines 643-657:

Consistent with the concept of distinct classes of functional CBSs⁶², we observe that ppCBSs interact more frequently with enhancers compared to bCBSs. In addition, we observe that genes with a ppCBS are more likely to engage in long-range interactions and to be downregulated upon CTCF depletion compared to genes without a ppCBS. These and previous observations⁶¹ suggest that ppCBSs may contribute to the formation of (distal) enhancer-promoter interactions. However, the general importance of ppCBSs for gene regulation remains unclear since we do not observe significant changes in gene expression in many of the targeted loci that contain a ppCBS and are characterized by long-range enhancer-promoter interactions. It is possible that there are subtle changes in the expression of these genes that are difficult to detect due to previously observed increased variability of gene expression in the context of cohesin and CTCF perturbations⁷⁷. In addition, it is of interest that we observe a tendency for proximal enhancers to interact more frequently with their cognate promoters upon CTCF depletion. Although speculative at this stage, this could provide a mechanism to compensate for loss of long-range CTCF-dependent interactions and thereby to buffer gene expression changes after CTCF depletion.

The Reviewer is correct that the *TRIB1* promoter is close to a boundary. (Please note that this particular boundary is not called as a TAD boundary though; it appears to be a sub-TAD boundary.) With respect to categorizing the CBSs, we have classified CBSs as promoter-proximal (pp) when they are close (< 5 kb) to a promoter regardless of whether they are also close to a boundary. The Reviewer is correct that there are a few ppCBSs that are close to a boundary; these are categorized as ppCBSs in our analysis. We have clarified this in the text (lines 880-883). Based on visual inspection, the ppCBSs close to a boundary show similar patterns in their interaction profiles as the ppCBSs that are further from a boundary.

4) - Lines 443-444: “However, interestingly, cooperative interactions involving only enhancers and promoters are also notably decreased.”

The global data from Fig. 5j supports this observation. Yet, this effect does not seem to occur at the *NFKBIZ* locus (cyan circles in Fig 5b). In this case, cooperative interactions between enhancers and promoter are largely preserved. I wonder if this effect could be explained by the lack of CBS inside the *NFKBIZ* TAD, in comparison with the *CCR1* and *TRIB1* loci. Can the authors check if there is an influence of intra-TAD CBS density on enhancer-promoter cooperative interactions?

We thank the Reviewer for this suggestion and agree that the low density of CBSs in the *NFKBIZ* locus is a likely explanation for the smaller effect of CTCF depletion on multi-way interactions at this locus compared to e.g., the *CCR1* locus. It is not very common for tissue-specific gene loci to have a low density of CBSs and we have unfortunately targeted too few of such regions to assess whether there is a general correlation between the intra-TAD CBS density and the effect of CTCF depletion on multi-way enhancer-promoter interactions. However, we have added a sentence to the main text to acknowledge this possibility (lines 451-454; pasted below).

Lines 451-454:

However, interestingly, cooperative interactions involving only enhancers and promoters are also decreased. This effect appears stronger in CBS-dense regions (e.g., CCR1) compared to regions with relatively little CTCF binding (e.g., NFKBIZ), although it is detectable across all targeted regions.

5) - Fig 4e. Does the increase of promoter-promoter interaction upon CTCF depletion occur preferentially between genes that would be otherwise located in different TADs?

We observe examples of increased promoter-promoter interactions upon CTCF depletion both within TADs and spanning TAD boundaries. We have included an example of each for the Reviewer in the figure below (**Rebuttal Fig. 2**), added an additional example locus with clearly increased promoter-promoter interactions to **Fig. 4e**, and clarified this point in the text (lines 405-407; pasted below).

Lines 405-407:

*As exemplified in the *TRIB1* and *SRGN* loci, we observe that many promoter-promoter interactions are increased after CTCF depletion, both within and beyond TAD boundaries (**Fig. 4d,e**).*

Rebuttal Fig. 2a. Example of increased intra-TAD promoter-promoter interactions (highlighted in red).

Rebuttal Fig. 2b. Example of increased inter-TAD promoter-promoter interactions (highlighted in red).

6) - The analysis on Figs 1h and I are interesting as they suggest that Mediator and Cohesin are important to support enhancer-promoter interactions. In that respect, it would have been nice to complement these analyses with ChIP-seq experiments for Mediator in Cohesin in CTCF-depleted cells and subsequent analogous correlation analyses with differential enhancer-promoter interactions against control cells.

We agree with the Reviewer and have performed ChIPmentation experiments for both Mediator and cohesin after CTCF depletion. We have included example tracks in **Fig. 4** and further analyses in **Extended Data Fig. 4c,d** (pasted below). As expected, CTCF depletion leads to a reduction of Cohesin occupancy at CBSs, as can be clearly appreciated from **Fig. 4**. However, CTCF depletion has little effect on the distribution of Mediator and cohesin at enhancers and therefore does not strongly correlate with the changes in enhancer-promoter interactions. We have clarified this in the main text (lines 412-419; pasted below).

Fig. 4: CTCF is not required for pair-wise enhancer-promoter interactions. (b) Chromatin interactions in the *CCR1* locus (chr3:45,902,299-46,427,299; 525 kb) in control and CTCF-depleted cells at 96 h after differentiation induction. From top to bottom: gene annotation; chromatin accessibility (ATAC-seq); Mediator occupancy (MED26 ChIPmentation); Cohesin occupancy (SMC1A ChIPmentation); CTCF occupancy (CTCF ChIPmentation); Micro-Capture-C (MCC) data from the viewpoint of the promoter. The axes of the profiles are scaled to signal and have the following ranges: Accessibility = 0–1565; Mediator = 0–1463; Cohesin = 0–975; CTCF = 0–468; MCC = 0–40. The orientations of CTCF motifs at prominent CBSs are indicated by arrowheads (forward orientation in red; reverse orientation in blue). MCC interactions with CBSs, enhancers, and promoters are annotated with cyan, magenta, and orange triangles, respectively. (c) Chromatin interactions in the *NFKBIZ* locus (chr3:101,766,932-102,266,932; 500 kb) in control and CTCF-depleted cells, as described in panel b. The axes of the profiles are scaled to signal and have the following ranges: Accessibility = 0–4883; Mediator = 0–760; Cohesin = 0–367; CTCF = 0–286; MCC = 0–40. (d) Chromatin interactions in the *TRIB1* locus (chr8:125,079,965-125,739,965; 660 kb) in control and CTCF-depleted cells, as described in panel b. The axes of the profiles are scaled to signal and have the following ranges: Accessibility = 0–8276; Mediator = 0–1858; Cohesin = 0–1660; CTCF = 0–468; MCC = 0–40. (e) Chromatin interactions in the *SRGN* locus (chr10:68,884,514-69,234,514; 350 kb) in control and CTCF-depleted cells, as described in panel b. The axes of the profiles are scaled to signal and have the following ranges: Accessibility = 0–5900; Mediator = 0–1430; Cohesin = 0–777; CTCF = 0–318; MCC = 0–50.

Extended Data Fig. 4: Characterization of CTCF depletion during lymphoid-to-myeloid transdifferentiation. (c) Correlation between differential enhancer-promoter interaction frequencies and differential Mediator binding levels at the interacting elements during lymphoid-to-myeloid transdifferentiation (grey datapoints; 24 h vs 0h and 96 h vs 0 h) and upon CTCF depletion (red datapoints; 96 h control vs 96 h CTCF depletion), based on Spearman's correlation test. (d) Correlation between differential enhancer-promoter interaction frequencies and differential cohesin binding levels at the interacting elements, as described in panel c.

Lines 412-419:

Given the strong correlation between differential Mediator and cohesin binding levels and enhancer-promoter interaction frequencies during differentiation, we performed additional ChIPmentation experiments after CTCF depletion to investigate whether we can explain the observed changes in interaction patterns by changes in the binding levels of Mediator and cohesin. As expected, CTCF depletion leads to a reduction in cohesin occupancy at CBSs (Fig. 4b-e). In contrast, we observe minor changes in the distribution of Mediator and cohesin at enhancers, which do not correlate well with the minor changes we observe in enhancer-promoter interactions after CTCF depletion (Extended Data Fig. 4c,d).

7) - Lines 567-569: "...but that the higher-order configuration of enhancers and promoters in CTCF-dependent hub structures does not have a specific function in gene regulation".

This is a strong statement that cannot be supported by the data included in the manuscript. One cannot simply exclude that the alterations in chromatin hubs have also certain contribution on the observed changes in transcription. Furthermore, gene regulation is more complex than what can be inferred by simply looking at expression levels in a individual timepoint. For example, temporal changes in gene activation or alterations in spatial patterns are aspects that cannot be measured in this differentiation system, and that may well be compromised by the impairment of chromatin hubs. I would suggest that the authors tone down this statement.

We agree with the Reviewer and have toned down this statement. In addition, we have added a section to the discussion to clarify that the loss of CTCF and CTCF-dependent chromatin hubs may impact gene regulation and cell functioning in ways that we have not assessed in this manuscript (lines 672-676; pasted below).

Lines 672-676:

Furthermore, it is possible that CTCF-dependent hubs have a more pronounced function in specific cellular contexts or in regulating dynamic aspects of gene expression that are not reflected in RNA-seq data. In this regard, it is of interest that it has previously been shown that CTCF-depleted iMacs have impairments in their acute inflammatory response⁵⁴.

Minor comments:

1) - Line 129: "Characterization of this system by qPCR and FACS shows that cellular

differentiation occurs synchronously and completely, with >90% of B-cells converting into iMacs over the differentiation course”

This percentage is not correct, based on the flow cytometry profile at 168 h.

We thank the Reviewer for spotting this mistake. We have corrected it; the sentence now reads *"Characterization of this system by qPCR and FACS shows that cellular differentiation occurs synchronously and completely, with approximately 90% of B-cells converting into iMacs over the differentiation course"*.

2) - Capture-C experiments were also performed in this study. Yet, the only reference in the manuscript are a few profiles in Extended Data Fig 1c and d. It would be appropriate to refer to these experiments in the main text, explaining the motivation to perform them as well as discussing the results obtained.

We thank the Reviewer for pointing this out. We performed the Capture-C experiments before we had established a robust MCC protocol in our lab in Göttingen. When we were able to generate high-quality MCC data, we realized that MCC data would be very beneficial for this project, as these data would allow us to analyze changes in the interaction profiles of *cis*-regulatory elements with more precision. As shown in **Fig. 1d,e** and **Extended Data Fig. 1c,d**, the MCC and Capture-C data generally show the same trends of increased and decreased interactions in up- and downregulated gene loci, respectively. However, the MCC data reveal more details and can be quantified across loci more reliably. This is clear when calculating the correlation between changes in gene expression and enhancer-promoter interaction frequencies. While analysis based on the MCC data results in a correlation coefficient $R = 0.821$ (**Fig. 6f**), the correlation coefficient based on Capture-C data equals $R = 0.14$ (**Rebuttal Fig. 3**).

Rebuttal Fig 3. Correlation between differential expression levels and total enhancer-promoter interaction frequencies based on Capture-C data (24 h vs 0h and 96 h vs 0 h). $R = 0.14$ (Spearman's correlation test).

We apologize for not properly referring to the Capture-C data in the main text previously and have now added references in lines 140 and 153.

3) - It would be useful to add the CTCF ChIP-seq profiles of the different timepoints in Fig 2 and Extended Data Fig. 2, as it is shown for the other experiments in these figures.

We thank the Reviewer for noticing this and have added the CTCF ChIP-seq profiles for the different timepoints to **Fig. 2** and **Extended Data Fig. 2**.

4) - It could be good for the flow of the text to provide background information on the modeling framework in the main text. Alternatively, the authors may indicate in the main text that the principles of the model are extensively described in the Methods section.

We have provided additional background information about the modeling framework in the Methods section (lines 950-972; pasted below) and a clear reference to this section in the main text (line 462).

Lines 950-972:

2 Mb regions around the CCR1, NFKBIZ, and TRIB1 loci were modelled based on a previously described modelling framework^{66,67} using the multipurpose EspressoMD package¹⁰⁷. Briefly, the chromatin fiber is modelled as a self-avoiding polymer chain consisting of equisized beads representing 2 kb of chromatin. Beads were classified as binding the transcription machinery (i.e., promoters and enhancers; based on Mediator ChIPmentation peaks), binding CTCF (based on CTCF ChIPmentation peaks, including motif orientation), as transcribed genic regions (gene bodies of active genes; based on TT-seq data), or as neutral (with none of the above-mentioned features). Polymers were used in molecular dynamics simulations in a 3D space following Langevin equations to model thermal motion of chromatin and its binding factors in an implicit solvent (the nucleoplasm), with the following postulations⁶⁶: (1) the transcription machinery has affinity for promoters and enhancers and traverses gene bodies; (2) components of the transcription machinery (RNAPII, Mediator) also have affinity for each other to simulate condensate formation; (3) cohesin complexes can bind anywhere on the polymer and extrude loops, but have a preference for loading at enhancers and promoters. To account for the experimentally observed accumulation of cohesin at CTCF- and Mediator-bound sites, extrusion of loops by cohesin stalls when encountering a CTCF-bound site with a motif oriented towards the direction of extrusion or when meeting the transcription machinery. By dynamically forming and dissolving protein-chromatin bonds, this framework simulates chromatin loop extrusion by the cohesin complex and traversing of RNAPII during transcription. The rates of loop extrusion and RNAPII translocation along chromatin were set within the range of experimentally deduced values (RNAPII: 1-5 kb/min; cohesin: 15-30 kb/min)⁶⁷. RNAPII-cohesin and cohesin-cohesin crossing rates were set at 1.5 and 0.15 crossings per second, respectively.

Reviewer #3 (Remarks to the Author):

The work by Karpinska et al. combines Micro-Capture-C, Tri-C, and simulations to investigate the interplay between gene expression and 3D genome structure over the course of cell differentiation. This is a challenging problem, with a literature showing contradictory evidence supporting alternative theories. The study is remarkable in that it produced rich and high-resolution information clarifying how CTCF and cohesin shape the structure of chromatin and support the formation of enhancer-promoter contacts that likely explain changes in gene expression. One major finding to me is the observation of a clear correlation between enhancer-promoter contacts and gene expression level, likely made possible by the high resolution of their data, which is important considering that the role of 3D chromatin conformation for gene regulation is still being questioned in the literature. Another is the observation that the minor effect of CTCF degradation on gene expression is due to the fact that the original co-operative 3-way contacts favored by stable stalling at CBS is compensated by pairwise contacts in a way that the overall strength of enhancer-promoter interactions remains mostly unaffected. While these observations are perhaps not entirely surprising, I believe that they are very important for both the chromatin and gene regulation fields. Overall, I find the methodology and analysis appropriate and the conclusions robust.

I would like to comment mainly on the simulation methods and their presentation. I find that the simulations are critical to rationalize the observations that CTCF depletion affects 3-way contact while leaving pairwise enhancer-promoter interactions unaffected. For this, the specific way in which CTCF, cohesin and RNAPol are treated is fundamental, but this is not clarified in the paper itself. I believe adding more information about the model is important to explain what is actually going on inside the real system. My major issue about the modeling results is that the authors do not show evidence from the model that CTCF depletion would be expected to leave the pairwise enhancer-promoter contacts not affected, as only the effect on 3-way interactions is actually reported. A more detailed analysis of the simulations would therefore be helpful.

We thank the Reviewer for their helpful and constructive feedback on our work. We agree that the simulations are an important part of the manuscript and have added more detailed information to describe the modeling framework (lines 950-972; pasted below). The Reviewer also raises an important point with respect to the effect of CTCF depletion on pair-wise interactions in the model. Consistent with our experimental data, the model shows that CTCF depletion leads to a reduction of multi-way interactions but has no significant effect on the pair-wise interactions. To clarify this, we have added a comparison of quantified pair-wise interactions and multi-way interactions in the three modeled loci based on both the experimental data and the model (**Extended Data Fig. 5d-g**; pasted below).

Lines 950-972:

2 Mb regions around the CCR1, NFKBIZ, and TRIB1 loci were modelled based on a previously described modelling framework^{66,67} using the multipurpose EspressoMD package¹⁰⁷. Briefly, the chromatin fiber is modelled as a self-avoiding polymer chain consisting of equisized beads representing 2 kb of chromatin. Beads were classified as binding the transcription machinery (i.e., promoters and enhancers; based on Mediator ChIPmentation peaks), binding CTCF (based on CTCF ChIPmentation peaks, including motif orientation), as transcribed genic regions (gene bodies of active genes; based on TT-seq data), or as neutral (with none of the above-mentioned features). Polymers were used in molecular dynamics simulations in a 3D space following Langevin equations to model thermal motion of chromatin and its binding factors in an implicit solvent (the nucleoplasm), with the following postulations⁶⁶: (1) the transcription machinery has affinity for promoters and enhancers and traverses gene bodies;

(2) components of the transcription machinery (RNAPII, Mediator) also have affinity for each other to simulate condensate formation; (3) cohesin complexes can bind anywhere on the polymer and extrude loops, but have a preference for loading at enhancers and promoters. To account for the experimentally observed accumulation of cohesin at CTCF- and Mediator-bound sites, extrusion of loops by cohesin stalls when encountering a CTCF-bound site with a motif oriented towards the direction of extrusion or when meeting the transcription machinery. By dynamically forming and dissolving protein-chromatin bonds, this framework simulates chromatin loop extrusion by the cohesin complex and traversing of RNAPII during transcription. The rates of loop extrusion and RNAPII translocation along chromatin were set within the range of experimentally deduced values (RNAPII: 1-5 kb/min; cohesin: 15-30 kb/min)⁶⁷. RNAPII-cohesin and cohesin-cohesin crossing rates were set at 1.5 and 0.15 crossings per second, respectively.

Extended Data Fig. 5: CTCF supports the formation of enriched multi-way interactions in chromatin hubs. (d) Interaction frequencies of the promoters of modelled gene loci (*CCR1*, *NFKBIZ*, and *TRIB1*) with enhancers and CBSs at 96 h in control and CTCF-depleted cells, derived from experimental MCC data. Boxplots show the interquartile range (IQR) and median of the data; whiskers indicate the minima and maxima within 1.5 * IQR; asterisks indicate significance ($P < 0.01$, two-sided paired Wilcoxon signed rank test). (e) Interaction frequencies of the promoters of modelled gene loci (*CCR1*, *NFKBIZ*, and *TRIB1*) with enhancers and CBSs at 96 h in control and CTCF-depleted cells, extracted from the models. Boxplots as described in panel d. (f) Multi-way interaction frequencies of E-E-P and E-C-X hubs in the modelled gene loci at 96 h in control and CTCF-depleted cells, derived from experimental Tri-C data. Boxplots as described in panel d. (g) Multi-way interaction frequencies of E-E-P and E-C-X hubs in the modelled gene loci at 96 h in control and CTCF-depleted cells, extracted from the models. Boxplots as described in panel d.